# Online Time Series Forecasting with Theoretical Guarantees

**Zijian Li**[1,2]    **Changze Zhou**[3]    **Minghao Fu**[5,2]    **Sanjay Manjunath**[2]    **Fan Feng**[2,5]
**Guangyi Chen**[1,2]    **Yingyao Hu**[4,†]    **Ruichu Cai**[3]    **Kun Zhang**[1,2,†]

[1]Carnegie Mellon University
[2]Mohamed bin Zayed University of Artificial Intelligence
[3]Guangdong University of Technology
[4] Johns Hopkins University
[5] University of California San Diego

## Abstract

This paper is concerned with online time series forecasting, where unknown distribution shifts occur over time, i.e., latent variables influence the mapping from historical to future observations. To develop an automated way of online time series forecasting, we propose a **T**heoretical framework for **O**nline **T**ime-series forecasting (**TOT** in short) with theoretical guarantees. Specifically, we prove that supplying a forecaster with latent variables tightens the Bayes risk—the benefit endures under estimation uncertainty of latent variables and grows as the latent variables achieve a more precise identifiability. To better introduce latent variables into online forecasting algorithms, we further propose to identify latent variables with minimal adjacent observations. Based on these results, we devise a model-agnostic blueprint by employing a temporal decoder to match the distribution of observed variables and two independent noise estimators to model the causal inference of latent variables and mixing procedures of observed variables, respectively. Experiment results on synthetic data support our theoretical claims. Moreover, plug-in implementations built on several baselines yield general improvement across multiple benchmarks, highlighting the effectiveness in real-world applications.

## 1 Introduction

Time series data stream in sequentially like an endless tide. Online time-series forecasting [Anava et al., 2013, Pham et al., 2023, Lau et al.] aims to leverage recent $\tau$ observed variables $\mathbf{x}_{t-\tau:t}$ to predict the future segment $\mathbf{x}_{t+1:T}$. In real-world scenarios, each observation $\mathbf{x}_t$ may be partially observed, while a set of unobserved latent variables $\mathbf{z}_t$ governs the evolving relationship between past and future. These latent variables introduce unknown distribution shifts [Zhang et al., 2024b, Zhao and Shen, 2024], i.e., the conditional distribution $p(\mathbf{x}_{t+1:T} \mid \mathbf{x}_{t-\tau:t})$ changes over time, resulting in suboptimal performance of time series forecasting algorithms. Therefore, how to adapt to these distribution shifts automatically is a key challenge for online time series forecasting.

To overcome this challenge, different methods are proposed to handle the time-varying distributions. Specifically, some methods devise different model architectures to adapt to nonstationarity. For example, Pham et al. [2022] considers that the fast adaptation capability of neural networks can handle the distribution changes, so they propose the fast and slow learning networks to balance fast adaptation to recent changes while preserving the old knowledge. Another similar idea is yee Ava Lau et al. [2025], which harnesses slow and fast streams for coarse predictions and near-future forecasting, respectively. Another solution focuses on concept drift estimation. For example, Zhang et al. [2024b] detects the temporal distribution shift and updates the model in an aggressive way. Considering that

---

[†]Corresponding authors.

39th Conference on Neural Information Processing Systems (NeurIPS 2025).

the ground-truth future values are delayed until after the forecast horizon, Zhao and Shen [2024] first estimates the concept drift and then incorporates it into forecasters. And Cai et al. [2025] estimates the nonstationarity with sparsity constraint by assuming the temporal distribution shifts are led by the unknown interventions. Please refer to Appendix A for more discussion of the related works.

Despite demonstrating empirical gains in mitigating temporal shifts, these methods leave some theoretical questions unanswered. Specifically, existing approaches incorporating distribution shifts into model architectures [Pham et al., 2023, yee Ava Lau et al., 2025, Zhang et al., 2024b, Zhao and Shen, 2024] offer few theoretical guarantees explaining why conditioning on such estimated shifts yields a systematic reduction in forecasting error. Although recent work Cai et al. [2025] provides theoretical results, it often imposes strict conditions on the data generation process, i.e., nonstationarity is led by interventions on latent variables. Moreover, the uncertainty inherent in distribution shift estimation is rarely considered in the generalization bounds of error risk, leaving the discrepancy between empirical and ground-truth generalization risks uncharacterized. Furthermore, how to shrink the aforementioned gap theoretically and empirically remains underexplored. Therefore, a general theoretical framework for online time-series forecasting is urgently needed.

In this paper, we propose a **T**hretical framework for **O**nline **T**ime-series forecasting (**TOT**) with theoretical guarantees. Specifically, we first consider a general time series generation process, where the temporal distribution shifts are led by latent variables $\mathbf{z}_t$. Sequentially, we show that conditioning the forecaster on $\mathbf{z}_t$ tightens the Bayes risk; the reduction persists under estimation noise and improves as identifiability sharpens. We further show that both the latent variables and their causal dynamics can be identified using only four adjacent observations, yielding a concrete, minimal-data criterion. Building on these theoretical results, we devise a model-agnostic blueprint with a temporal decoder to match the marginal distributions of observations and two independent noise estimators to model the temporal dynamics of observed and latent variables. Experiment results on synthetic data verify our theoretical results, and plug-in versions atop different baselines achieve general improvement across several benchmarks. The key contributions of our work are summarized as follows:

- We establish formal risk-bound guarantees for online forecasting under latent-driven distribution shift, furnishing explicit theoretical guidance to enhance forecasting performance.
- We prove that both latent states and their temporal causal dynamics can be uniquely identified from only four consecutive observations.
- We propose a plug-and-play model architecture and achieve general improvement on several benchmark datasets of online time series forecasting.

## 2 Problem Setup

We first introduce a data generation process as shown in Figure 1. Specifically, we let $\mathbf{x}_t = \{x_{t,1}, \cdots, x_{t,n}\}$ be a $n-$dimension random vector that represents the observations. We further assume that they are generated by the historical observations $\mathbf{x}_{t-1}$, hidden variables $\mathbf{z}_t = \{z_{t,1}, \cdots, z_{t,n}\}$, and independent noise $\epsilon_t^{\mathbf{o}}$ via a nonlinear mixing function $\mathbf{g}$. Moreover, the latent variables $\mathbf{z}_{t,i}$ is generated by the time-delayed parents $\mathrm{Pa}_d(z_{t,i})$, instantaneous parents $\mathrm{Pa}_e(z_{t,i})$, and independent noise $\epsilon_{t,i}^{\mathbf{z}}$ via latent causal influence $\mathbf{f}_i$. Putting them together, the data generation process can be formulated as Equation (1).

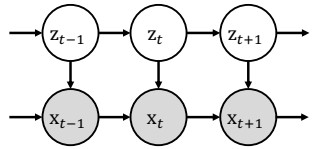

Figure 1: Illustration of the generation process for time series data, where the mapping of observed variables $\mathbf{x}_{t-1} \to \mathbf{x}_t$ is influenced by latent variables $\mathbf{z}_t$.

$$\underbrace{\mathbf{x}_t = \mathbf{g}(\mathbf{z}_t, \mathbf{x}_{t-1}, \epsilon_{t,i}^{\mathbf{o}}),}_{\text{Nonlinear Mixing Procedure}} \quad \underbrace{z_{t,i} = \mathbf{f}_i(\mathrm{Pa}_d(z_{t,i}), \mathrm{Pa}_e(z_{t,i}), \epsilon_{t,i}^{\mathbf{z}}),}_{\text{Latent Causal Influence}} \quad \epsilon_{t,i}^{\mathbf{o}} \sim p_{\epsilon_{t,i}^{\mathbf{o}}}, \quad \epsilon_{t,i}^{\mathbf{z}} \sim p_{\epsilon_{t,i}^{\mathbf{z}}}. \quad (1)$$

To better understand our theoretical results, we provide the definitions of subspace-wise [Von Kügelgen et al., 2021] and component-wise identifiability [Kong et al., 2022]. Please refer to Appendix B for the description of the notations.

**Definition 1** (**Block-wise Identifiability of Latent Variables** $\mathbf{z}_t$). *The block-wise identifiability of $\mathbf{z}_t \in \mathbb{R}^n$ means that for ground-truth $\mathbf{z}_t$, there exists $\hat{\mathbf{z}}_t$ and an invertible function $H : \mathbb{R}^n \to \mathbb{R}^n$, such that $\mathbf{z}_t = H(\hat{\mathbf{z}}_t)$.*

**Definition 2** (**Component-wise Identifiability of Latent Variables** $z_{t,i}$). *The component-wise identifiability of $\mathbf{z}_t \in \mathbb{R}^n$ is that for each $\mathbf{z}_{t,i}, i \in \{1, \cdots, n\}$, there exists a corresponding estimated component $\hat{\mathbf{z}}_{t,j}, j \in \{1, \cdots, n\}$ and an invertible function $h_i : \mathbb{R} \to \mathbb{R}$, such that $z_{t,i} = h_i(\hat{z}_{t,j})$.*

# 3 Theoretical Framework for Online Time Series Forecasting

Based on the aforementioned data generation process, we present theoretical results for online time series forecasting. Specifically, we first demonstrate that incorporating latent variables $\mathbf{z}_t$ reduces forecasting risk, and that improved identifiability of these latent variables narrows the gap between estimated and ground-truth risk (Theorem 1). To ensure identifiability, we first identify the joint distribution of $\mathbf{x}_t$ and $\mathbf{z}_t$ by matching the marginal distribution of four consecutive observations (Theorem 2). Subsequently, by imposing a sparse constraint on the estimated mixing procedure, i.e. $\hat{\mathbf{z}} \to \hat{\mathbf{x}}$, we further establish component-wise identifiability of the latent variables $\mathbf{z}_t$ (Theorem 3).

## 3.1 Predictive-Risk Analysis

We begin with the predictive risk analysis regarding different types of inputs of a time series forecaster.

**Theorem 1.** (*Predictive-Risk Reduction via Temporal Latent Variables*) *Let $\mathbf{x}_t, \mathbf{z}_t$, and $\hat{\mathbf{z}}_t$ be the observed variables, ground-truth latent variables, and the estimated latent variables, respectively. We let $\mathbf{x}_{t-\tau:t} = \{\mathbf{x}_{t-\tau}, \cdots, \mathbf{x}_t\}$ be the historical $(\tau+1)$-step observed variables. Moreover, we let $\mathcal{R}_\mathbf{o}, \mathcal{R}_\mathbf{z}$, and $\mathcal{R}_{\hat{\mathbf{z}}}$ be the expected mean squared error for the models that consider $\{\mathbf{x}_{t-\tau:t}\}, \{\mathbf{x}_{t-\tau:t}, \mathbf{z}_t\}$, and $\{\mathbf{x}_{t-\tau:t}, \hat{\mathbf{z}}_t\}$, respectively. Then, in general, we have $\mathcal{R}_\mathbf{o} \geq \mathcal{R}_{\hat{\mathbf{z}}} \geq \mathcal{R}_\mathbf{z}$, and if $\mathbf{z}_t$ is identifiable we have $\mathcal{R}_\mathbf{o} > \mathcal{R}_{\hat{\mathbf{z}}} = \mathcal{R}_\mathbf{z}$.*

**Discussion:** The proof can be found in Appendix C.1. This risk bound can be derived by leveraging the law of total expectation and the law of total variance to decompose the expected mean squared error. Intuitively, Theorem 1 highlights the critical role latent variables play in reducing predictive risk for online time series forecasting. Specifically, it reveals three distinct scenarios:

First, if the ground-truth latent variables $\mathbf{z}_t$ have no influence on the observed variables $\mathbf{x}_t$, the predictive risk remains unchanged irrespective of whether latent variables are considered, i.e., $\mathcal{R}_\mathbf{o} = \mathcal{R}_{\hat{\mathbf{z}}} = \mathcal{R}_\mathbf{z}$. However, this scenario rarely occurs in practice because $\mathbf{z}_t$ having no influence on the mapping $\mathbf{x}_{t-\tau:t} \to \mathbf{x}_{t+1}$ implies that the observed time series data is stationary, and it is typically challenging to collect all relevant observations without any influence from exogenous variables.

Second, when latent variables do influence observed variables, incorporating the ground-truth latent variables reduces the predictive risk compared to using observations alone, leading to $\mathcal{R}_\mathbf{o} > \mathcal{R}_\mathbf{z}$. Practically, however, ground-truth latent variables $\mathbf{z}_t$ are unknown, and we can only access the estimated latent variables $\hat{\mathbf{z}}_t$. If these latent variables are fully identifiable, the estimated latent variables can achieve the same risk reduction as the true latent variables, resulting in $\mathcal{R}_\mathbf{o} > \mathcal{R}_{\hat{\mathbf{z}}} = \mathcal{R}_\mathbf{z}$.

Third, if the estimated latent variables $\hat{\mathbf{z}}_t$ partially identify the ground-truth latent variables, meaning there exists at least one dimension $j \in 1, \cdots, n$ for which, for all $i \in 1, \cdots, n$, no function $h_i$ satisfies $\mathbf{z}_{t,i} = h_i(\hat{\mathbf{z}}_{t,j})$. Consequently, the estimated latent variables $\hat{\mathbf{z}}_t$ capture only certain aspects of the temporal dynamics while omitting others due to incomplete identifiability, resulting in $\mathcal{R}_\mathbf{o} > \mathcal{R}_{\hat{\mathbf{z}}} > \mathcal{R}_\mathbf{z}$. In the worst case, when the estimated latent variables are completely non-identifiable (e.g., $\hat{\mathbf{z}}_t$ are purely random noise), we have $\mathcal{R}_\mathbf{o} = \mathcal{R}_{\hat{\mathbf{z}}} > \mathcal{R}_\mathbf{z}$.

Based on the aforementioned discussion, we have the following two takeaway conclusions.

  i. *Incorporating latent variables to characterize distribution shifts helps reduce forecasting error.*
  ii. *The more accurately the latent variables are identified, the lower the predictive risk.*

## 3.2 Identify Joint Distribution of Latent and Observed Variables

Based on the aforementioned results, we further propose to identify the latent variables from observations. By leveraging four consecutive observed variables, i.e., $\mathbf{x}_{t-2}, \mathbf{x}_{t-1}, \mathbf{x}_t$, and $\mathbf{x}_{t+1}$, we can find that the block $(\mathbf{x}_t, \mathbf{z}_t)$ is block-wise identifiable. For a better explanation of these results, we first introduce the definition of the linear operator as follows:

**Definition 3** (**Linear Operator** Hu and Schennach [2008], Dunford and Schwartz [1971]). *Consider two random variables $\mathbf{a}$ and $\mathbf{b}$ with support $\mathcal{A}$ and $\mathcal{B}$, the linear operator $L_{\mathbf{b}|\mathbf{a}}$ is defined as a*

*mapping from a probability function $p_{\mathbf{a}}$ in some function space $\mathcal{F}(\mathcal{A})$ onto the probability function $p_{\mathbf{b}} = L_{\mathbf{b}|\mathbf{a}} \circ p_{\mathbf{a}}$ in some function space $\mathcal{F}(\mathcal{B})$,*

$$\mathcal{F}(\mathcal{A}) \to \mathcal{F}(\mathcal{B}) : p_{\mathbf{b}} = L_{\mathbf{b}|\mathbf{a}} \circ p_{\mathbf{a}} = \int_{\mathcal{A}} p_{\mathbf{b}|\mathbf{a}}(\cdot|\mathbf{a}) p_{\mathbf{a}}(\mathbf{a}) d\mathbf{a}. \tag{2}$$

**Theorem 2.** *(**Block-wise Identification under 4 Adjacent Observed Variables.**) Suppose that the observed and latent variables follow the data generation process. By matching the true joint distribution of 4 adjacent observed variables, i.e., $\{\mathbf{x}_{t-2}, \mathbf{x}_{t-1}, \mathbf{x}_t, \mathbf{x}_{t+1}\}$, we further consider the following assumptions:*

- *A1 (Bound and Continuous Density): The joint distribution of $\mathbf{x}, \mathbf{z}$ and their marginal and conditional densities are bounded and continuous.*

- *A2 (Injectivity): There exists observed variables $\mathbf{x}_t$ such that for any $\mathbf{x}_t \in \mathcal{X}_t$, there exist a $\mathbf{x}_{t-1} \in \mathcal{X}_{t-1}$ and a neighborhood [2] $\mathcal{N}^r$ around $(\mathbf{x}_t, \mathbf{x}_{t-1})$ such that, for any $(\bar{\mathbf{x}}_t, \bar{\mathbf{x}}_{t-1}) \in \mathcal{N}^r$, $L_{\bar{\mathbf{x}}_t, \mathbf{x}_{t+1} | \mathbf{x}_{t-2}, \bar{\mathbf{x}}_{t-1}}$ is injective; $L_{\mathbf{x}_{t+1}|\mathbf{x}_t, \mathbf{z}_t}$, $L_{\mathbf{x}_t|\mathbf{x}_{t-2}, \mathbf{x}_{t-1}}$ is injective for any $\mathbf{x}_t \in \mathcal{X}_t$ and $\mathbf{x}_{t-1} \in \mathcal{X}_{t-1}$, respectively.*

- *A3 (Uniqueness of Spectral Decomposition) For any $\mathbf{x}_t \in \mathcal{X}_t$ and any $\bar{\mathbf{z}}_t \neq \tilde{\mathbf{z}}_t \in \mathcal{Z}_t$, there exists a $\mathbf{x}_{t-1} \in \mathcal{X}_{t-1}$ and corresponding neighborhood $\mathcal{N}^r$ satisfying Assumption A2 such that, for some $(\bar{\mathbf{x}}_t, \bar{\mathbf{x}}_{t-1}) \in \mathcal{N}^r$ with $\bar{\mathbf{x}}_t \neq \mathbf{x}_t, \bar{\mathbf{x}}_{t-1} \neq \mathbf{x}_{t-1}$:*

  - *i $k(\mathbf{x}_t, \bar{\mathbf{x}}_t, \mathbf{x}_{t-1}, \bar{\mathbf{x}}_{t-1}, \bar{\mathbf{z}}_t) < C < \infty$ for any $\mathbf{z}_t \in \mathcal{Z}_t$ and some constant $C$.*
  - *ii $k(\mathbf{x}_t, \bar{\mathbf{x}}_t, \mathbf{x}_{t-1}, \bar{\mathbf{x}}_{t-1}, \bar{\mathbf{z}}_t) \neq k(\mathbf{x}_t, \bar{\mathbf{x}}_t, \mathbf{x}_{t-1}, \bar{\mathbf{x}}_{t-1}, \tilde{\mathbf{z}}_t)$, where*

$$k(\mathbf{x}_t, \bar{\mathbf{x}}_t, \mathbf{x}_{t-1}, \bar{\mathbf{x}}_{t-1}, \mathbf{z}_t) = \frac{p_{\mathbf{x}_t|\mathbf{x}_{t-1}, \mathbf{z}_t}(\mathbf{x}_t \mid \mathbf{x}_{t-1}, \mathbf{z}_t) p_{\mathbf{x}_t|\mathbf{x}_{t-1}, \mathbf{z}_t}(\bar{\mathbf{x}}_t \mid \bar{\mathbf{x}}_{t-1}, \mathbf{z}_t)}{p_{\mathbf{x}_t|\mathbf{x}_{t-1}, \mathbf{z}_t}(\bar{\mathbf{x}}_t \mid \mathbf{x}_{t-1}, \mathbf{z}_t) p_{\mathbf{x}_t|\mathbf{x}_{t-1}, \mathbf{z}_t}(\mathbf{x}_t \mid \bar{\mathbf{x}}_{t-1}, \mathbf{z}_t)}. \tag{3}$$

*Suppose that we have learned $(\hat{\mathbf{g}}, \hat{\mathbf{f}}, p_{\hat{\epsilon}})$ to achieve Equations (1), then the combination of Markov state $\mathbf{z}_t, \mathbf{x}_t$ is identifiable, i.e., $[\hat{\mathbf{z}}_t, \hat{\mathbf{x}}_t] = H(\mathbf{z}_t, \mathbf{x}_t)$, where $H$ is invertible and differentiable.*

**Implication and Proof Sketch.** This theory is based on Hu and Shum [2012] and it shows that the block $(\mathbf{x}_t, \mathbf{z}_t)$ can be identified via consecutive observations. Please find the proof in Appendix C.2. Although both [Fu et al., 2025] and our theory employ the technique of eigenvalue-eigenfunction decomposition, our result is a general case of Fu et al. [2025], which demonstrates the identifiability results under the assumption that there is no causal relationship between the observed variables. Moreover, we further relax the monotonicity and normalization assumption (Please find the details of this assumption in Appendix C.4) in Hu and Shum [2012], which requires that the function form of $p(\mathbf{x}_{t+1}|\mathbf{z}_t, \mathbf{x}_t) \to \mathbf{z}_t$ is known. This adjustment allows our conclusion to better align with real-world time series data.

The proof can be summarized into the following three steps. First, under the data generation process in Figure 1, we establish the relationship between the linear operators acting on the observed and latent variables. Sequentially, by introducing the neighborhood of $(\mathbf{x}_{t-1}, \mathbf{x}_t)$, we derive an eigenvalue-eigenfunction decomposition for the observations, accounting for different transition probabilities of observed variables. Finally, by leveraging the uniqueness property of spectral decomposition (Theorem XV.4.3.5 Dunford and Schwartz [1971]), we demonstrate that the latent variables are block-wise identifiable when the marginal distribution of the observed variables is matched.

**Connection with Previous Results.** To better understand our results and highlight the contributions of our work in comparison to previous studies, we further discuss the different intuitions between our work and prior works. In Fu et al. [2025], due to the absence of causal edges between the observed variables $\mathbf{x}_t$, conditioning on $\mathbf{z}_t$, the three consecutive observations (i.e., $\mathbf{x}_{t-1}, \mathbf{x}_t, \mathbf{x}_{t+1}$) are conditionally independent, which allows us the use different observations to describe the variance of latent variable. Meanwhile, because there are causal edges between the observed variables, and the latent variable $\mathbf{z}_t$ influences the mapping $\mathbf{x}_{t-1} \to \mathbf{x}_t$, we use different transitions of observations, i.e., $p(\mathbf{x}_t|\mathbf{x}_{t-1}, \mathbf{z}_t)$ to describe the variance of $\mathbf{z}_t$, making the identifiability of latent variables possible.

**Discussion on Assumptions.** We provide a detailed explanation of the assumptions to clarify their real-world implications and enhance the understanding of our results.

---

[2]Please refer to Appendix C.5 for the definition of neighborhood.

First, the assumption of the bound and continuous conditional densities is a standard assumption in the works of identifiability of latent variables [Hu and Schennach, 2008, Hu, 2008, Fu et al., 2025]. It implies that the transition probability densities of observed and latent variables are bounded and continuous. This assumption is easy to meet in practice, for instance, the temperature records in the climate data are changing continuously within a reasonable range.

Second, according to Definition 3, the linear operator $L_{\mathbf{b}|\mathbf{a}}$ denotes the mapping from $p_{\mathbf{a}}$ to $p_{\mathbf{b}}$, and the injectivity of the linear operator means that there is enough variation in the density of $b$ for different distributions of $a$. For example, $L_{\mathbf{x}_t,\mathbf{x}_{t+1}|\mathbf{x}_{t-1},\mathbf{x}_{t-2}}$ means that the historical observations have sufficient influence on the future observations, i.e., the historical values of temperature should have a strong influence on the future values. Please refer to Appendix C.3 for more examples of the injectivity of linear operators.

Third, the third assumption implies that the changing of latent variables $\mathbf{z}_t$ produces a visibly different influence on the mapping from $\mathbf{x}_{t-1}$ to $\mathbf{x}_t$. Concretely, we let $\rho_{\mathbf{z}_t,1} = \frac{p_{\mathbf{x}_t|\mathbf{x}_{t-1},\mathbf{z}_t}(\mathbf{x}_t|\mathbf{x}_{t-1},\mathbf{z}_t)}{p_{\mathbf{x}_t|\mathbf{x}_{t-1},\mathbf{z}_t}(\mathbf{x}_t|\mathbf{x}_{t-1}^-,\mathbf{z}_t)}$ and $\rho_{\mathbf{z}_t,2} = \frac{p_{\mathbf{x}_t|\mathbf{x}_{t-1},\mathbf{z}_t}(\bar{\mathbf{x}}_t|\mathbf{x}_{t-1},\mathbf{z}_t)}{p_{\bar{\mathbf{x}}_t|\bar{\mathbf{x}}_{t-1},\mathbf{z}_t}(\mathbf{x}_t|\mathbf{x}_{t-1}^-,\mathbf{z}_t)}$ be the probability of transitioning rate from two different historical state (i.e., $\mathbf{x}_{t-1}$ and $\bar{\mathbf{x}}_{t-1}$) to the same current states ($\mathbf{x}_t$ or $\bar{\mathbf{x}}_t$). Assumption A3 says that for any two different values of latent values, there exist different historical states $\mathbf{x}_{t-1}, \bar{\mathbf{x}}_{t-1}$, so that at least one of these ratios changes. Intuitively, this means the latent variable must have a sufficiently large influence on $\mathbf{x}_{t-1} \rightarrow \mathbf{x}_t$. For example, if $\mathbf{z}_t$ encodes the information of seasons and $\mathbf{x}_t$ denotes the temperature, then the typical day-to-day temperature jump in winter versus summer will differ markedly. Since A3 asks for the existence of any two such states, it is easy to be satisfied in several real-world scenarios with continuous $\mathbf{x}_t$. More discussion can be found in Appendix C.6.

### 3.3 Component-wise Identification of Latent Variables

Based on the block-wise identifiability of $(\mathbf{z}_t, \mathbf{x}_t)$ from Theorem 2, we further show that $\mathbf{z}_t$ is component-wise identifiable. For a better understanding of our results, we first provide the definition of the **Intimate Neighbor Set** [Li et al., 2024b, Zhang et al., 2024a] as follows.

**Definition 4** (Intimate Neighbor Set Li et al. [2024b], Zhang et al. [2024a]). *Consider a Markov network $\mathcal{M}_U$ over variables set $U$, and the intimate neighbor set of variable $u_{t,i}$ is*

$$\Psi_{\mathcal{M}_{\mathbf{u}_t}}(u_{t,i}) \triangleq \{ u_{t,j} \mid u_{t,j} \text{ is adjacent to } u_{t,i},$$
$$\text{and it is also adjacent to all other neighbors of } u_{t,i}, \ u_{t,j} \in \mathbf{u}_t \backslash \{u_{t,i}\} \}$$

Based on this definition, we consider $\mathbf{u}_t = \{\mathbf{z}_{t-1}, \mathbf{x}_{t-1}, \mathbf{z}_t, \mathbf{x}_t\}$, then the identifiability can be ensured with the help of historical information $(\mathbf{x}_{t-1}, \mathbf{z}_{t-1})$ and the sparse mixing procedure.

**Theorem 3.** *(Component-wise Identification of $\mathbf{z}_t$ under sparse mixing procedure.) For a series of observations $\mathbf{x}_t \in \mathbb{R}^n$ and estimated latent variables $\hat{\mathbf{z}}_t \in \mathbb{R}^n$ with the corresponding process $\hat{\mathbf{f}}_i, \hat{p}(\epsilon), \hat{g}$, suppose the marginal distribution of observed variables is matched. Let $\mathcal{M}_{\mathbf{u}_t}$ be the Markov network over $\mathbf{u}_t \triangleq \{\mathbf{z}_{t-1}, \mathbf{x}_{t-1}, \mathbf{z}_t, \mathbf{x}_t\}$. Besides the common assumptions like smooth, positive density, and sufficient variability assumptions in [Li et al., 2025], we further assume:*

- *A4 (Sparse Mixing Procedure): For any $\mathbf{z}_{t,i} \in \mathbf{z}_t$, the intimate neighbor set of $\mathbf{z}_{t,i}$ is an empty set.*

*When the observational equivalence is achieved with the minimal number of edges of the estimated mixing procedure, there exists a permutation $\pi$ of the estimated latent variables, such that $\mathbf{z}_{t,i}$ and $\hat{\mathbf{z}}_{t,\pi(i)}$ is one-to-one corresponding, i.e., $\mathbf{z}_{t,i}$ is component-wise identifiable.*

**Proof Sketch and Connection with Existing Results.** Proof can be found in the Appendix C.7. For simplicity, we omit the subscript in conditional distributions. We exploit the sparsity of the mixing procedure from $\mathbf{z}_t$ to $\mathbf{x}_t$. Concretely, if a latent component $\mathbf{z}_{t,i}$ does not contribute to an observed variable $\mathbf{x}_{t,j}$, where $i, j \in \{1, \cdots, n\}$, there is no causal edge between them. In the Markov network over $\mathbf{u}_t$, implying the conditional independence of $z_{t,i} \perp\!\!\!\perp x_{t,j} | \mathbf{u}_t \backslash \{z_{t,i}, x_{t,j}\}, \forall i, j \in \{1, \cdots, n\}$. This conditional independence implies that the corresponding mixed second-order partial derivative of $p(\mathbf{z}_t, \mathbf{x}_t | \mathbf{z}_{t-1}, \mathbf{x}_{t-1})$ is zero, i.e. $\frac{\partial^2 p(\mathbf{z}_t, \mathbf{x}_t | \mathbf{z}_{t-1}, \mathbf{x}_{t-1})}{\partial z_{t,i} \partial x_{t,j}} = 0$, which can further yields a linear system via sufficient changes assumptions in Li et al. [2025] and further results in identifiability of $\mathbf{z}_t$. More discussion can be found in Appendix C.8.

While our argument leverages the same technique used in Li et al. [2025], deriving zero-derivative conditions from sparsity in the Markov network, our contribution is orthogonal in three main respects: **1) Noisy Mixing Procedure.** Li et al. [2025] assume an invertible, noise-free mixing from latent to observed variables. In contrast, we allow additional noise in the mixing procedure $\mathbf{z}_t \rightarrow \mathbf{x}_t$, thereby accommodating measurement error in $\mathbf{x}_t$. Since the real-world datasets always contain measurement uncertainty, our model explicitly accounts for observation noise, making our theoretical results suitable for real-world scenarios. **2)**

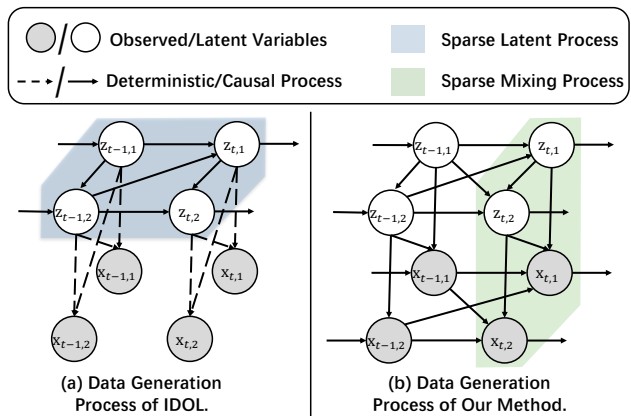

(a) Data Generation Process of IDOL.

(b) Data Generation Process of Our Method.

Figure 2: Illustration of the difference between sparse causal influence and sparse mixing procedure.

**Temporal relations among Observations.** Previous methods on temporal causal representation learning Li et al. [2025], Yao et al. [2021, 2022] usually assume that there are no causal links between $\mathbf{x}_{t-1}$ and $\mathbf{x}_t$ due to the difficulties of reconstructing $\mathbf{z}_t$ directly. However, our method allows the temporal relationships among observed variables, which is more suitable for real-world scenarios.

**3) Different Sparsity Assumptions.** Although IDOL Li et al. [2024b] and LSTD Cai et al. [2025] also leverage the sparsity assumption, we would like to highlight that these sparsity assumptions are different. Specifically, IDOL imposes sparsity constraints on the latent process as shown in Figure 2 (a). We instead assume sparsity in the mixing procedure from $\mathbf{z}_t$ to $\mathbf{x}_t$ as shown in Figure 2 (b). According to the data generation process shown in Equation (1), there is no instantaneous relations within $\mathbf{x}_t$, the intimate set of any $z_{t,i}$ are more easy to be empty, implying that the sparsity condition in our work is simpler to be met in practice.

## 4   Model-Agnostic Blueprint

The aforementioned results tell us the importance of introducing the latent variables $\mathbf{z}_t$ into online time series forecasting and how to identify them theoretically. To show how to identify the latent variables empirically, we further devise a general neural architecture as shown in Figure 3, where the gray blocks can be replaced with different backbone networks by decomposing the forecasting models into encoders and forecasters. We let $\mathbf{x}_{1:t}$ and $\mathbf{x}_{t+1:T}$ be historical and future observation, respectively. Since the proposed architecture is built upon the variational autoencoder [Kingma, 2013, Zhang et al., 2018, Blei et al., 2017], we first derive the evidence lower bound (ELBO) in Equation (4).

$$ELBO = \underbrace{\mathbb{E}_{q(\mathbf{z}_{1:T}|\mathbf{x}_{1:t})} \ln p(\mathbf{x}_{1:T}|\mathbf{z}_{1:T})}_{L_r \text{ and } L_y}$$
$$- \underbrace{\beta D_{KL}(q(\mathbf{z}_{1:T}|\mathbf{x}_{1:t})||p(\mathbf{z}_{1:T}))}_{L_{KL}^{\mathbf{z}}}, \quad (4)$$

where $\beta$ denotes the hyper-parameters and $D_{KL}$ is the Kullback–Leibler divergence. $q(\mathbf{z}_{1:T}|\mathbf{x}_{1:t})$ is used to approximate the prior distribution and implemented by

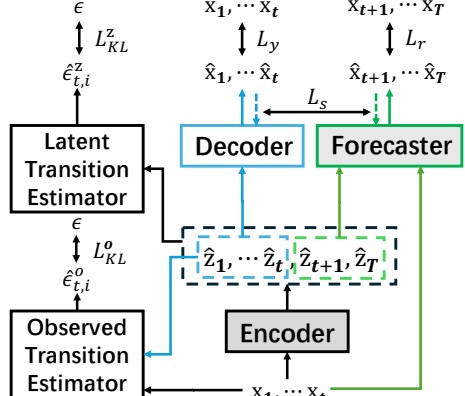

Figure 3: Illustration of the proposed plug-and-play architecture, where the gray blocks (encoder and forecaster) can be replaced with different backbones. The dashed lines denote the backpropagation.

encoders. $p(\mathbf{x}_{1:T}|\mathbf{z}_{1:T})$ is used to reconstruct the historical observation $\mathbf{x}_{1:t}$ and predict the future observation $\mathbf{x}_{t+1:T}$, which are implemented by decoders and forecasters. The encoder $\phi(\cdot)$ and decoder $\psi(\cdot)$ can be formalized as:.

$$\hat{\mathbf{z}}_{1:T} = \phi(\mathbf{x}_{1:t}), \quad \hat{\mathbf{x}}_{1:t} = \psi(\hat{\mathbf{z}}_{1:t}). \quad (5)$$

And the details of the forecaster can be found in Subsection 4.2.

## 4.1 Latent and Observed Transition Estimators

To estimate the prior distribution of latent variables $\mathbf{z}_t$, we further model the transitions $\mathbf{z}_{t-1} \to \mathbf{z}_t$ and $\mathbf{z}_t, \mathbf{x}_{t-1} \to \mathbf{x}_t$ by estimating the independent noise $\hat{\epsilon}_t^{\mathbf{z}}$ and $\hat{\epsilon}_t^{\mathbf{o}}$, respectively. For the latent transition, we first let $r_i^{\mathbf{z}}$ be a set of trainable inverse transition function that take $z_{t,i}$ and $\mathbf{z}_{t-1}$ as input and estimate the noise term $\hat{\epsilon}_{t,i}^{\mathbf{z}}$, i.e., $\hat{\epsilon}_{t,i}^{\mathbf{z}} = r_i^{\mathbf{z}}(z_{t,i}, \mathbf{z}_{t-1})$. And each $r_i$ is implemented by Multi-layer Perceptron networks (MLPs). As a result, we can develop a transformation $\kappa_{\mathbf{z}} : (\hat{\mathbf{z}}_{t-1}, \hat{\mathbf{z}}_t) \to (\hat{\mathbf{z}}_{t-1}, \hat{\epsilon}_t^{\mathbf{z}})$, whose Jacobian is $\mathbf{J}_{\kappa_{\mathbf{z}}} = \begin{pmatrix} \mathbb{I} & 0 \\ \mathbf{J}_d^{\mathbf{z}} & \mathbf{J}_e^{\mathbf{z}} \end{pmatrix}$, where $\mathbf{J}_d^{\mathbf{z}} = \mathrm{diag}\left(\frac{\partial r_i^{\mathbf{z}}}{\partial \hat{z}_{t-1,i}}\right)$ and $\mathbf{J}_e^{\mathbf{z}} = \mathrm{diag}\left(\frac{\partial r_i^{\mathbf{z}}}{\partial \hat{z}_{t,i}}\right)$. Sequentially, we have Equation (6) via the change of variables formula.

$$\log p(\hat{\mathbf{z}}_t, \hat{\mathbf{z}}_{t-1}) = \log p(\hat{\mathbf{z}}_{t-1}, \hat{\epsilon}_t^{\mathbf{z}}) + \log |\frac{\partial r_i^{\mathbf{z}}}{\partial z_{t,i}}|. \tag{6}$$

According to the generation process, the noise $\hat{\epsilon}_{t,i}^{\mathbf{z}}$ should be independent of $\mathbf{z}_{t-1}$, so we can enforce the independence of the estimated noise term $\hat{\epsilon}_{t,i}^{\mathbf{z}}$. And Equation (6) can be further rewritten as

$$\log p(\hat{\mathbf{z}}_{1:T}) = \log p(\hat{z}_1) + \sum_{\tau=2}^{T} \left( \sum_{i=1}^{n} \log p(\hat{\epsilon}_{\tau,i}^{\mathbf{z}}) + \sum_{i=1}^{n} \log |\frac{\partial r_i^{\mathbf{z}}}{\partial z_{t,i}}| \right), \tag{7}$$

where $p(\hat{\epsilon}_{\tau,i})$ is assumed a Gaussian distribution. Details of prior derivation are in Appendix D.1.3.

According to the ELBO in Equation (4), we can model the mapping from $\mathbf{z}_{1:T}$ to $\mathbf{x}_{1:T}$ via any neural architecture directly. But the temporal relations might be omitted since they are not explicitly modeled, resulting a suboptimal performance. Therefore, we employ a similar solution to model the transition of observed variables. Specifically, we use another MLP to implement $r_i^{\mathbf{o}}$ to estimate noise term $\hat{\epsilon}_{t,i}^{\mathbf{o}}$, i.e., $\hat{\epsilon}_{t,i}^{\mathbf{o}} = r_i^{\mathbf{o}}(\hat{\mathbf{z}}_t, \mathbf{x}_{t-1}, \mathbf{x}_{t,i})$. Therefore, we can devise another transformation $\kappa_{\mathbf{o}} : (\hat{\mathbf{z}}_t, \mathbf{x}_{t-1}, \mathbf{x}_t) \to (\hat{\mathbf{z}}_t, \mathbf{x}_{t-1}, \hat{\epsilon}_t^{\mathbf{o}})$ and the corresponding Jacobian $\mathbf{J}_{\kappa} = \begin{pmatrix} \mathbb{I} & 0 & 0 \\ 0 & \mathbb{I} & 0 \\ * & * & \mathbf{J}^{\mathbf{o}} \end{pmatrix}$, where $\mathbf{J}^{\mathbf{o}} = \mathrm{diag}\left(\frac{\partial r_i^{\mathbf{o}}}{\partial x_{t,i}}\right)$. And we further have Equation (8):

$$\log p(\mathbf{x}_t, \mathbf{x}_{t-1}, \hat{\mathbf{z}}_t) = \log p(\mathbf{x}_{t-1}, \hat{\mathbf{z}}_t, \hat{\epsilon}_t^{\mathbf{o}}) + \log |\frac{\partial r_i^{\mathbf{o}}}{\partial x_{t,i}}| \Rightarrow \log p(\mathbf{x}_t | \mathbf{x}_{t-1}, \hat{\mathbf{z}}_t) = \log p(\hat{\epsilon}_t^{\mathbf{o}}) + \log |\frac{\partial r_i^{\mathbf{o}}}{\partial x_{t,i}}| \tag{8}$$

As a results, we can model the relationship among $\hat{\mathbf{z}}_t$, $\mathbf{x}_{t-1}$, and $\mathbf{x}_t$ by enforcing the independence of the estimated noise term $\hat{\epsilon}_{t,i}^{\mathbf{o}}$. In practice, we assume $p(\hat{\epsilon}_{\tau,i}^{\mathbf{o}})$ also follows a Gaussian distribution, so we employ a prior Gaussian distribution to constrain $p(\hat{\epsilon}_{\tau,i}^{\mathbf{o}})$, which is denoted by $L_{KL}^{\mathbf{o}}$.

## 4.2 Observation Residual Forecaster and Sparsity Constraint on Mixing Procedure

According to Theorem 1, we should leverage the historical observations and the estimated latent variables for forecasting. Hence, different from previous methods [Cai et al., 2023, Fu et al., 2025] that only use latent variables, we devise a "residual" forecaster network as shown in Equation (9).

$$\hat{\mathbf{x}}_{t+1:T} = \varphi(\hat{\mathbf{z}}_{t+1:T}, \eta(\mathbf{x}_{1:t})), \tag{9}$$

where $\eta$ is an MLP for dimensionality reduction. Guided by the results of Theorem 3, the mixing procedure is assumed to be sparse. However, without further constraints, the estimated mixing structure induced by the MLP-based architecture $r_i^{\mathbf{o}}$ may be dense since we only restrict the independence of the estimated noise. Moreover, the spurious causal edges may lead to the incorrect estimation of $p(\mathbf{x}_{1:T})$, which further results in the suboptimal forecasting performance. To solve this problem, we propose to enforce the sparsity of the estimated mixing procedure by applying L1 penalty on the partial derivative of $\hat{\mathbf{x}}_t$ with respect to $\hat{\mathbf{z}}_t$, which are shown as follows:

$$L_s = \sum_{t=1}^{T} \sum_{i,j \in \{1,\cdots,n\}} \left| \frac{\partial \hat{x}_{t,i}}{\partial \hat{z}_{t,j}} \right|_1. \tag{10}$$

Table 1: Experiment results on simulation data for identifiability and forecasting evaluation.

| Method | TOT | IDOL | TDRL | | Inputs | Ours ($\mathbf{x}_t$ and $\hat{\mathbf{z}}_t$) | Upper Bound ($\mathbf{x}_t$ and $\mathbf{z}_t$) | Baseline ($\mathbf{x}_t$) |
|---|---|---|---|---|---|---|---|---|
| A | **0.9258(0.0034)** | 0.3788(0.0245) | 0.3572(0.0523) | | A | 0.5027(0.0045) | **0.4978(0.0002)** | 0.7005(0.0088) |
| B | **0.9324(0.0078)** | 0.8593(0.0092) | 0.8073(0.0786) | | B | 0.4395(0.0097) | **0.4195(0.0017)** | 0.6532(0.0062) |

(a) MCC results for identifiability evaluation.      (b) MSE results for forecasting evaluation.

Table 2: Mean Square Error (MSE) results on the different datasets and different backbones.

| Models | Len | LSTD | LSTD+ TOT | proceed-T | proceed-T+ TOT | OneNet | OneNet+ TOT | OneNet-T | OneNet-T+ TOT | Online-T | Online-T+ TOT |
|---|---|---|---|---|---|---|---|---|---|---|---|
| ETTh2 | 1 | 0.377 | **0.374** | 1.537 | **1.001** | 0.380 | **0.361** | 0.411 | **0.364** | 0.502 | **0.385** |
| | 24 | 0.543 | **0.532** | 2.908 | **2.360** | 0.532 | **0.525** | 0.772 | **0.691** | 0.830 | **0.733** |
| | 48 | 0.616 | **0.564** | 4.056 | **3.956** | 0.609 | **0.562** | 0.806 | **0.773** | 1.183 | **0.874** |
| ETTm1 | 1 | **0.081** | **0.081** | 0.106 | **0.102** | **0.082** | **0.082** | 0.082 | **0.077** | 0.085 | **0.077** |
| | 24 | **0.102** | 0.107 | 0.531 | **0.530** | 0.098 | **0.096** | 0.212 | **0.154** | 0.258 | **0.221** |
| | 48 | **0.115** | 0.117 | 0.704 | **0.697** | 0.108 | **0.098** | 0.223 | **0.177** | 0.283 | **0.246** |
| WTH | 1 | 0.153 | **0.153** | 0.346 | **0.313** | 0.156 | **0.151** | 0.171 | **0.150** | 0.206 | **0.145** |
| | 24 | 0.136 | **0.116** | 0.707 | **0.703** | 0.175 | **0.149** | 0.293 | **0.263** | 0.308 | **0.265** |
| | 48 | 0.157 | **0.152** | 0.959 | **0.956** | 0.200 | **0.158** | 0.310 | **0.263** | 0.302 | **0.276** |
| ECL | 1 | 2.112 | **2.038** | 3.270 | **3.131** | 2.351 | **2.301** | 2.470 | **2.211** | 3.309 | **2.172** |
| | 24 | 1.422 | **1.390** | 5.907 | **5.852** | 2.074 | **2.000** | 4.713 | **4.671** | 11.339 | **4.381** |
| | 48 | **1.411** | 1.413 | 7.192 | 7.403 | 2.201 | **2.143** | 4.567 | **4.445** | 11.534 | **4.574** |
| Traffic | 1 | 0.231 | **0.229** | 0.333 | **0.312** | 0.241 | **0.222** | 0.236 | **0.211** | 0.334 | **0.212** |
| | 24 | 0.398 | **0.397** | 0.413 | **0.410** | 0.438 | **0.354** | 0.425 | **0.401** | 0.481 | **0.386** |
| | 48 | 0.426 | **0.421** | 0.454 | **0.454** | 0.473 | **0.377** | 0.451 | **0.407** | 0.503 | **0.413** |
| Exchange | 1 | **0.013** | **0.013** | 0.012 | **0.009** | 0.017 | **0.015** | 0.031 | **0.017** | 0.113 | **0.010** |
| | 24 | 0.039 | **0.037** | 0.129 | **0.102** | 0.047 | **0.030** | 0.060 | **0.042** | 0.116 | **0.035** |
| | 48 | 0.043 | **0.042** | 0.267 | **0.195** | 0.062 | **0.039** | 0.065 | **0.051** | 0.168 | **0.040** |

Finally, the total loss of the proposed method can be summarized as follows:

$$L_{\text{total}} = L_y + \alpha L_r - \beta(L_{KL}^{\mathbf{z}} - L_{KL}^{\mathbf{o}}) + \gamma L_s, \tag{11}$$

where $\alpha, \beta$, and $\gamma$ are hyper-parameters.

## 5 Experiment

### 5.1 Synthetic Experiment

**Data Generation** We generate the simulated time series data with the fixed latent and observed causal process. We provide different synthetic datasets with different time lags, different dimensions of latent variables, latent structure, and mixing structure, respectively. Due to the page limitation, we put other synthetic experiment results in Appendix D.1.5 and report the results of Dataset A and B in the main text. Dataset A strictly follows the data generation process in Figure 1. For dataset B, we do not consider the causal influence of $\mathbf{x}_{t-1}$ when generating $\mathbf{x}_t$, i.e., $\mathbf{x}_t = \mathbf{g}(\mathbf{z}_t, \epsilon_t^{\mathbf{o}})$. Please refer to Appendix D.1.1 for the details of the synthetic data and implementation details of our method.

**Baselines.** To evaluate our theoretical claims, we consider two different tasks. Specifically, to evaluate Theorem 2 and 3, we consider the identifiability performance of latent variables with Mean Correlation Coefficient (MCC) metric and choose IDOL Li et al. [2025] and TDRL Yao et al. [2022] as baselines. To evaluate Theorem 1, we consider the performance of forecasting with Mean Square Error (MSE) metric. We choose MLPs as a forecasting model. Moreover, we let the models with only $\mathbf{x}_t$, with $\mathbf{x}_t, \hat{\mathbf{z}}_t$, and with $\mathbf{x}_t, \mathbf{z}_t$ be baselines, our method and upper bound, respectively. We repeat each method over three different random seeds and report the mean and standard deviation.

**Results and Discussion.** Experiment results of the simulation datasets are shown in Table 1. The identification results in Table 1 (a) show that 1) the proposed TOT can well identify the latent variables, while the other methods can hardly achieve it since they do not model the temporal causal inference of observations. 2) Even when the temporal causal inference of observations is omitted, TOT still achieves a better performance, since IDOL and TDRL are not suitable to the noisy mixing procedure. The forecasting results in Table 1 (b) also align with the results from Theorem 1. Specifically, the MSE results of our method and upper bound method outperform than that of baseline, i.e., only considering the historical observations, demonstrating the importance of latent variables. We also find that our forecasting results are close to but weaker than that of the upper bound method, indirectly reflecting that the proposed method can well identify the latent variables.

Table 3: Mean Absolute Error (MAE) results on the different datasets and different backbones.

| Models | Len | LSTD | LSTD+TOT | proceed-T | proceed-T+TOT | OneNet | OneNet+TOT | OneNet-T | OneNet-T+TOT | Online-T | Online-T+TOT |
|---|---|---|---|---|---|---|---|---|---|---|---|
| ETTh2 | 1 | 0.347 | **0.346** | 0.447 | **0.401** | 0.348 | **0.346** | 0.374 | **0.350** | 0.436 | **0.348** |
| | 24 | 0.411 | **0.390** | 0.659 | **0.615** | 0.407 | **0.403** | 0.511 | **0.508** | 0.547 | **0.490** |
| | 48 | 0.423 | **0.420** | 0.767 | **0.728** | 0.436 | **0.435** | 0.543 | **0.521** | 0.589 | **0.530** |
| ETTm1 | 1 | **0.187** | **0.187** | 0.190 | **0.185** | 0.187 | **0.184** | 0.191 | **0.183** | 0.214 | **0.180** |
| | 24 | **0.217** | 0.237 | 0.447 | **0.442** | 0.225 | **0.224** | 0.319 | **0.288** | 0.381 | **0.346** |
| | 48 | **0.249** | **0.249** | 0.521 | **0.505** | 0.238 | **0.229** | 0.371 | **0.312** | 0.403 | **0.369** |
| WTH | 1 | 0.200 | **0.200** | 0.143 | **0.138** | 0.201 | **0.192** | 0.221 | **0.198** | 0.276 | **0.189** |
| | 24 | 0.223 | **0.207** | 0.382 | **0.376** | 0.225 | **0.229** | 0.345 | **0.333** | 0.367 | **0.326** |
| | 48 | 0.242 | **0.229** | 0.493 | **0.491** | 0.279 | **0.239** | 0.356 | **0.340** | 0.362 | **0.336** |
| ECL | 1 | 0.226 | **0.221** | 0.286 | **0.282** | 0.254 | **0.253** | 0.411 | **0.302** | 0.635 | **0.214** |
| | 24 | 0.292 | **0.278** | 0.387 | **0.380** | 0.333 | **0.333** | 0.513 | **0.403** | 1.196 | **0.324** |
| | 48 | 0.294 | **0.289** | 0.431 | **0.422** | **0.348** | 0.357 | 0.534 | **0.402** | 1.235 | **0.343** |
| Traffic | 1 | 0.225 | **0.224** | 0.268 | **0.256** | 0.240 | **0.221** | 0.236 | **0.203** | 0.284 | **0.200** |
| | 24 | 0.316 | **0.313** | 0.291 | **0.289** | 0.346 | **0.300** | 0.346 | **0.313** | 0.385 | **0.298** |
| | 48 | 0.332 | **0.328** | **0.308** | 0.309 | 0.371 | **0.318** | 0.355 | **0.322** | 0.380 | **0.320** |
| Exchange | 1 | 0.070 | **0.069** | 0.063 | **0.051** | 0.085 | **0.078** | 0.117 | **0.085** | 0.169 | **0.055** |
| | 24 | 0.132 | **0.129** | 0.211 | **0.189** | 0.148 | **0.118** | 0.166 | **0.137** | 0.213 | **0.124** |
| | 48 | **0.142** | **0.142** | 0.300 | **0.260** | 0.170 | **0.137** | 0.173 | **0.152** | 0.258 | **0.132** |

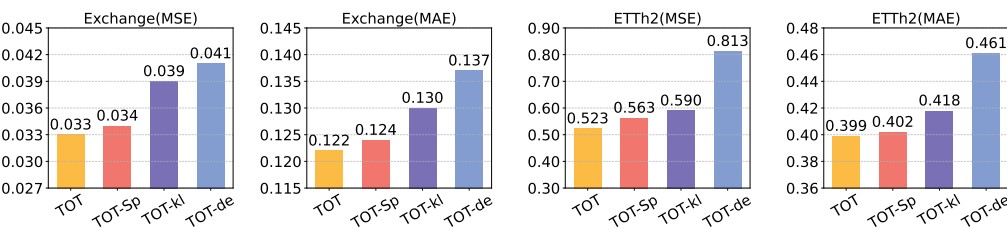

Figure 4: Ablation study on the Exchange and ETTh2 datasets.

## 5.2 Real-world Experiment

## 5.3 Experiment Setup

**Datasets** We follow the setting of Wen et al. [2023] and consider the following datasets. **ETT** is an electricity transformer temperature dataset, which contains two separate datasets {ETTh2, ETTm1}. **Exchange** is the daily exchange rate dataset from eight foreign countries. **Weather** is recorded at the Weather Station at the Max Planck Institute for Biogeochemistry in Germany. **ECL** is an electricity-consuming load dataset with the electricity consumption. **Traffic** is a dataset of traffic speeds collected from the California Transportation Agencies Performance Measurement System.

**Baseline**: We consider the following methods as our backbone networks: OneNet Wen et al. [2024] (including two model variants), Procced-T (TCN is abbreviated as T) Zhao and Shen [2024], Online-T Zinkevich [2003], and LSTD Cai et al. [2025]. For each method, we only modify the model architecture and keep the training strategy fixed. We repeat each method over three random seeds. Since some methods report the best results on the original paper, we show the best results on the main text. We also provide the experiment results with mean and variance over three random seeds in Appendix D.2.4. Please refer to Appendix D.2.2 and D.2.3 for implementation details and more experiment results based on other backbone networks, respectively.

## 5.4 Results and Discussions

Experiment results of different benchmarks under different backbone networks are shown in Table 2 and 3. According to the experiment results, we can draw the following conclusions: (1) Our framework consistently outperforms all backbones on most datasets, demonstrating its effectiveness in real-world scenarios. (2) The gains with the OneNet backbone are larger than with LSTD or Proceed, since OneNet models only dependencies of observed variables, whereas LSTD and Proceed already leverage some information of distribution shifts led by latent variables. (3) Compared to LSTD, our improvements are smaller on a few tasks like ECL, this is because LSTM employs a similar neural architecture to identify latent variables. However, we still match or exceed its performance on

most benchmarks since our method is suitable for more flexible and realistic real-world scenarios. We further devise three model variants named TOT-Sp, TOT-kl, and TOT-de, which remove the sparsity constraint $L_s$, KL divergence $L_{KL}$, and the reconstruction loss $L_r$, respectively. Experiment results in Figure 4 reflect the necessity of each loss term and model architecture.

# 6  Conclusion

This paper presents a general theoretical and practical framework for online time-series forecasting under distribution shifts caused by latent variables. We first show that introducing these latent variables yields a provable reduction in Bayes risk, and that this benefit scales with the precision of latent-state identifiability. We then show that both the latent states and their underlying causal transitions can be uniquely identified from just four consecutive observations, under mild injectivity and variability conditions. Building on these results, we design a plug-and-play architecture with an additional temporal decoder and two independent noise estimators. Extensive experiments on synthetic and real-world benchmarks confirm our theoretical guarantees and demonstrate consistent improvements over several baselines. Future work aims to extend these results to related tasks, such as causal discovery on time series data and video understanding. **Limitation:** Our theoretical results assume fully observed, discretely sampled data. Extending this framework to continuous, irregular, or multi-rate sampling time series data remains an important and meaningful direction.

# 7  Acknowledgment

The authors would like to thank the anonymous reviewers for helpful comments and suggestions during the reviewing process. The authors would like to acknowledge the support from NSF Award No. 2229881, AI Institute for Societal Decision Making (AI-SDM), the National Institutes of Health (NIH) under Contract R01HL159805, and grants from Quris AI, Florin Court Capital, and MBZUAI-WIS Joint Program, and the Al Deira Causal Education project. Moreover, this research was supported in part by National Science and Technology Major Project (2021ZD0111501), National Science Fund for Excellent Young Scholars (62122022), Natural Science Foundation of China (U24A20233, 62206064, 62206061, 62476163, 62406078), Guangdong Basic and Applied Basic Research Foundation (2024A1515011901, 2023A04J1700, 2023B1515120020).

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

# "Online Time Series Forecasting with Theoretical Guarantees"

Appendix organization:

# A  Related Works

## A.1  Time Series Forecasting

Recent advancements in time series forecasting have been driven by the application of deep learning techniques, which have proven to be highly effective in this area. These methods can be broadly categorized into several groups. First, Recurrent Neural Networks (RNNs) are commonly used for capturing temporal dependencies by leveraging their recursive structure and memory to model hidden state transitions over time [Graves and Graves, 2012, Lai et al., 2018, Salinas et al., 2020]. Another popular approach is based on Temporal Convolutional Networks (TCNs), which employ a shared convolutional kernel to model hierarchical temporal patterns and extract relevant features [Bai et al., 2018, Wang et al., Wu et al., 2022]. Additionally, simpler yet highly effective methods, such as Multi-Layer Perceptrons (MLP) [Oreshkin et al., 2019, Zeng et al., 2023, Zhang et al., 2022, Li et al., 2024a] and state-space models [Gu et al., 2022, 2021b,a], have also been utilized in forecasting tasks. Among these, Transformer-based models have emerged as particularly noteworthy, demonstrating significant progress in the time series forecasting domain [Kitaev et al., 2020, Liu et al., 2021, Wu et al., 2021, Zhou et al., 2021]. Despite the success of these methods, they are generally designed for offline data processing, which limits their applicability to real-time, online training scenarios.

## A.2  Online Time Series Forecasting

The rapid growth of training data and the need for real-time updates have made online time series forecasting more popular than offline methods [Anava et al., 2013, Liu et al., 2016, Gultekin and Paisley, 2018, Aydore et al., 2019]. Recent approaches include Pan et al. [2024], which uses structural consistency regularization and memory replay to retain temporal dependencies, and Luan et al. [2024], which applies tensor factorization for low-complexity online updates. Additionally, Mejri et al. [2024] addresses nonlinear forecasting by mapping low-dimensional series to high-dimensional spaces for better adaptation. Online forecasting is widely used in practice due to continuous data and frequent concept drift. Models are trained over multiple rounds, where they predict and incorporate new observations to refine performance. Recent work, such as [Pham et al., 2022, Cai et al., 2025, yee Ava Lau et al., 2025] and Wen et al. [2024], focuses on optimizing fast adaptation and information retention. However, simultaneously adapting to new data while retaining past knowledge can lead to suboptimal results, highlighting the need to decouple long- and short-term dependencies for improved predictions. However, most of these methods rarely explore the theoretical guarantees for online time series forecasting.

## A.3  Continual Learning

Our work is also related to continual learning. Continual learning is an emerging field focused on developing intelligent systems that can sequentially learn tasks with limited access to prior experience [Lopez-Paz and Ranzato, 2017]. A key challenge in continual learning is balancing the retention of knowledge from current tasks with the flexibility to learn future tasks, known as the stability-plasticity dilemma [Lin, 1992, Grossberg, 2013]. Inspired by neuroscience, various continual learning algorithms have been developed. This approach aligns with the needs of online time series forecasting, where continuous learning allows models to update in real time as new data arrives, improving their ability to adapt to changing data dynamics and enhancing forecasting accuracy.

## A.4  Causal Representation Learning

To ensure the identifiability of latent variables, Independent Component Analysis (ICA) has been widely used for causal representation learning [Yao et al., 2023, Schölkopf et al., 2021, Liu et al., 2023, Gresele et al., 2020]. Traditional ICA methods assume a linear mixing function between latent and observed variables [Comon, 1994, Hyvärinen, 2013, Lee and Lee, 1998, Zhang and Chan, 2007], but this is often impractical. To address this, studies have proposed assumptions for nonlinear ICA, such as sparse generation processes and auxiliary variables [Zheng et al., 2022, Hyvärinen and Pajunen, 1999, Hyvärinen et al., 2024, Khemakhem et al., 2020b, Li et al., 2023]. For example, Aapo et al. confirmed identifiability by assuming latent sources belong to the exponential family, with auxiliary variables like domain and time indices [Khemakhem et al., 2020a, Hyvarinen and Morioka, 2016, 2017, Hyvarinen et al., 2019]. In contrast, Zhang et al. showed that nonlinear ICA can achieve

component-wise identifiability without the exponential family assumption [Kong et al., 2022, Xie et al., 2023, Kong et al., 2023, Yan et al., 2024].

Other studies also use sparsity assumptions to achieve identifiability without supervised signals. For instance, Lachapelle et al. applied sparsity regularization to discover latent components [Lachapelle et al., 2023, Lachapelle and Lacoste-Julien, 2022], while Zhang et al. used sparse structures to maintain identifiability under distribution shifts [Zhang et al., 2024a]. Nonlinear ICA has been used for time series identifiability [Hyvarinen and Morioka, 2016, Yan et al., 2024, Huang et al., 2023, Hälvä and Hyvarinen, 2020, Lippe et al., 2022]. Aapo et al. used variance changes to detect nonstationary time series data identifiability, while permutation-based contrastive learning was applied for stationary time series [Hyvarinen and Morioka, 2016]. More recently, techniques like TDRL [Yao et al., 2022], LEAP [Yao et al., 2021] and IDOL [Li et al., 2025] incorporated independent noise and variability features. Additionally, Song et al. identified latent variables without domain-specific observations [Song et al., 2024], and Imant et al. used multimodal comparative learning for modality identifiability [Daunhawer et al., 2023]. Yao et al. showed that multi-perspective causal representations remain identifiable despite incomplete observations [Yao et al., 2023]. However, these methods typically assume invertibility in the mixing process. This paper relaxes that assumption and provides identifiability guarantees for online time series forecasting.

## B  Notations

This section collects the notations used in the theorem proofs for clarity and consistency.

Table A4: List of notations, explanations, and corresponding values.

| Index | Explanation | Support |
|---|---|---|
| $n$ | Number of variables | $n \in \mathbb{N}^+$ |
| $i, j, k, l$ | Index of latent variables | $i, j, k, l \in \{1, \cdots, n\}$ |
| $t$ | Time index | $t \in \mathbb{N}^+$ |
| **Variable** | | |
| $\mathcal{X}_t$ | Support of observed variables in time-index $t$ | $\mathcal{X}_t \subseteq \mathbb{R}^n$ |
| $\mathcal{Z}_t$ | Support of latent variables | $\mathcal{Z}_t \subseteq \mathbb{R}^n$ |
| $\mathbf{x}_t$ | Observed variables in time-index $t$ | $\mathbf{x}_t \in \mathbb{R}^n$ |
| $\mathbf{z}_t$ | Latent variables in time-index $t$ | $\mathbf{z}_t \in \mathbb{R}^n$ |
| $\mathbf{u}_t$ | $\{\mathbf{z}_{t-1}, \mathbf{x}_{t-1}, \mathbf{z}_t, \mathbf{x}_t\}$ | $\mathbf{u}_t \in \mathbb{R}^{4 \times n}$ |
| $\boldsymbol{\epsilon}_t^{\mathbf{o}}$ | Independent noise of mixing procedure | $\boldsymbol{\epsilon}_t^{\mathbf{o}} \sim p_{\epsilon_t^{\mathbf{o}}}$ |
| $\boldsymbol{\epsilon}_{t,i}^{\mathbf{z}}$ | Independent noise of the latent transition of $\mathbf{z}_{t,i}$ | $\boldsymbol{\epsilon}_{t,i}^{\mathbf{z}} \sim p_{\epsilon_{t,i}^{\mathbf{z}}}$ |
| **Function** | | |
| $p_{a\|b}(\cdot \mid b)$ | Density function of $a$ given $b$ | / |
| $\mathbf{g}(\cdot)$ | Nonlinear mixing function | $\mathbb{R}^{2 \times n+1} \to \mathbb{R}^n$ |
| $f_i(\cdot)$ | Transition function of $\mathbf{z}_{t,i}$ | $\mathbb{R}^{n+1} \to \mathbb{R}$ |
| $h(\cdot)$ | Invertible transformation from $\mathbf{z}_t$ to $\hat{\mathbf{z}}_t$ | $\mathbb{R}^n \to \mathbb{R}^n$ |
| $\pi(\cdot)$ | Permutation function | $\mathbb{R}^n \to \mathbb{R}^n$ |
| $\mathcal{F}$ | Function space | / |
| $\mathcal{M}_{\mathbf{u}_t}$ | Markov network over $\mathbf{u}_t$ | / |
| $\phi_l$ | Encoder for $\hat{\mathbf{z}}_t^l$ | / |
| $\psi_l$ | Decoder | / |
| $r_{t,i}^{\mathbf{z}}$ | Noise estimator of $\hat{\epsilon}_{t,i}^{\mathbf{z}}$. | / |
| $r_{t,i}^{\mathbf{o}}$ | Noise estimator of $\hat{\epsilon}_{t,i}^{\mathbf{o}}$. | / |
| **Symbol** | | |
| $\mathcal{R}$ | Bayes Risk | / |
| $\mathbf{J}_\kappa$ | Jacobian matrix of $r_t^l$ | / |

## C    Proof

### C.1    Proof of Theorem 1

**Theorem A1.** *(Predictive-Risk Reduction via Temporal Latent Variables) Let $\mathbf{x}_t, \mathbf{z}_t$, and $\hat{\mathbf{z}}_t$ be the observed variables, ground-truth latent variables, and the estimated latent variables, respectively. We let $\mathbf{x}_{t-\tau:t} = \{\mathbf{x}_{t-\tau}, \cdots, \mathbf{x}_t\}$ be the historical $(\tau + 1)$-step observed variables. Moreover, we let $\mathcal{R}_\mathbf{o}, \mathcal{R}_\mathbf{z}$, and $\mathcal{R}_{\hat{\mathbf{z}}}$ be the expected mean squared error for the models that consider $\{\mathbf{x}_{t-\tau:t}\}, \{\mathbf{x}_{t-\tau:t}, \mathbf{z}_t\}$, and $\{\mathbf{x}_{t-\tau:t}, \hat{\mathbf{z}}_t\}$, respectively. Then, in general, we have $\mathcal{R}_\mathbf{o} \geq \mathcal{R}_{\hat{\mathbf{z}}} \geq \mathcal{R}_\mathbf{z}$, and if $\mathbf{z}_t$ is identifiable we have $\mathcal{R}_\mathbf{o} > \mathcal{R}_{\hat{\mathbf{z}}} = \mathcal{R}_\mathbf{z}$.*

*Proof.* Suppose that the observed variables $\mathbf{x}_t$, ground-truth latent variables $\mathbf{z}_t$, and estimated latent variables $\hat{\mathbf{z}}_t$ follow the data generation process as shown in Figure 1. We let $\mathcal{F}_\mathbf{o} := \sigma(\mathbf{x}_{t-\tau:t}), \mathcal{F}_\mathbf{z} := \sigma(\mathbf{x}_{t-\tau:t}, \mathbf{z}_t)$, and $\mathcal{F}_{\hat{\mathbf{z}}} := \sigma(\mathbf{x}_{t-\tau:t}, \hat{\mathbf{z}})$ be the information $\sigma$-algebras generated by the variables available to the forecaster in the three settings (only observed variables, observed and ground truth latent variables, observed and the estimated latent variables). And the corresponding optimal Bayes forecaster can be formalized as $\hat{\mathbf{x}}_{t+1}^{(\mathbf{o})} := \mathbb{E}[\mathbf{x}_{t+1}|\mathbf{x}_{t-\tau:t}], \hat{\mathbf{x}}_{t+1}^{(\mathbf{z})} := \mathbb{E}[\mathbf{x}_{t+1}|\mathbf{x}_{t-\tau:t}, \mathbf{z}_t]$, and $\hat{\mathbf{x}}_{t+1}^{(\hat{\mathbf{z}})} := \mathbb{E}[\mathbf{x}_{t+1}|\mathbf{x}_{t-\tau:t}, \hat{\mathbf{z}}_t]$, respectively. Then we let $\mathcal{R}_\mathbf{o}, \mathcal{R}_\mathbf{z}$, and $\mathcal{R}_{\hat{\mathbf{z}}}$ be the corresponding Bayes risk. By using the law of total expectation, we have:

$$
\begin{aligned}
\mathcal{R}_\mathbf{o} = \mathbb{E}_{\mathbf{x}_{t+1}, \mathbf{x}_{t-\tau:t}}[(\mathbf{x}_{t+1} - \hat{\mathbf{x}}_{t+1}^{(\mathbf{o})})^2] &= \mathbb{E}_{\mathbf{x}_{t-\tau:t}}[\mathbb{E}_{\mathbf{x}_{t+1}|\mathbf{x}_{t-\tau:t}}[(\mathbf{x}_{t+1} - \hat{\mathbf{x}}_{t+1}^{(\mathbf{o})})^2]] \\
&= \mathbb{E}_{\mathbf{x}_{t-\tau:t}}[\text{Var}(\mathbf{x}_{t+1}|\mathbf{x}_{t-\tau:t})],
\end{aligned}
\tag{A1}
$$

$$
\begin{aligned}
\mathcal{R}_\mathbf{z} = \mathbb{E}_{\mathbf{x}_{t+1}, \mathbf{x}_{t-\tau:t}, \mathbf{z}_t}[(\mathbf{x}_{t+1} - \hat{\mathbf{x}}_{t+1}^{(\mathbf{z})})^2] &= \mathbb{E}_{\mathbf{x}_{t-\tau:t}, \mathbf{z}_t}[\mathbb{E}_{\mathbf{x}_{t+1}|\mathbf{x}_{t-\tau:t}, \mathbf{z}_t}[(\mathbf{x}_{t+1} - \hat{\mathbf{x}}_{t+1}^{(\mathbf{z})})^2]] \\
&= \mathbb{E}_{\mathbf{x}_{t-\tau:t}, \mathbf{z}_t}[\text{Var}(\mathbf{x}_{t+1}|\mathbf{x}_{t-\tau:t}, \mathbf{z}_t)],
\end{aligned}
\tag{A2}
$$

$$
\begin{aligned}
\mathcal{R}_{\hat{\mathbf{z}}} = \mathbb{E}_{\mathbf{x}_{t+1}, \mathbf{x}_{t-\tau:t}, \hat{\mathbf{z}}_t}[(\mathbf{x}_{t+1} - \hat{\mathbf{x}}_{t+1}^{(\hat{\mathbf{z}})})^2] &= \mathbb{E}_{\mathbf{x}_{t-\tau:t}, \hat{\mathbf{z}}_t}[\mathbb{E}_{\mathbf{x}_{t+1}|\mathbf{x}_{t-\tau:t}, \hat{\mathbf{z}}_t}[(\mathbf{x}_{t+1} - \hat{\mathbf{x}}_{t+1}^{(\hat{\mathbf{z}})})^2]] \\
&= \mathbb{E}_{\mathbf{x}_{t-\tau:t}, \hat{\mathbf{z}}_t}[\text{Var}(\mathbf{x}_{t+1}|\mathbf{x}_{t-\tau:t}, \hat{\mathbf{z}}_t)].
\end{aligned}
\tag{A3}
$$

By using the law of total variance, Equation (A1) can be written as:

$$
\begin{aligned}
\underbrace{\mathbb{E}_{\mathbf{x}_{t-\tau:t}}[\text{Var}(\mathbf{x}_{t+1}|\mathbf{x}_{t-\tau:t})]}_{\mathcal{R}_\mathbf{o}} = &\underbrace{\mathbb{E}_{\mathbf{x}_{t-\tau:t}, \mathbf{z}_t}[\text{Var}(\mathbf{x}_{t+1}|\mathbf{x}_{t-\tau:t}, \mathbf{z}_t)]}_{\mathcal{R}_\mathbf{z}} \\
&+ \underbrace{\mathbb{E}_{\mathbf{x}_{t-\tau:t}}[\text{Var}_{\mathbf{z}_t|\mathbf{x}_{t-\tau:t}}(\mathbb{E}[\mathbf{x}_{t+1}|\mathbf{x}_{t-\tau:t}, \mathbf{z}_t])]}_{c},
\end{aligned}
\tag{A4}
$$

and

$$
\begin{aligned}
\underbrace{\mathbb{E}_{\mathbf{x}_{t-\tau:t}}[\text{Var}(\mathbf{x}_{t+1}|\mathbf{x}_{t-\tau:t})]}_{\mathcal{R}_\mathbf{o}} = &\underbrace{\mathbb{E}_{\mathbf{x}_{t-\tau:t}, \hat{\mathbf{z}}_t}[\text{Var}(\mathbf{x}_{t+1}|\mathbf{x}_{t-\tau:t}, \hat{\mathbf{z}}_t)]}_{\mathcal{R}_{\hat{\mathbf{z}}}} \\
&+ \underbrace{\mathbb{E}_{\mathbf{x}_{t-\tau:t}}[\text{Var}_{\hat{\mathbf{z}}_t|\mathbf{x}_{t-\tau:t}}(\mathbb{E}[\mathbf{x}_{t+1}|\mathbf{x}_{t-\tau:t}, \hat{\mathbf{z}}_t])]}_{e}.
\end{aligned}
\tag{A5}
$$

Suppose in Equation (A4), $c = 0$. That implies $\text{Var}_{\mathbf{z}_t|\mathbf{x}_{t-\tau:t}}(\mathbb{E}[\mathbf{x}_{t+1}|\mathbf{x}_{t-\tau:t}, \mathbf{z}_t]) = 0$, meaning that $\mathbf{z}_t$ does not have any influence on the mapping $\mathbf{x}_{t-\tau:t} \to \mathbf{x}_{t+1}$, which is false because $\mathbf{z}_t \not\perp \mathbf{x}_{t+1} | \mathbf{x}_{t-\tau:t}$. This leads to a contradiction, which implies $c > 0$ and hence $\mathcal{R}_\mathbf{o} > \mathcal{R}_\mathbf{z}$.

Consider Equation (A5). Here, in general, for any $\hat{\mathbf{z}}_t$ we have $e \geq 0$ and hence $\mathcal{R}_\mathbf{o} \geq \mathcal{R}_{\hat{\mathbf{z}}}$.

Then we leverage the law of total variance again and have:

$$
\begin{aligned}
&\text{Var}_{\mathbf{z}_t|\mathbf{x}_{t-\tau:t}}(\mathbb{E}(\mathbf{x}_{t+1}|\mathbf{x}_{t-\tau:t}, \mathbf{z}_t)) \\
=&\mathbb{E}_{\hat{\mathbf{z}}_t|\mathbf{x}_{t-\tau:t}}[\text{Var}_{\mathbf{z}_t|\mathbf{x}_{t-\tau:t}, \hat{\mathbf{z}}_t}(\mathbb{E}[\mathbf{x}_{t+1}|\mathbf{x}_{t-\tau:t}, \mathbf{z}_t])] + \text{Var}_{\hat{\mathbf{z}}_t|\mathbf{x}_{t-\tau:t}}(\mathbb{E}_{\mathbf{z}_t|\mathbf{x}_{t-\tau:t}, \hat{\mathbf{z}}_t}\mathbb{E}[\mathbf{x}_{t+1}|\mathbf{x}_{t-\tau:t}, \mathbf{z}_t]) \\
=&\mathbb{E}_{\hat{\mathbf{z}}_t|\mathbf{x}_{t-\tau:t}}[\text{Var}_{\mathbf{z}_t|\mathbf{x}_{t-\tau:t}, \hat{\mathbf{z}}_t}(\mathbb{E}[\mathbf{x}_{t+1}|\mathbf{x}_{t-\tau:t}, \mathbf{z}_t])] + \text{Var}_{\hat{\mathbf{z}}_t|\mathbf{x}_{t-\tau:t}}(\mathbb{E}[\mathbf{x}_{t+1}|\mathbf{x}_{t-\tau:t}, \hat{\mathbf{z}}_t])
\end{aligned}
\tag{A6}
$$

Then we take the expectation on both sides of Equation (A6) and have :

$$
\begin{aligned}
\underbrace{\mathbb{E}_{\mathbf{x}_{t-\tau:t}}[\text{Var}_{\mathbf{z}_t|\mathbf{x}_{t-\tau:t}}(\mathbb{E}[\mathbf{x}_{t+1}|\mathbf{x}_{t-\tau:t}, \mathbf{z}_t])]}_{c} = &\underbrace{\mathbb{E}_{\mathbf{x}_{t-\tau:t}}[\text{Var}_{\hat{\mathbf{z}}_t|\mathbf{x}_{t-\tau:t}}(\mathbb{E}[\mathbf{x}_{t+1}|\mathbf{x}_{t-\tau:t}, \hat{\mathbf{z}}_t])]}_{e} \\
&+ \mathbb{E}_{\hat{\mathbf{z}}_t, \mathbf{x}_{t-\tau:t}}[\text{Var}_{\mathbf{z}_t|\mathbf{x}_{t-\tau:t}, \hat{\mathbf{z}}_t}(\mathbb{E}[\mathbf{x}_{t+1}|\mathbf{x}_{t-\tau:t}, \mathbf{z}_t])]
\end{aligned}
\tag{A7}
$$

Similarly, $\mathbb{E}_{\hat{\mathbf{z}}_t, \mathbf{x}_{t-\tau:t}}[\mathrm{Var}_{\mathbf{z}_t|\mathbf{x}_{t-\tau:t}, \hat{\mathbf{z}}_t}(\mathbb{E}[\mathbf{x}_{t+1}|\mathbf{x}_{t-\tau:t}, \mathbf{z}_t])]$ $=$ $0$ means that $\mathrm{Var}_{\mathbf{z}_t|\mathbf{x}_{t-\tau:t}, \hat{\mathbf{z}}_t}(\mathbb{E}[\mathbf{x}_{t+1}|\mathbf{x}_{t-\tau:t}, \mathbf{z}_t])$ $=$ $0$, implying that $\mathbb{E}[\mathbf{x}_{t+1}|\mathbf{x}_{t-\tau:t}, \mathbf{z}_t]$ is a constant, i.e., $\mathbf{z}_t$ and $\hat{\mathbf{z}}_t$ have a one-to-one correspondence. Therefore, in general, $c \geq e$, and $c = e$ iff $\mathbf{z}_t$ is identifiable.

By combining Equation (A4) and (A5), in general, we have $\mathcal{R}_\mathbf{o} \geq \mathcal{R}_{\hat{\mathbf{z}}} \geq \mathcal{R}_\mathbf{z}$, and if $\mathbf{z}_t$ is identifiable we have $\mathcal{R}_\mathbf{o} > \mathcal{R}_{\hat{\mathbf{z}}} = \mathcal{R}_\mathbf{z}$.

$\square$

## C.2 Proof of Theorem 2.

For a better understanding of our proof, we begin by introducing an additional operator to represent the point-wise distributional transformation. For generality, we denote two variables as $\mathbf{a}$ and $\mathbf{b}$, with corresponding support sets $\mathcal{A}$ and $\mathcal{B}$.

**Definition 5.** *(Diagonal Operator) Consider two random variable $a$ and $b$, density functions $p_a$ and $p_b$ are defined on some support $\mathcal{A}$ and $\mathcal{B}$, respectively. The diagonal operator $D_{b|a}$ maps the density function $p_a$ to another density function $D_{b|a} \circ p_a$ defined by the pointwise multiplication of the function $p_{b|a}$ at a fixed point $b$:*

$$p_{b|a}(b \mid \cdot)p_a = D_{b|a} \circ p_a, where\ D_{b|a} = p_{b|a}(b \mid \cdot). \tag{A8}$$

**Theorem A2.** *(Block-wise Identification under 4 Adjacent Observed Variables.) Suppose that the observed and latent variables follow the data generation process. By matching the true joint distribution of 4 adjacent observed variables, i.e., $\{\mathbf{x}_{t-2}, \mathbf{x}_{t-1}, \mathbf{x}_t, \mathbf{x}_{t+1}\}$, we further consider the following assumptions:*

- *A1 (Bound and Continuous Density): The joint distribution of $\mathbf{x}, \mathbf{z}$ and their marginal and conditional densities are bounded and continuous.*

- *A2 (Injectivity): There exists observed variables $\mathbf{x}_t$ such that for any $\mathbf{x}_t \in \mathcal{X}_t$, there exist a $\mathbf{x}_{t-1} \in \mathcal{X}_{t-1}$ and a neighborhood $^3$ $\mathcal{N}^r$ around $(\mathbf{x}_t, \mathbf{x}_{t-1})$ such that, for any $(\bar{\mathbf{x}}_t, \bar{\mathbf{x}}_{t-1}) \in \mathcal{N}^r$, $L_{\bar{\mathbf{x}}_t, \mathbf{x}_{t+1}|\mathbf{x}_{t-2}, \bar{\mathbf{x}}_{t-1}}$ is injective; $L_{\mathbf{x}_{t+1}|\mathbf{x}_t, \mathbf{z}_t}, L_{\mathbf{x}_t|\mathbf{x}_{t-2}, \mathbf{x}_{t-1}}$ is injective for any $\mathbf{x}_t \in \mathcal{X}_t$ and $\mathbf{x}_{t-1} \in \mathcal{X}_{t-1}$, respectively.*

- *A3 (Uniqueness of Spectral Decomposition) For any $\mathbf{x}_t \in \mathcal{X}_t$ and any $\bar{\mathbf{z}}_t \neq \tilde{\mathbf{z}}_t \in \mathcal{Z}_t$, there exists a $\mathbf{x}_{t-1} \in \mathcal{X}_{t-1}$ and corresponding neighborhood $\mathcal{N}^r$ satisfying Assumption A2 such that, for some $(\bar{\mathbf{x}}_t, \bar{\mathbf{x}}_{t-1}) \in \mathcal{N}^r$ with $\bar{\mathbf{x}}_t \neq \mathbf{x}_t, \bar{\mathbf{x}}_{t-1} \neq \mathbf{x}_{t-1}$:*

    *i $k(\mathbf{x}_t, \bar{\mathbf{x}}_t, \mathbf{x}_{t-1}, \bar{\mathbf{x}}_{t-1}, \bar{\mathbf{z}}_t) < C < \infty$ for any $\mathbf{z}_t \in \mathcal{Z}_t$ and some constant $C$.*

    *ii $k(\mathbf{x}_t, \bar{\mathbf{x}}_t, \mathbf{x}_{t-1}, \bar{\mathbf{x}}_{t-1}, \bar{\mathbf{z}}_t) \neq k(\mathbf{x}_t, \bar{\mathbf{x}}_t, \mathbf{x}_{t-1}, \bar{\mathbf{x}}_{t-1}, \tilde{\mathbf{z}}_t)$, where*

$$k(\mathbf{x}_t, \bar{\mathbf{x}}_t, \mathbf{x}_{t-1}, \bar{\mathbf{x}}_{t-1}, \mathbf{z}_t) = \frac{p_{\mathbf{x}_t|\mathbf{x}_{t-1}, \mathbf{z}_t}(\mathbf{x}_t \mid \mathbf{x}_{t-1}, \mathbf{z}_t)p_{\mathbf{x}_t|\mathbf{x}_{t-1}, \mathbf{z}_t}(\bar{\mathbf{x}}_t \mid \bar{\mathbf{x}}_{t-1}, \mathbf{z}_t)}{p_{\mathbf{x}_t|\mathbf{x}_{t-1}, \mathbf{z}_t}(\bar{\mathbf{x}}_t \mid \mathbf{x}_{t-1}, \mathbf{z}_t)p_{\mathbf{x}_t|\mathbf{x}_{t-1}, \mathbf{z}_t}(\mathbf{x}_t \mid \bar{\mathbf{x}}_{t-1}, \mathbf{z}_t)}. \tag{A9}$$

*Suppose that we have learned $(\hat{\mathbf{g}}, \hat{\mathbf{f}}, p_{\hat{\epsilon}})$ to achieve Equations (1), then the combination of Markov state $\mathbf{z}_t, \mathbf{x}_t$ is identifiable, i.e., $[\hat{\mathbf{z}}_t, \hat{\mathbf{x}}_t] = H(\mathbf{z}_t, \mathbf{x}_t)$, where $H$ is invertible and differentiable.*

---

$^3$Please refer to Appendix C.5 for the definition of neighborhood.

*Proof.* By the definition of data generation process, the observed density $p_{\mathbf{x}_{t+1},\mathbf{x}_t,\mathbf{x}_{t-1},\mathbf{x}_{t-2}}$ equals

$$p_{\mathbf{x}_{t+1},\mathbf{x}_t,\mathbf{x}_{t-1},\mathbf{x}_{t-2}}$$

$$= \iint p_{\mathbf{x}_{t+1},\mathbf{x}_t,\mathbf{z}_t,\mathbf{z}_{t-1},\mathbf{x}_{t-1},\mathbf{x}_{t-2}} \, d\mathbf{z}_t d\mathbf{z}_{t-1}$$

$$= \iint p_{\mathbf{x}_{t+1}|\mathbf{x}_t,\mathbf{x}_{t-1},\mathbf{x}_{t-2},\mathbf{z}_t,\mathbf{z}_{t-1}} p_{\mathbf{x}_t,\mathbf{z}_t|\mathbf{x}_{t-1},\mathbf{x}_{t-2},\mathbf{z}_{t-1}} p_{\mathbf{z}_{t-1},\mathbf{x}_{t-1},\mathbf{x}_{t-2}} \, d\mathbf{z}_t d\mathbf{z}_{t-1}$$

$$= \iint p_{\mathbf{x}_{t+1}|\mathbf{x}_t,\mathbf{z}_t} p_{\mathbf{x}_t,\mathbf{z}_t|\mathbf{x}_{t-1},\mathbf{z}_{t-1}} p_{\mathbf{z}_{t-1},\mathbf{x}_{t-1},\mathbf{x}_{t-2}} \, d\mathbf{z}_t d\mathbf{z}_{t-1}$$

$$= \iint p_{\mathbf{x}_{t+1}|\mathbf{x}_t,\mathbf{z}_t} p_{\mathbf{x}_t|\mathbf{x}_{t-1},\mathbf{z}_t,\mathbf{z}_{t-1}} p_{\mathbf{z}_t|\mathbf{x}_{t-1},\mathbf{x}_{t-2},\mathbf{z}_{t-1}} p_{\mathbf{x}_{t-1},\mathbf{x}_{t-2},\mathbf{z}_{t-1}} \, d\mathbf{z}_t d\mathbf{z}_{t-1}.$$

$$= \iint p_{\mathbf{x}_{t+1}|\mathbf{x}_t,\mathbf{z}_t} p_{\mathbf{x}_t|\mathbf{x}_{t-1},\mathbf{z}_t,\mathbf{z}_{t-1}} p_{\mathbf{z}_t,\mathbf{x}_{t-1},\mathbf{x}_{t-2},\mathbf{z}_{t-1}} \, d\mathbf{z}_t d\mathbf{z}_{t-1}.$$

According to the property of Markov process,

$$p_{\mathbf{x}_{t+1},\mathbf{x}_t,\mathbf{x}_{t-1},\mathbf{x}_{t-2}} = \int p_{\mathbf{x}_{t+1}|\mathbf{x}_t,\mathbf{z}_t} p_{\mathbf{x}_t|\mathbf{x}_{t-1},\mathbf{z}_t} \left( \int p_{\mathbf{z}_t,\mathbf{z}_{t-1},\mathbf{x}_{t-1},\mathbf{x}_{t-2}} \, d\mathbf{z}_{t-1} \right) d\mathbf{z}_t$$

$$= \int p_{\mathbf{x}_{t+1}|\mathbf{x}_t,\mathbf{z}_t} p_{\mathbf{x}_t|\mathbf{x}_{t-1},\mathbf{z}_t} p_{\mathbf{z}_t,\mathbf{x}_{t-1},\mathbf{x}_{t-2}} \, d\mathbf{z}_t. \qquad (A10)$$

In operator notation, given values of $(\mathbf{x}_t, \mathbf{x}_{t-1}) \in \mathcal{X}_t \times \mathcal{X}_{t-1}$, this is

$$L_{\mathbf{x}_{t+1},\mathbf{x}_t,\mathbf{x}_{t-1},\mathbf{x}_{t-2}} = L_{\mathbf{x}_{t+1}|\mathbf{x}_t,\mathbf{z}_t} D_{\mathbf{x}_t|\mathbf{x}_{t-1},\mathbf{z}_t} L_{\mathbf{z}_t,\mathbf{x}_{t-1},\mathbf{x}_{t-2}}. \qquad (A11)$$

After obtaining the representation of observed density function, furthermore, the structure of Markov process implies the following two equalities:

$$p_{\mathbf{x}_{t+1},\mathbf{x}_t,\mathbf{x}_{t-1},\mathbf{x}_{t-2}} = \int p_{\mathbf{x}_{t+1}|\mathbf{x}_t,\mathbf{z}_t} p_{\mathbf{x}_t,\mathbf{z}_t,\mathbf{x}_{t-1},\mathbf{x}_{t-2}} \, d\mathbf{z}_t,$$

$$p_{\mathbf{x}_t,\mathbf{z}_t,\mathbf{x}_{t-1},\mathbf{x}_{t-2}} = \int p_{\mathbf{x}_t,\mathbf{z}_t|\mathbf{x}_{t-1},\mathbf{z}_{t-1}} p_{\mathbf{z}_{t-1},\mathbf{x}_{t-1},\mathbf{x}_{t-2}} \, d\mathbf{z}_{t-1}. \qquad (A12)$$

In operator notation, for fixed $\mathbf{x}_t, \mathbf{x}_{t-1}$, the above equations are expressed:

$$L_{\mathbf{x}_{t+1},\mathbf{x}_t,\mathbf{x}_{t-1},\mathbf{x}_{t-2}} = L_{\mathbf{x}_{t+1}|\mathbf{x}_t,\mathbf{z}_t} L_{\mathbf{x}_t,\mathbf{z}_t,\mathbf{x}_{t-1},\mathbf{x}_{t-2}},$$

$$L_{\mathbf{x}_t,\mathbf{z}_t,\mathbf{x}_{t-1},\mathbf{x}_{t-2}} = L_{\mathbf{x}_t,\mathbf{z}_t|\mathbf{x}_{t-1},\mathbf{z}_{t-1}} L_{\mathbf{z}_{t-1},\mathbf{x}_{t-1},\mathbf{x}_{t-2}}. \qquad (A13)$$

Substituting the second line into the first, we get

$$L_{\mathbf{x}_{t+1},\mathbf{x}_t,\mathbf{x}_{t-1},\mathbf{x}_{t-2}} = L_{\mathbf{x}_{t+1}|\mathbf{x}_t,\mathbf{z}_t} L_{\mathbf{x}_t,\mathbf{z}_t|\mathbf{x}_{t-1},\mathbf{z}_{t-1}} L_{\mathbf{z}_{t-1},\mathbf{x}_{t-1},\mathbf{x}_{t-2}}$$

$$\Leftrightarrow L_{\mathbf{x}_t,\mathbf{z}_t|\mathbf{x}_{t-1},\mathbf{z}_{t-1}} L_{\mathbf{z}_{t-1},\mathbf{x}_{t-1},\mathbf{x}_{t-2}} = L^{-1}_{\mathbf{x}_{t+1}|\mathbf{x}_t,\mathbf{z}_t} L_{\mathbf{x}_{t+1},\mathbf{x}_t,\mathbf{x}_{t-1},\mathbf{x}_{t-2}}. \qquad (A14)$$

The second line uses Assumption A2. Next, we eliminate $L_{\mathbf{z}_{t-1},\mathbf{x}_{t-1},\mathbf{x}_{t-2}}$ from the above. Again, using the conditional independence of Markov process, we have:

$$p_{\mathbf{x}_t,\mathbf{x}_{t-1},\mathbf{x}_{t-2}} = \int p_{\mathbf{x}_t|\mathbf{x}_{t-1},\mathbf{z}_{t-1}} p_{\mathbf{z}_{t-1},\mathbf{x}_{t-1},\mathbf{x}_{t-2}} \, d\mathbf{z}_{t-1}, \qquad (A15)$$

which can be represented in terms of operator (for fixed $\mathbf{x}_{t-1}$) as:

$$L_{\mathbf{x}_t,\mathbf{x}_{t-1},\mathbf{x}_{t-2}} = L_{\mathbf{x}_t|\mathbf{x}_{t-1},\mathbf{z}_{t-1}} L_{\mathbf{z}_{t-1},\mathbf{x}_{t-1},\mathbf{x}_{t-2}},$$

$$\Rightarrow \qquad L_{\mathbf{z}_{t-1},\mathbf{x}_{t-1},\mathbf{x}_{t-2}} = L^{-1}_{\mathbf{x}_t|\mathbf{x}_{t-1},\mathbf{z}_{t-1}} L_{\mathbf{x}_t,\mathbf{x}_{t-1},\mathbf{x}_{t-2}}. \qquad (A16)$$

The R.H.S. applies Assumption A2. Hence, substituting the above into Eq. A14, we obtain the desired representation:

$$L_{\mathbf{x}_t,\mathbf{z}_t|\mathbf{x}_{t-1},\mathbf{z}_{t-1}} L^{-1}_{\mathbf{x}_t|\mathbf{x}_{t-1},\mathbf{z}_{t-1}} L_{\mathbf{x}_t,\mathbf{x}_{t-1},\mathbf{x}_{t-2}} = L^{-1}_{\mathbf{x}_{t+1}|\mathbf{x}_t,\mathbf{z}_t} L_{\mathbf{x}_{t+1},\mathbf{x}_t,\mathbf{x}_{t-1},\mathbf{x}_{t-2}}$$

$$\Rightarrow \qquad L_{\mathbf{x}_t,\mathbf{z}_t|\mathbf{x}_{t-1},\mathbf{z}_{t-1}} = L^{-1}_{\mathbf{x}_{t+1}|\mathbf{x}_t,\mathbf{z}_t} L_{\mathbf{x}_{t+1},\mathbf{x}_t,\mathbf{x}_{t-1},\mathbf{x}_{t-2}} L^{-1}_{\mathbf{x}_t,\mathbf{x}_{t-1},\mathbf{x}_{t-2}} L_{\mathbf{x}_t,\mathbf{x}_{t-1},\mathbf{z}_{t-1}}. \qquad (A17)$$

The second line applies Assumption A2 to post-multiply by $L^{-1}_{\mathbf{x}_t,\mathbf{x}_{t-1},\mathbf{x}_{t-2}}$, while in the third line, we postmultiply both sides by $L_{\mathbf{x}_t|\mathbf{x}_{t-1},\mathbf{z}_{t-1}}$.

For each $\mathbf{x}_t$, choose a $\mathbf{x}_{t-1}$ and a neighborhood $\mathcal{N}^r$ around $(\mathbf{x}_t,\mathbf{x}_{t-1})$ to satisfy Assumption A2 and A2, and pick a $(\bar{\mathbf{x}}_t,\bar{\mathbf{x}}_{t-1})$ within the neighborhood $\mathcal{N}^r$ to satisfy Assumption A2. Because $(\bar{\mathbf{x}}_t,\bar{\mathbf{x}}_{t-1}) \in \mathcal{N}^r$, also $(\mathbf{x}_t,\bar{\mathbf{x}}_{t-1})$, $(\bar{\mathbf{x}}_t,\mathbf{x}_{t-1}) \in \mathcal{N}^r$. By the representation of observations in Eq. A11, we have

$$L_{\mathbf{x}_{t+1},\mathbf{x}_t,\mathbf{x}_{t-1},\mathbf{x}_{t-2}} = L_{\mathbf{x}_{t+1}|\mathbf{x}_t,\mathbf{z}_t} D_{\mathbf{x}_t|\mathbf{x}_{t-1},\mathbf{z}_t} L_{\mathbf{z}_t,\mathbf{x}_{t-1},\mathbf{x}_{t-2}}.$$

The first term on the R.H.S., $L_{\mathbf{x}_{t+1}|\mathbf{x}_t,\mathbf{z}_t}$, does not depend on $\mathbf{x}_{t-1}$, and the last term $L_{\mathbf{z}_t,\mathbf{x}_{t-1},\mathbf{x}_{t-2}}$ does not depend on $\mathbf{x}_t$. This feature suggests that, by evaluating Eq. A1 at the four pairs of points $(\mathbf{x}_t,\mathbf{x}_{t-1})$, $(\bar{\mathbf{x}}_t,\mathbf{x}_{t-1})$, $(\mathbf{x}_t,\bar{\mathbf{x}}_{t-1})$, $(\bar{\mathbf{x}}_t,\bar{\mathbf{x}}_{t-1})$, each pair of equations will share one operator in common. Specifically:

$$L_{\mathbf{x}_{t+1},\mathbf{x}_t,\mathbf{x}_{t-1},\mathbf{x}_{t-2}} = L_{\mathbf{x}_{t+1}|\mathbf{x}_t,\mathbf{z}_t} D_{\mathbf{x}_t|\mathbf{x}_{t-1},\mathbf{z}_t} L_{\mathbf{z}_t,\mathbf{x}_{t-1},\mathbf{x}_{t-2}}, \tag{A18}$$

$$L_{\mathbf{x}_{t+1},\bar{\mathbf{x}}_t,\mathbf{x}_{t-1},\mathbf{x}_{t-2}} = L_{\mathbf{x}_{t+1}|\bar{\mathbf{x}}_t,\mathbf{z}_t} D_{\bar{\mathbf{x}}_t|\mathbf{x}_{t-1},\mathbf{z}_t} L_{\mathbf{z}_t,\mathbf{x}_{t-1},\mathbf{x}_{t-2}}, \tag{A19}$$

$$L_{\mathbf{x}_{t+1},\mathbf{x}_t,\bar{\mathbf{x}}_{t-1},\mathbf{x}_{t-2}} = L_{\mathbf{x}_{t+1}|\mathbf{x}_t,\mathbf{z}_t} D_{\mathbf{x}_t|\bar{\mathbf{x}}_{t-1},\mathbf{z}_t} L_{\mathbf{z}_t,\bar{\mathbf{x}}_{t-1},\mathbf{x}_{t-2}}, \tag{A20}$$

$$L_{\mathbf{x}_{t+1},\bar{\mathbf{x}}_t,\bar{\mathbf{x}}_{t-1},\mathbf{x}_{t-2}} = L_{\mathbf{x}_{t+1}|\bar{\mathbf{x}}_t,\mathbf{z}_t} D_{\bar{\mathbf{x}}_t|\bar{\mathbf{x}}_{t-1},\mathbf{z}_t} L_{\mathbf{z}_t,\bar{\mathbf{x}}_{t-1},\mathbf{x}_{t-2}}. \tag{A21}$$

Assumption A2 implies that $L_{\mathbf{x}_{t+1}|\bar{\mathbf{x}}_t,\mathbf{z}_t}$ is invertible. Moreover, Assumption A2 implies $p_{\mathbf{x}_t|\mathbf{x}_{t-1},\mathbf{z}_t}(\mathbf{x}_t \mid \mathbf{x}_{t-1},\mathbf{z}_t) > 0$ for all $\mathbf{z}_t$, so that $D_{\bar{\mathbf{x}}_t|\mathbf{x}_{t-1},\mathbf{z}_t}$ is invertible. We can then solve for $L_{\mathbf{z}_t,\mathbf{x}_{t-1},\mathbf{x}_{t-2}}$ from Eq. A19 as

$$D^{-1}_{\bar{\mathbf{x}}_t|\mathbf{x}_{t-1},\mathbf{z}_t} L^{-1}_{\mathbf{x}_{t+1}|\bar{\mathbf{x}}_t,\mathbf{z}_t} L_{\mathbf{x}_{t+1},\bar{\mathbf{x}}_t,\mathbf{x}_{t-1},\mathbf{x}_{t-2}} = L_{\mathbf{z}_t,\mathbf{x}_{t-1},\mathbf{x}_{t-2}}. \tag{A22}$$

Plugging this expression into Eq. A18 leads to

$$L_{\mathbf{x}_{t+1},\mathbf{x}_t,\mathbf{x}_{t-1},\mathbf{x}_{t-2}} = L_{\mathbf{x}_{t+1}|\mathbf{x}_t,\mathbf{z}_t} D_{\mathbf{x}_t|\mathbf{x}_{t-1},\mathbf{z}_t} D^{-1}_{\bar{\mathbf{x}}_t|\mathbf{x}_{t-1},\mathbf{z}_t} L^{-1}_{\mathbf{x}_{t+1}|\bar{\mathbf{x}}_t,\mathbf{z}_t} L_{\mathbf{x}_{t+1},\bar{\mathbf{x}}_t,\mathbf{x}_{t-1},\mathbf{x}_{t-2}}. \tag{A23}$$

Lemma 1 of Hu and Schennach [2008] shows that, given the injectivity of $L_{\mathbf{x}_{t-2},\bar{\mathbf{x}}_{t-1},\mathbf{x}_t,\mathbf{x}_{t+1}}$ as in Assumption A2, we can postmultiply by $L^{-1}_{\mathbf{x}_{t+1},\mathbf{x}_t,\mathbf{x}_{t-1},\mathbf{x}_{t-2}}$ to obtain:

$$\mathbf{A} \equiv L_{\mathbf{x}_{t+1},\mathbf{x}_t,\mathbf{x}_{t-1},\mathbf{x}_{t-2}} L^{-1}_{\mathbf{x}_{t+1},\mathbf{x}_t,\mathbf{x}_{t-1},\mathbf{x}_{t-2}} = L_{\mathbf{x}_{t+1}|\mathbf{x}_t,\mathbf{z}_t} D_{\mathbf{x}_t|\mathbf{x}_{t-1},\mathbf{z}_t} D^{-1}_{\bar{\mathbf{x}}_t|\mathbf{x}_{t-1},\mathbf{z}_t} L^{-1}_{\mathbf{x}_{t+1}|\bar{\mathbf{x}}_t,\mathbf{z}_t}. \tag{A24}$$

Similarly, manipulations of Eq. A20 and Eq. A21 lead to

$$\mathbf{B} \equiv L_{\mathbf{x}_{t+1},\bar{\mathbf{x}}_t,\mathbf{x}_{t-1},\mathbf{x}_{t-2}} L^{-1}_{\mathbf{x}_{t+1},\mathbf{x}_t,\bar{\mathbf{x}}_{t-1},\mathbf{x}_{t-2}} = L_{\mathbf{x}_{t+1}|\bar{\mathbf{x}}_t,\mathbf{z}_t} D_{\bar{\mathbf{x}}_t|\bar{\mathbf{x}}_{t-1},\mathbf{z}_t} D^{-1}_{\mathbf{x}_t|\bar{\mathbf{x}}_{t-1},\mathbf{z}_t} L^{-1}_{\mathbf{x}_{t+1}|\mathbf{x}_t,\mathbf{z}_t}. \tag{A25}$$

Assumption A2 guarantees that, for any $\mathbf{x}_t$, $(\bar{\mathbf{x}}_t,\mathbf{x}_{t-1},\bar{\mathbf{x}}_{t-1})$ exist so that Eq. A24 and Eq. A25 are valid operations. Finally, we postmultiply Eq. A24 by Eq. A25 to obtain:

$$\mathbf{AB} = L_{\mathbf{x}_{t+1}|\mathbf{x}_t,\mathbf{z}_t} D_{\mathbf{x}_t|\mathbf{x}_{t-1},\mathbf{z}_t} D^{-1}_{\bar{\mathbf{x}}_t|\mathbf{x}_{t-1},\mathbf{z}_t} \left( L_{\mathbf{x}_{t+1}|\bar{\mathbf{x}}_t,\mathbf{z}_t} L_{\mathbf{x}_{t+1}|\bar{\mathbf{x}}_t,\mathbf{z}_t} \right) \times D_{\bar{\mathbf{x}}_t|\bar{\mathbf{x}}_{t-1},\mathbf{z}_t} D^{-1}_{\mathbf{x}_t|\bar{\mathbf{x}}_{t-1},\mathbf{z}_t} L^{-1}_{\mathbf{x}_{t+1}|\mathbf{x}_t,\mathbf{z}_t}$$

$$= L_{\mathbf{x}_{t+1}|\mathbf{x}_t,\mathbf{z}_t} \left( D_{\mathbf{x}_t|\mathbf{x}_{t-1},\mathbf{z}_t} D^{-1}_{\bar{\mathbf{x}}_t|\mathbf{x}_{t-1},\mathbf{z}_t} D_{\bar{\mathbf{x}}_t|\bar{\mathbf{x}}_{t-1},\mathbf{z}_t} D^{-1}_{\mathbf{x}_t|\bar{\mathbf{x}}_{t-1},\mathbf{z}_t} \right) L^{-1}_{\mathbf{x}_{t+1}|\mathbf{x}_t,\mathbf{z}_t}$$

$$\equiv L_{\mathbf{x}_{t+1}|\mathbf{x}_t,\mathbf{z}_t} D_{\mathbf{x}_t,\bar{\mathbf{x}}_t,\mathbf{x}_{t-1},\bar{\mathbf{x}}_{t-1},\mathbf{z}_t} L^{-1}_{\mathbf{x}_{t+1}|\mathbf{x}_t,\mathbf{z}_t}, \tag{A26}$$

where

$$\left( D_{\mathbf{x}_t,\bar{\mathbf{x}}_t,\mathbf{x}_{t-1},\bar{\mathbf{x}}_{t-1},\mathbf{z}_t} h \right)(\mathbf{z}_t) = \left( D_{\mathbf{x}_t|\mathbf{x}_{t-1},\mathbf{z}_t} D^{-1}_{\bar{\mathbf{x}}_t|\mathbf{x}_{t-1},\mathbf{z}_t} D_{\bar{\mathbf{x}}_t|\bar{\mathbf{x}}_{t-1},\mathbf{z}_t} D^{-1}_{\mathbf{x}_t|\bar{\mathbf{x}}_{t-1},\mathbf{z}_t} h \right)(\mathbf{z}_t)$$

$$= \frac{p_{\mathbf{x}_t|\mathbf{x}_{t-1},\mathbf{z}_t}(\mathbf{x}_t \mid \mathbf{x}_{t-1},\mathbf{z}_t) p_{\mathbf{x}_t|\mathbf{x}_{t-1},\mathbf{z}_t}(\bar{\mathbf{x}}_t \mid \bar{\mathbf{x}}_{t-1},\mathbf{z}_t)}{p_{\mathbf{x}_t|\mathbf{x}_{t-1},\mathbf{z}_t}(\bar{\mathbf{x}}_t \mid \mathbf{x}_{t-1},\mathbf{z}_t) p_{\mathbf{x}_t|\mathbf{x}_{t-1},\mathbf{z}_t}(\mathbf{x}_t \mid \bar{\mathbf{x}}_{t-1},\mathbf{z}_t)} h(\mathbf{z}_t)$$

$$\equiv k(\mathbf{x}_t,\bar{\mathbf{x}}_t,\mathbf{x}_{t-1},\bar{\mathbf{x}}_{t-1},\mathbf{z}_t) h(\mathbf{z}_t). \tag{A27}$$

By matching the marginal distribution of observed variables, we can define the operator $\hat{\mathbf{A}}\hat{\mathbf{B}}$ as the estimated counterpart of the operator $\mathbf{AB}$, constructed using the estimated densities of $\hat{\mathbf{x}}_{t-2}$, $\hat{\mathbf{x}}_{t-1}$, $\hat{\mathbf{x}}_t$, $\hat{\mathbf{x}}_{t+1}$, and $\hat{\mathbf{z}}_t$. Since both the marginal and conditional distributions of the observed variables

are matched, the true model and the estimated model yield the same distribution over the observed variables. Therefore, we also have:

$$\hat{\mathbf{A}}\hat{\mathbf{B}} = \mathbf{A}\mathbf{B}. \tag{A28}$$

Eq. A26 implies that the observed operator $\mathbf{AB}$ has an inherent eigenvalue–eigenfunction decomposition, with the eigenvalues corresponding to the function $k(\mathbf{x}_t, \bar{\mathbf{x}}_t, \mathbf{x}_{t-1}, \bar{\mathbf{x}}_{t-1}, \mathbf{z}_t)$ and the eigenfunctions corresponding to the density $p_{\mathbf{x}_{t+1}|\mathbf{x}_t,\mathbf{z}_t}(\cdot \mid \mathbf{x}_t, \mathbf{z}_t)$. Furthermore, Eq. A28 implies that $\mathbf{AB}$ and $\hat{\mathbf{A}}\hat{\mathbf{B}}$ admit the same eigendecompositions, which are similar to the decomposition in Hu and Schennach [2008] or Carroll et al. [2010]. Assumption A2 ensures that this decomposition is unique. Specifically, the operator $\mathbf{AB}$ on the L.H.S. has the same spectrum as the diagonal operator $D_{\mathbf{x}_t,\bar{\mathbf{x}}_t,\mathbf{x}_{t-1},\bar{\mathbf{x}}_{t-1},\mathbf{z}_t}$. Assumption A2 guarantees that the spectrum of the diagonal operator is bounded. Since an operator is bounded by the largest element of its spectrum, Assumption A2 also implies that the operator $\mathbf{AB}$ is bounded, whence we can apply Theorem XV.4.3.5 from Dunford and Schwartz [1971] to show the uniqueness of the spectral decomposition of bounded linear operators:

$$L_{\mathbf{x}_{t+1}|\mathbf{x}_t,\mathbf{z}_t} = C L_{\hat{\mathbf{x}}_{t+1}|\hat{\mathbf{x}}_t,\hat{\mathbf{z}}_t} P^{-1}. \quad D_{\mathbf{x}_t,\bar{\mathbf{x}}_t,\mathbf{x}_{t-1},\bar{\mathbf{x}}_{t-1},\mathbf{z}_t} = P D_{\hat{\mathbf{x}}_t,\hat{\bar{\mathbf{x}}}_t,\hat{\mathbf{x}}_{t-1},\hat{\bar{\mathbf{x}}}_{t-1},\hat{\mathbf{z}}_t} P^{-1} \tag{A29}$$

where $C$ is a scalar accounting for scaling indeterminacy and $P$ is a permutation on the order of elements in $L_{\hat{\mathbf{x}}_{t+1}|\hat{\mathbf{x}}_t,\hat{\mathbf{z}}_t}$, as discussed in [Dunford and Schwartz, 1971]. These forms of indeterminacy are analogous to those in eigendecomposition, which can be viewed as a finite-dimensional special case. We will show how to resolve the indeterminacies in eigen(spectral) decomposition.

First, Eq. A29 itself does not imply that the eigenvalues $k(\mathbf{x}_t, \bar{\mathbf{x}}_t, \mathbf{x}_{t-1}, \bar{\mathbf{x}}_{t-1}, \mathbf{z}_t)$ are distinct for different values $\mathbf{z}_t$. When the eigenvalues are the same for multiple values of $\mathbf{z}_t$, the corresponding eigenfunctions are only determined up to an arbitrary linear combination, implying that they are not identified. Assumption A2 rules out this possibility, and implies that for each $\mathbf{x}_t$, we can find values $\bar{\mathbf{x}}_t, \mathbf{x}_{t-1}, \bar{\mathbf{x}}_{t-1}$ such that the eigenvalues are distinct across all $\mathbf{z}_t$.

Second, since the normalizing condition

$$\int_{\hat{\mathcal{X}}_{t+1}} p_{\hat{\mathbf{x}}_{t+1}|\hat{\mathbf{x}}_t,\hat{\mathbf{z}}_t}\, d\hat{\mathbf{x}}_{t+1} = 1 \tag{A30}$$

must hold for every $\hat{\mathbf{z}}_t$, one only solution is to set $C = 1$, that is, the scaling indeterminacy is resolved.

Ultimately, the unorder of eigenvalues/eigenfunctions is left. We have match the observational distributions $\mathbf{x}_t, \mathbf{x}_{t-1}, \mathbf{x}_{t+1}$, hence, the operator, $L_{\mathbf{x}_{t+1}|\mathbf{x}_t,\mathbf{z}_t}$, corresponding to the set $\{p_{\mathbf{x}_{t+1}|\mathbf{x}_t,\mathbf{z}_t}(\cdot \mid \mathbf{x}_t, \mathbf{z}_t)\}$ for all $\mathbf{x}_t, \mathbf{z}_t$, admits a unique solution (orderibng ambiguity of eigendecomposition only changes the entry position):

$$\{p_{\mathbf{x}_{t+1}|\mathbf{x}_t,\mathbf{z}_t}(\cdot \mid \mathbf{x}_t, \mathbf{z}_t)\} = \{p_{\mathbf{x}_{t+1}|\hat{\mathbf{x}}_t,\hat{\mathbf{z}}_t}(\mathbf{x}_{t+1} \mid \hat{\mathbf{x}}_t, \hat{\mathbf{z}}_t)\}, \quad \text{for all } \mathbf{x}_t, \mathbf{z}_t, \hat{\mathbf{x}}_t, \hat{\mathbf{z}}_t \tag{A31}$$

Due to the set is unorder, the only way to match the R.H.S. with the L.H.S. in a consistent order is to exchange the conditioning variables, that is,

$$\{p_{\mathbf{x}_{t+1}|\mathbf{x}_t,\mathbf{z}_t}(\cdot \mid \mathbf{x}_t^{(1)}, \mathbf{z}_t^{(1)}), p_{\mathbf{x}_{t+1}|\mathbf{x}_t,\mathbf{z}_t}(\cdot \mid \mathbf{x}_t^{(2)}, \mathbf{z}_t^{(2)}), \dots\}$$
$$= \{p_{\mathbf{x}_{t+1}|\hat{\mathbf{x}}_t,\hat{\mathbf{z}}_t}(\cdot \mid \hat{\mathbf{x}}_t^{(1)}, \hat{\mathbf{z}}_t^{(1)}), p_{\mathbf{x}_{t+1}|\hat{\mathbf{x}}_t,\hat{\mathbf{z}}_t}(\cdot \mid \hat{\mathbf{x}}_t^{(2)}, \hat{\mathbf{z}}_t^{(2)}), \dots\}$$

$$\Rightarrow \quad [p_{\mathbf{x}_{t+1}|\mathbf{x}_t,\mathbf{z}_t}(\cdot \mid \mathbf{x}_t^{(\pi(1))}, \mathbf{z}_t^{(\pi(1))}), p_{\mathbf{x}_{t+1}|\mathbf{x}_t,\mathbf{z}_t}(\cdot \mid \mathbf{x}_t^{(\pi(2))}, \mathbf{z}_t^{(\pi(2))}), \dots]$$
$$= [p_{\mathbf{x}_{t+1}|\hat{\mathbf{x}}_t,\hat{\mathbf{z}}_t}(\cdot \mid \hat{\mathbf{x}}_t^{(\pi(1))}, \hat{\mathbf{z}}_t^{(\pi(1))}), p_{\mathbf{x}_{t+1}|\hat{\mathbf{x}}_t,\hat{\mathbf{z}}_t}(\cdot \mid \hat{\mathbf{x}}_t^{(\pi(2))}, \hat{\mathbf{z}}_t^{(\pi(2))}), \dots]$$

where superscript $(\cdot)$ denotes the index of the conditioning variables $[\mathbf{x}_t, \mathbf{z}_t]$, and $\pi$ is reindexing the conditioning variables. We use a relabeling map $H$ to represent its corresponding value mapping:

$$p_{\mathbf{x}_{t+1}|\mathbf{x}_t,\mathbf{z}_t}(\cdot \mid H(\mathbf{x}_t, \mathbf{z}_t)) = p_{\mathbf{x}_{t+1}|\hat{\mathbf{x}}_t,\hat{\mathbf{z}}_t}(\cdot \mid \hat{\mathbf{x}}_t, \hat{\mathbf{z}}_t), \quad \text{for all } \mathbf{x}_t, \mathbf{z}_t, \hat{\mathbf{x}}_t, \hat{\mathbf{z}}_t \tag{A32}$$

By Assumption A2, different $x^*$ corresponds to different $p_{\mathbf{x}_{t+1}|\mathbf{x}_t,\mathbf{z}_t}(\cdot \mid H(\mathbf{x}_t, \mathbf{z}_t))$, there is no repeated element in $\{p_{\mathbf{x}_{t+1}|\mathbf{x}_t,\mathbf{z}_t}(\cdot \mid H(\mathbf{x}_t, \mathbf{z}_t))\}$ (and $\{p_{\mathbf{x}_{t+1}|\hat{\mathbf{x}}_t,\hat{\mathbf{z}}_t}(\cdot \mid \hat{\mathbf{x}}_t, \hat{\mathbf{z}}_t)\}$). Hence, the relabelling map $H$ is one-to-one.

Furthermore, Assumption A2 implies that $p_{\mathbf{x}_{t+1},|\mathbf{x}_t,\mathbf{z}_t}(\cdot \mid H(\mathbf{x}_t, \mathbf{z}_t))$ determines a unique $H(\mathbf{x}_t, \mathbf{z}_t)$. The same holds for the $p_{\mathbf{x}_{t+1}|\hat{\mathbf{x}}_t,\hat{\mathbf{z}}_t}(\cdot \mid \hat{\mathbf{x}}_t, \hat{\mathbf{z}}_t)$, implying that

$$p_{\mathbf{x}_{t+1}|\mathbf{x}_t,\mathbf{z}_t}(\cdot \mid H(\mathbf{x}_t, \mathbf{z}_t)) = p_{\mathbf{x}_{t+1}|\hat{\mathbf{x}}_t,\hat{\mathbf{z}}_t}(\cdot \mid \hat{\mathbf{x}}_t, \hat{\mathbf{z}}_t) \implies \hat{\mathbf{x}}_t, \hat{\mathbf{z}}_t = H(\mathbf{x}_t, \mathbf{z}_t), \tag{A33}$$

implying that $\mathbf{x}_t, \mathbf{z}_t$ is block-wise identifiable.

Next, suppose the implemented MLP used in the transition module is differentiable, then we can assert that there exists a functional $M$ such that $M\left[p_{\mathbf{x}_{t+1}|\mathbf{x}_t,\mathbf{z}_t}(\cdot \mid \mathbf{x}_t, \mathbf{z}_t)\right] = H(\mathbf{x}_t, \mathbf{z}_t)$ for all $\mathbf{z}_t \in \mathcal{Z}_t$ and $\mathbf{x}_t \in \mathcal{X}_t$, where $H$ is differentiable, that is, we can learn a differentiable function $H$ that

$$M\left[p_{\mathbf{x}_{t+1}|\hat{\mathbf{x}}_t,\hat{\mathbf{z}}_t}(\cdot \mid \mathbf{x}_t, \mathbf{z}_t)\right] = M\left[p_{\mathbf{x}_{t+1}|\mathbf{x}_t,\mathbf{z}_t}(\cdot \mid H(\mathbf{x}_t, \mathbf{z}_t))\right] = H(\mathbf{x}_t, \mathbf{z}_t), \qquad \text{(A34)}$$

which is equal to $\hat{\mathbf{x}}_t, \hat{\mathbf{z}}_t$ only if $H$ is differentiable. $\qquad \square$

## C.3 More Discussion of injective linear operators

A linear operator can be intuitively understood as a function that maps one distribution of random variables to another. Specifically, when we assume the injectivity of a linear operator in the context of nonparametric identification, we are asserting that distinct input distributions of the operator correspond to distinct output distributions. This injectivity ensures that there is no ambiguity in the transformation from the input space to the output space, making the operator's behavior predictable and identifiable. An example from a real-world scenario can be seen in weather forecasting. The temperature on a given day can be influenced by several previous days' temperatures. If we view the relationship between past and future temperatures as a linear operator, injectivity would mean that each unique pattern of past temperatures leads to a distinct forecast for the future temperature. The injectivity of this operator ensures that the mapping from past weather data to future forecasts does not result in ambiguity, allowing for more accurate and reliable predictions.

For a better understanding of this assumption, we provide several examples that describe the mapping from $p_{\mathbf{a}} \to p_{\mathbf{b}}$, where $\mathbf{a}$ and $\mathbf{b}$ are random variables.

**Example 1** (**Inverse Transformation**). $b = g(a)$, where $g$ is an invertible function.

**Example 2** (**Additive Transformation**). $b = a + \epsilon$, where $p(\epsilon)$ must not vanish everywhere after the Fourier transform (Theorem 2.1 in Mattner [1993]).

**Example 3.** $b = g(a) + \epsilon$, where the same conditions from Examples 1 and 2 are required.

**Example 4** (**Post-linear Transformation**). $b = g_1(g_2(a) + \epsilon)$, a post-nonlinear model with invertible nonlinear functions $g_1, g_2$, combining the assumptions in **Examples 1-3**.

**Example 5** (**Nonlinear Transformation with Exponential Family**). $b = g(a, \epsilon)$, where the joint distribution $p(a, b)$ follows an exponential family.

**Example 6** (**General Nonlinear Transformation**). $b = g(a, \epsilon)$, a general nonlinear formulation. Certain deviations from the nonlinear additive model (**Example 3**), e.g., polynomial perturbations, can still be tractable.

## C.4 Monotonicity and Normalization Assumption

**Assumption 1** (Monotonicity and Normalization Assumption [Hu and Shum, 2012]). *For any* $\mathbf{x}_t \in \mathcal{X}_t$, *there exists a known functional* $G$ *such that* $G\left[p_{\mathbf{x}_{t+1}|\mathbf{x}_t,\mathbf{z}_t}(\cdot|\mathbf{x}_t, \mathbf{z}_t)\right]$ *is monotonic in* $\mathbf{z}_t$. *We normalize* $\mathbf{z}_t = G\left[p_{\mathbf{x}_{t+1}|\mathbf{x}_t,\mathbf{z}_t}(\cdot|\mathbf{x}_t, \mathbf{z}_t)\right]$.

## C.5 Definition of Neighborhood

**Definition 6** (**Neighborhood**). *Givien a point* $x$ *in a metric space, and a positive number* $r$, *the neighborhood* $N^t$ *of* $x$ *is defined as:*

$$N^r(x) = \{y : d(x, y) < r\}. \qquad \text{(A35)}$$

## C.6 More Discussion of Uniqueness of Spectral Decomposition

This assumption essentially states that, in order to identify the latent variables of the system, it is necessary to observe four different transitions of the observed variables that are governed by the same latent variables. For a better understanding of this assumption, we provide an economic model. Consider an economic model where $\mathbf{x}_t$ represents the inflation rate at time $t$, and $\mathbf{z}_t$ represents the economic regime (such as a recession or a period of growth). To accurately identify the economic regime, we would need to observe inflation under four distinct scenarios: transitions from a high-inflation state to a low-inflation state, and from a low-inflation state to a high-inflation state, under

different historical conditions. These four observed inflation transitions allow us to identify whether the economy is in a recession or growth phase, based on the changes in inflation behavior.

This assumption is straightforward to satisfy in real-world economic modeling, especially when there is access to sufficient historical inflation data. In practice, there are often multiple transitions between inflation states over time, corresponding to shifts in the economic regime (e.g., moving from high inflation during an economic boom to low inflation during a recession). By collecting enough observations across different periods of economic change, this assumption can be easily fulfilled, ensuring that we can identify the underlying economic regime with confidence.

## C.7 Proof of Theorem 3.

**Lemma A1.** *(Component-wise Identification of $\mathbf{z}_t$ with instantaneous dependencies under sparse causal influence on latent dynamics.)Li et al. [2025] For a series of observed variables $\mathbf{x}_t \in \mathbb{R}^n$ and estimated latent variables $\hat{\mathbf{z}}_t \in \mathbb{R}^n$ with the corresponding process $\hat{f}_i, \hat{p}(\epsilon), \hat{\mathbf{g}}$, where $\hat{\mathbf{g}}$ is invertible, suppose the process subject to observational equivalence $\mathbf{x}_t = \hat{\mathbf{g}}(\hat{\mathbf{z}}_t)$. Let $\mathbf{c}_t \triangleq \{\mathbf{z}_{t-1}, \mathbf{z}_t\} \in \mathbb{R}^{2n}$ and $\mathcal{M}_{\mathbf{c}_t}$ be the variable set of two consecutive timestamps and the corresponding Markov network, respectively. Suppose the following assumptions hold:*

- *A4 (Smooth and Positive Density): The conditional probability function of the latent variables $\mathbf{c}_t$ is smooth and positive, i.e., $p(\mathbf{c}_t|\mathbf{z}_{t-2})$ is third-order differentiable and $p(\mathbf{c}_t|\mathbf{z}_{t-2}) > 0$ over $\mathbb{R}^{2n}$,*

- *A5 (Sufficient Variability): Denote $|\mathcal{M}_{\mathbf{c}_t}|$ as the number of edges in Markov network $\mathcal{M}_{\mathbf{c}_t}$. Let*

$$
w(m) = \Big( \frac{\partial^3 \log p(\mathbf{c}_t|\mathbf{z}_{t-2})}{\partial c_{t,1}^2 \partial z_{t-2,m}}, \cdots, \frac{\partial^3 \log p(\mathbf{c}_t|\mathbf{z}_{t-2})}{\partial c_{t,2n}^2 \partial z_{t-2,m}} \Big) \oplus
$$
$$
\Big( \frac{\partial^2 \log p(\mathbf{c}_t|\mathbf{z}_{t-2})}{\partial c_{t,1} \partial z_{t-2,m}}, \cdots, \frac{\partial^2 \log p(\mathbf{c}_t|\mathbf{z}_{t-2})}{\partial c_{t,2n} \partial z_{t-2,m}} \Big) \oplus \Big( \frac{\partial^3 \log p(\mathbf{c}_t|\mathbf{z}_{t-2})}{\partial c_{t,i} \partial c_{t,j} \partial z_{t-2,m}} \Big)_{(i,j)\in\mathcal{E}(\mathcal{M}_{\mathbf{c}_t})},
$$
(A36)

*where $\oplus$ denotes concatenation operation and $(i,j) \in \mathcal{E}(\mathcal{M}_{\mathbf{c}_t})$ denotes all pairwise indice such that $c_{t,i}, c_{t,j}$ are adjacent in $\mathcal{M}_{\mathbf{c}_t}$. For $m \in [1, \cdots, n]$, there exist $4n + |\mathcal{M}_{\mathbf{c}_t}|$ different values of $\mathbf{z}_{t-2,m}$, such that the $4n + |\mathcal{M}_{\mathbf{c}_t}|$ values of vector functions $w(m)$ are linearly independent.*

- *A6 (Latent Process Sparsity): For any $z_{it} \in \mathbf{z}_t$, the intimate neighbor set of $z_{it}$ is an empty set.*

*When the observational equivalence is achieved with the minimal number of edges of the estimated Markov network of $\mathcal{M}_{\hat{\mathbf{c}}_t}$, there exists a permutation $\pi$ of the estimated latent variables, such that $z_{it}$ and $\hat{z}_{\pi(i)t}$ is one-to-one corresponding, i.e., $z_{it}$ is component-wise identifiable.*

*Proof.* The proof can be summarized into three steps. First, we leverage the sparsity among latent variables to show the relationships between ground-truth and estimated latent variables. Sequentially, we show that the estimated Markov network $\mathcal{M}_{\hat{\mathbf{c}}_t}$ is isomorphic to the ground-truth Markov networks $\mathcal{M}_{\mathbf{c}_t}$. Finally, we show that the latent variables are component-wise identifiable under the sparse mixture procedure condition.

**Step1: Relationships between Ground-truth and Estimated Latent Variables.** We start from the matched marginal distribution to develop the relationship between $\mathbf{z}_t$ and $\hat{\mathbf{z}}_t$ as follows:

$$
p(\hat{\mathbf{x}}_t) = p(\mathbf{x}_t) \Longleftrightarrow p(\hat{g}(\hat{\mathbf{z}}_t)) = p(g(z_t)) \Longleftrightarrow p((g^{-1} \circ \hat{g})(\hat{\mathbf{z}}_t)) = p(\mathbf{z}_t) \Longleftrightarrow p(h_z(\hat{\mathbf{z}}_t)) = p(\mathbf{z}_t), \quad \text{(A37)}
$$

where $\hat{g} : \mathcal{Z} \to \mathcal{X}$ denotes the estimated mixing function, and $h := g^{-1} \circ \hat{g}$ is the transformation between the ground-truth latent variables and the estimated ones. Since $\hat{g}$ and $g$ are invertible, $h$ is invertible as well. Since Equation (A37) holds true for all time steps, there must exist an invertible function $h_c$ such that $p(h_c(\hat{\mathbf{c}}_t)) = p(\mathbf{c}_t)$, whose Jacobian matrix at time step $t$ is

$$
\mathbf{J}_{h_c,t} = \begin{bmatrix} \mathbf{J}_{h_z,t-1} & 0 \\ 0 & \mathbf{J}_{h_z,t} \end{bmatrix}.
\quad \text{(A38)}
$$

Then for each value of $\mathbf{x}_{t-2}$, the Jacobian matrix of the mapping from $(\mathbf{x}_{t-2}, \hat{\mathbf{c}}_t)$ to $(\mathbf{x}_{t-2}, \mathbf{c}_t)$ can be written as follows:

$$
\begin{bmatrix} \mathbf{I} & \mathbf{0} \\ * & \mathbf{J}_{h_c,t} \end{bmatrix},
$$

where $*$ denotes any matrix. Since $\mathbf{x}_{t-2}$ can be fully characterized by itself, the left top and right top block are $\mathbf{1}$ and $\mathbf{0}$ respectively, and the determinant of this Jacobian matrix is the same as $|\mathbf{J}_{h_c,t}|$. Therefore, we have:

$$p(\hat{\mathbf{c}}_t, \mathbf{x}_{t-2}) = p(\mathbf{c}_t, \mathbf{x}_{t-2})|\mathbf{J}_{h_c,t}|. \tag{A39}$$

Dividing both sides of Equation (A39) by $p(\mathbf{x}_{t-2})$, we further have:

$$p(\hat{\mathbf{c}}_t|\mathbf{x}_{t-2}) = p(\mathbf{c}_t|\mathbf{x}_{t-2})|\mathbf{J}_{h_c,t}|. \tag{A40}$$

Since $p(\mathbf{c}_t|\mathbf{x}_{t-2}) = p(\mathbf{c}_t|g(\mathbf{z}_{t-2})) = p(\mathbf{c}_t|\mathbf{z}_{t-2})$, and similarly $p(\hat{\mathbf{c}}_t|\mathbf{x}_{t-2}) = p(\hat{\mathbf{c}}_t|\hat{\mathbf{z}}_{t-2})$, we have:

$$\log p(\hat{\mathbf{c}}_t|\hat{\mathbf{z}}_{t-2}) = \log p(\mathbf{c}_t|\mathbf{z}_{t-2}) + \log|\mathbf{J}_{h_c,t}|. \tag{A41}$$

Let $\hat{c}_{t,k}, \hat{c}_{t,l}$ be two different variables that are not adjacent in the estimated Markov network $\mathcal{M}_{\hat{\mathbf{c}}_t}$ over $\hat{\mathbf{c}}_t = \{\hat{\mathbf{z}}_{t-1}, \hat{\mathbf{z}}_t\}$. We conduct the first-order derivative w.r.t. $\hat{c}_{t,k}$ and have

$$\frac{\partial \log p(\hat{\mathbf{c}}_t|\hat{\mathbf{z}}_{t-2})}{\partial \hat{c}_{t,k}} = \sum_{i=1}^{2n} \frac{\partial \log p(\mathbf{c}_t|\mathbf{z}_{t-2})}{\partial c_{t,i}} \cdot \frac{\partial c_{t,i}}{\partial \hat{c}_{t,k}} + \frac{\partial \log|\mathbf{J}_{h_c,t}|}{\partial \hat{c}_{t,k}}. \tag{A42}$$

We further conduct the second-order derivative w.r.t. $\hat{c}_{t,k}$ and $\hat{c}_{t,l}$, then we have:

$$\begin{aligned}
\frac{\partial^2 \log p(\hat{\mathbf{c}}_t|\hat{\mathbf{z}}_{t-2})}{\partial \hat{c}_{t,k} \partial \hat{c}_{t,l}} = & \sum_{i=1}^{2n} \sum_{j=1}^{2n} \frac{\partial^2 \log p(\mathbf{c}_t|\mathbf{z}_{t-2})}{\partial c_{t,i} \partial c_{t,j}} \cdot \frac{\partial c_{t,i}}{\partial \hat{c}_{t,k}} \cdot \frac{\partial c_{t,j}}{\partial \hat{c}_{t,l}} \\
& + \sum_{i=1}^{2n} \frac{\partial \log p(\mathbf{c}_t|\mathbf{z}_{t-2})}{\partial c_{t,i}} \cdot \frac{\partial^2 c_{t,i}}{\partial \hat{c}_{t,k} \partial \hat{c}_{t,l}} + \frac{\partial^2 \log|\mathbf{J}_{h_c,t}|}{\partial \hat{c}_{t,k} \partial \hat{c}_{t,l}}.
\end{aligned} \tag{A43}$$

Since $\hat{c}_{t,k}, \hat{c}_{t,l}$ are not adjacent in $\mathcal{M}_{\hat{\mathbf{c}}_t}$, $\hat{c}_{t,k}$ and $\hat{c}_{t,l}$ are conditionally independent given $\hat{\mathbf{c}}_t \backslash \{\hat{c}_{t,k}, \hat{c}_{t,l}\}$. Utilizing the fact that conditional independence can lead to zero cross derivative [Lin, 1997], for each value of $\hat{\mathbf{z}}_{t-2}$, we have

$$\begin{aligned}
\frac{\partial^2 \log p(\hat{\mathbf{c}}_t|\hat{\mathbf{z}}_{t-2})}{\partial \hat{c}_{t,k} \partial \hat{c}_{t,l}} = & \frac{\partial^2 \log p(\hat{c}_{t,k}|\hat{\mathbf{c}}_t \backslash \{\hat{c}_{t,k}, \hat{c}_{t,l}\}, \hat{\mathbf{z}}_{t-2})}{\partial \hat{c}_{t,k} \partial \hat{c}_{t,l}} + \frac{\partial^2 \log p(\hat{c}_{t,l}|\hat{\mathbf{c}}_t \backslash \{\hat{c}_{t,k}, \hat{c}_{t,l}\}, \hat{\mathbf{z}}_{t-2})}{\partial \hat{c}_{t,k} \partial \hat{c}_{t,l}} \\
& + \frac{\partial^2 \log p(\hat{\mathbf{c}}_t \backslash \{\hat{c}_{t,k}, \hat{c}_{t,l}\}|\hat{\mathbf{z}}_{t-2})}{\partial \hat{c}_{t,k} \partial \hat{c}_{t,l}} = 0.
\end{aligned} \tag{A44}$$

Bring in Equation (A44), Equation (A43) can be further derived as

$$\begin{aligned}
0 = & \underbrace{\sum_{i=1}^{2n} \frac{\partial^2 \log p(\mathbf{c}_t|\mathbf{z}_{t-2})}{\partial c_{t,i}^2} \cdot \frac{\partial c_{t,i}}{\partial \hat{c}_{t,k}} \cdot \frac{\partial c_{t,i}}{\partial \hat{c}_{t,l}}}_{\textbf{(i) } i=j} + \underbrace{\sum_{i=1}^{2n} \sum_{j:(j,i)\in\mathcal{E}(\mathcal{M}_{\mathbf{c}_t})} \frac{\partial^2 \log p(\mathbf{c}_t|\mathbf{z}_{t-2})}{\partial c_{t,i} \partial c_{t,j}} \cdot \frac{\partial c_{t,i}}{\partial \hat{c}_{t,k}} \cdot \frac{\partial c_{t,j}}{\partial \hat{c}_{t,l}}}_{\textbf{(ii)} c_{t,i} \text{ and } c_{t,j} \text{ are adjacent in } \mathcal{M}_{\mathbf{c}_t}} \\
& + \underbrace{\sum_{i=1}^{2n} \sum_{j:(j,i)\notin\mathcal{E}(\mathcal{M}_{\mathbf{c}_t})} \frac{\partial^2 \log p(\mathbf{c}_t|\mathbf{z}_{t-2})}{\partial c_{t,i} \partial c_{t,j}} \cdot \frac{\partial c_{t,i}}{\partial \hat{c}_{t,k}} \cdot \frac{\partial c_{t,j}}{\partial \hat{c}_{t,l}}}_{\textbf{(iii)} c_{t,i} \text{ and } c_{t,j} \text{ are \textbf{not} adjacent in } \mathcal{M}_{\mathbf{c}_t}} \\
& + \sum_{i=1}^{2n} \frac{\partial \log p(\mathbf{c}_t|\mathbf{z}_{t-2})}{\partial c_{t,i}} \cdot \frac{\partial^2 c_{t,i}}{\partial \hat{c}_{t,k} \partial \hat{c}_{t,l}} + \frac{\partial \log|\mathbf{J}_{h_c,t}|}{\partial \hat{c}_{t,k} \partial \hat{c}_{t,l}},
\end{aligned} \tag{A45}$$

where $(j,i) \in \mathcal{E}(\mathcal{M}_{\mathbf{c}_t})$ denotes that $c_{t,i}$ and $c_{t,j}$ are adjacent in $\mathcal{M}_{\mathbf{c}_t}$. Similar to Equation (A44), we have $\frac{\partial^2 p(\mathbf{c}_t|\mathbf{z}_{t-2})}{\partial c_{t,i} \partial c_{t,j}} = 0$ when $c_{t,i}, c_{t,j}$ are not adjacent in $\mathcal{M}_{\mathbf{c}_t}$. Thus, Equation (A45) can be rewritten as

$$\begin{aligned}
0 = & \sum_{i=1}^{2n} \frac{\partial^2 \log p(\mathbf{c}_t|\mathbf{z}_{t-2})}{\partial c_{t,i}^2} \cdot \frac{\partial c_{t,i}}{\partial \hat{c}_{t,k}} \cdot \frac{\partial c_{t,i}}{\partial \hat{c}_{t,l}} + \sum_{i=1}^{2n} \sum_{j:(j,i)\in\mathcal{E}(\mathcal{M}_{\mathbf{c}})} \frac{\partial^2 \log p(\mathbf{c}_t|\mathbf{z}_{t-2})}{\partial c_{t,i} \partial c_{t,j}} \cdot \frac{\partial c_{t,i}}{\partial \hat{c}_{t,k}} \cdot \frac{\partial c_{t,j}}{\partial \hat{c}_{t,l}} \\
& + \sum_{i=1}^{2n} \frac{\partial \log p(\mathbf{c}_t|\mathbf{z}_{t-2})}{\partial c_{t,i}} \cdot \frac{\partial^2 c_{t,i}}{\partial \hat{c}_{t,k} \partial \hat{c}_{t,l}} + \frac{\partial \log|\mathbf{J}_{h_c,t}|}{\partial \hat{c}_{t,k} \partial \hat{c}_{t,l}}.
\end{aligned} \tag{A46}$$

Then for each $m = 1, 2, \cdots, n$ and each value of $z_{t-2,m}$, we conduct partial derivative on both sides of Equation (A46) and have:

$$
0 = \sum_{i=1}^{2n} \frac{\partial^3 \log p(\mathbf{c}_t | \mathbf{z}_{t-2})}{\partial c_{t,i}^2 \partial z_{t-2,m}} \cdot \frac{\partial c_{t,i}}{\partial \hat{c}_{t,k}} \cdot \frac{\partial c_{t,i}}{\partial \hat{c}_{t,l}} + \sum_{i=1}^{2n} \sum_{j:(j,i) \in \mathcal{E}(\mathcal{M}_{\mathbf{c}})} \frac{\partial^3 \log p(\mathbf{c}_t | \mathbf{z}_{t-2})}{\partial c_{t,i} \partial c_{t,j} \partial z_{t-2,m}} \cdot \frac{\partial c_{t,i}}{\partial \hat{c}_{t,k}} \cdot \frac{\partial c_{t,j}}{\partial \hat{c}_{t,l}}
$$
$$
+ \sum_{i=1}^{2n} \frac{\partial^2 \log p(c_t | \mathbf{z}_{t-2})}{\partial c_{t,i} \partial z_{t-2,m}} \cdot \frac{\partial c_{t,i}^2}{\partial \hat{c}_{t,k} \partial \hat{c}_{t,l}}
$$

$$\tag{A47}$$

Finally we have

$$
0 = \sum_{i=1}^{2n} \frac{\partial^3 \log p(\mathbf{c}_t | \mathbf{z}_{t-2})}{\partial c_{t,i}^2 \partial z_{t-2,m}} \cdot \frac{\partial c_{t,i}}{\partial \hat{c}_{t,k}} \cdot \frac{\partial c_{t,i}}{\partial \hat{c}_{t,l}} + \sum_{i=1}^{2n} \frac{\partial^2 \log p(c_t | \mathbf{z}_{t-2})}{\partial c_{t,i} \partial z_{t-2,m}} \cdot \frac{\partial c_{t,i}^2}{\partial \hat{c}_{t,k} \partial \hat{c}_{t,l}}
$$
$$
+ \sum_{i,j:(j,i) \in \mathcal{E}(\mathcal{M}_{\mathbf{c}})} \frac{\partial^3 \log p(\mathbf{c}_t | \mathbf{z}_{t-2})}{\partial c_{t,i} \partial c_{t,j} \partial z_{t-2,m}} \cdot \left( \frac{\partial c_{t,i}}{\partial \hat{c}_{t,k}} \cdot \frac{\partial c_{t,j}}{\partial \hat{c}_{t,l}} + \frac{\partial c_{t,j}}{\partial \hat{c}_{t,k}} \cdot \frac{\partial c_{t,i}}{\partial \hat{c}_{t,l}} \right)
$$

$$\tag{A48}$$

According to Assumption A2, we can construct $4n + |\mathcal{M}_{\mathbf{c}}|$ different equations with different values of $z_{t-2,m}$, and the coefficients of the equation system they form are linearly independent. To ensure that the right-hand side of the equations are always 0, the only solution is

$$
\frac{\partial c_{t,i}}{\partial \hat{c}_{t,k}} \cdot \frac{\partial c_{t,i}}{\partial \hat{c}_{t,l}} = 0,
$$

$$\tag{A49}$$

$$
\frac{\partial c_{t,i}}{\partial \hat{c}_{t,k}} \cdot \frac{\partial c_{t,j}}{\partial \hat{c}_{t,l}} + \frac{\partial c_{t,j}}{\partial \hat{c}_{t,k}} \cdot \frac{\partial c_{t,i}}{\partial \hat{c}_{t,l}} = 0,
$$

$$\tag{A50}$$

$$
\frac{\partial c_{t,i}^2}{\partial \hat{c}_{t,k} \partial \hat{c}_{t,l}} = 0.
$$

$$\tag{A51}$$

Bringing Eq A49 into Eq A50, at least one product must be zero, thus the other must be zero as well. That is,

$$
\frac{\partial c_{t,i}}{\partial \hat{c}_{t,k}} \cdot \frac{\partial c_{t,j}}{\partial \hat{c}_{t,l}} = 0.
$$

$$\tag{A52}$$

According to the aforementioned results, for any two different entries $\hat{c}_{t,k}, \hat{c}_{t,l} \in \hat{\mathbf{c}}_t$ that are **not adjacent** in the Markov network $\mathcal{M}_{\hat{\mathbf{c}}_t}$ over estimated $\hat{\mathbf{c}}_t$, we draw the following conclusions.
**(i)** Equation (A49) implies that, each ground-truth latent variable $c_{t,i} \in \mathbf{c}_t$ is a function of at most one of $\hat{c}_{t,k}$ and $\hat{c}_{t,l}$,
**(ii)** Equation (A52) implies that, for each pair of ground-truth latent variables $c_{t,i}$ and $c_{t,j}$ that are **adjacent** in $\mathcal{M}_{\mathbf{c}_t}$ over $\mathbf{c}_t$, they can not be a function of $\hat{c}_{t,k}$ and $\hat{c}_{t,l}$ respectively.

**Step2: Isomorphism of Markov Networks** First, we demonstrate that there always exists a row permutation for each invertible matrix such that the permuted diagonal entries are non-zero [Zhang et al., 2024a]. By contradiction, if the product of the diagonal entry of an invertible matrix $A$ is zero for every row permutation, then we have Equation

$$
\det(A) = \sum_{\sigma \in \mathcal{S}_n} \left( \operatorname{sgn}(\sigma) \prod_{i=1}^{n} a_{\sigma(i),i} \right),
$$

$$\tag{A53}$$

by the Leibniz formula, where $\mathcal{S}_n$ is the set of $n$-permutations. Thus, we have

$$
\prod_{i=1}^{n} a_{\sigma(i),i} = 0, \quad \forall \sigma \in \mathcal{S}_n,
$$

$$\tag{A54}$$

which indicates that $det(A) = 0$ and $A$ is non-invertible. It contradicts the assumption that $A$ is invertible, and a row permutation where the permuted diagonal entries are non-zero must exist. Since

$h_z$ is invertible, for $\mathbf{z}_t$ at time step $t$, there exists a permuted version of the estimated latent variables, such that

$$\frac{\partial z_{t,i}}{\partial \hat{z}_{t,\pi_t(i)}} \neq 0, \quad i = 1, \cdots, n, \tag{A55}$$

where $\pi_t$ is the corresponding permutation at time step $t$. Since $\mathbf{c}_t = \{\mathbf{z}_{t-1}, \mathbf{z}_t, \mathbf{z}_{t+1}\}$, by applying $\pi_{t-1}, \pi_t, \pi_{t+1}$, we have $\pi'$ such that

$$\frac{\partial c_{t,i}}{\partial \hat{c}_{t,\pi'(i)}} \neq 0, \quad i = 1, \cdots, 3n. \tag{A56}$$

Second, we demonstrate that $\mathcal{M}_{\mathbf{c}_t}$ is identical to $\mathcal{M}_{\hat{\mathbf{c}}_t^{\pi'}}$, where $\mathcal{M}_{\hat{\mathbf{c}}_t^{\pi'}}$ denotes the Markov network of the permuted version of $\pi'(\hat{\mathbf{c}}_t)$.

**Step3: Component-wise Identification of Latent Variables**  Finally, we prove that the latent variables are component-wise identifiable. On the one hand, for any pair of $(i, j)$ such that $c_{t,i}, c_{t,j}$ are **adjacent** in $\mathcal{M}_{\mathbf{c}_t}$ while $\hat{c}_{t,\pi'(i)}, \hat{c}_{t,\pi'(j)}$ are **not adjacent** in $\mathcal{M}_{\hat{\mathbf{c}}_t^{\pi'}}$, according to Equation (A52), we have $\frac{\partial c_{t,i}}{\partial \hat{c}_{t,\pi'(i)}} \cdot \frac{\partial c_{t,j}}{\partial \hat{c}_{t,\pi'(j)}} = 0$, which is a contradiction with how $\pi'$ is constructed. Thus, any edge presents in $\mathcal{M}_{\mathbf{c}_t}$ must exist in $\mathcal{M}_{\hat{\mathbf{c}}_t^{\pi'}}$. On the other hand, since observational equivalence can be achieved by the true latent process $(g, f, p_{\mathbf{c}_t})$, the true latent process is clearly the solution with minimal edges.

Under the sparsity constraint on the edges of $\mathcal{M}_{\hat{\mathbf{c}}_t^{\pi'}}$, the permuted estimated Markov network $\mathcal{M}_{\hat{\mathbf{c}}_t^{\pi'}}$ must be identical to the true Markov network $\mathcal{M}_{\mathbf{c}_t}$. Thus, we claim that

(i) the estimated Markov network $\mathcal{M}_{\hat{\mathbf{c}}_t}$ is isomorphic to the ground-truth Markov network $\mathcal{M}_{\mathbf{c}_t}$.

Sequentially, under the same permutation $\pi_t$, we further give the proof that $z_{t,i}$ is only the function of $\hat{z}_{t,\pi_t(i)}$. Since the permutation happens on each time step respectively, the cross-time disentanglement is prevented clearly.

Now let us focus on instantaneous disentanglement. Suppose there exists a pair of indices $i, j \in \{1, \cdots, n\}$. According to Equation (A55), we have $\frac{\partial z_{t,i}}{\partial \hat{z}_{t,\pi_t(i)}} = 0$ and $\frac{\partial z_{t,j}}{\partial \hat{z}_{t,\pi_t(j)}} = 0$. Let us discuss it case by case.

- If $z_{t,i}$ is not adjacent to $z_{t,j}$, we have $\hat{z}_{t,\pi_t(i)}$ is not adjacent to $\hat{z}_{t,\pi_t(j)}$ as well according to the conclusion of identical Markov network. Using Equation (A49), we have $\frac{\partial z_{t,i}}{\partial \hat{z}_{t,\pi_t(i)}} \cdot \frac{\partial z_{t,i}}{\partial \hat{z}_{t,\pi_t(j)}} = 0$, which leads to $\frac{\partial z_{t,i}}{\partial \hat{z}_{t,\pi_t(j)}} = 0$.

- If $z_{t,i}$ is adjacent to $z_{t,j}$, we have $\hat{z}_{t,\pi_t(i)}$ is adjacent to $\hat{z}_{t,\pi_t(j)}$. When the Assumption A3 (Sparse Latent Process) is assured, i.e., the intimate neighbor set of $z_{t,i}$ is empty, there exists at least one pair of $(t', k)$ such that $z_{t',k}$ is adjacent to $z_{t,i}$ but not adjacent to $z_{t,j}$. Similarly, we have the same structure on the estimated Markov network, which means that $\hat{z}_{t',\pi_{t'}(k)}$ is adjacent to $\hat{z}_{t,\pi_t(i)}$ but not adjacent to $\hat{z}_{t,\pi_t(j)}$. Using Equation (A52) we have $\frac{\partial z_{t,k}}{\partial \hat{z}_{t',\pi_{t'}(k)}} \cdot \frac{\partial z_{t,i}}{\partial \hat{z}_{t,\pi_t(j)}} = 0$, which leads to $\frac{\partial z_{t,i}}{\partial \hat{z}_{t,\pi_t(j)}} = 0$.

In conclusion, we always have $\frac{\partial z_{t,i}}{\partial \hat{z}_{t,\pi_t(j)}} = 0$. Thus, we have reached the conclusion that

(ii) there exists a permutation $\pi$ of the estimated latent variables, such that $z_{t,i}$ and $\hat{z}_{t,\pi(i)}$ is one-to-one corresponding, i.e., $z_{t,i}$ is component-wise identifiable.

$\square$

**Theorem A3.** *(Component-wise Identification of $\mathbf{z}_t$ under sparse mixing procedure.) For a series of observations $\mathbf{x}_t \in \mathbb{R}^n$ and estimated latent variables $\hat{\mathbf{z}}_t \in \mathbb{R}^n$ with the corresponding process $\hat{f}_i, \hat{p}(\epsilon), \hat{g}$, suppose the marginal distribution of observed variables is matched. Let $\mathcal{M}_{\mathbf{u}_t}$ be the Markov network over $\mathbf{u}_t \triangleq \{\mathbf{z}_{t-1}, \mathbf{x}_{t-1}, \mathbf{z}_t, \mathbf{x}_t\}$ and $\mathcal{M}_{\mathbf{u}_t}$. Besides the similar assumptions like smooth, positive density, and sufficient variability assumptions, we further assume:*

• *A7 (Sparse Mixing Procedure): For any $\mathbf{z}_{it} \in \mathbf{z}_t$, the intimate neighbor set of $\mathbf{z}_{it}$ is an empty set.*

*When the observational equivalence is achieved with the minimal number of edges of the estimated mixing procedure, there exists a permutation $\pi$ of the estimated latent variables, such that $\mathbf{z}_{it}$ and $\hat{\mathbf{z}}_{\pi(i)t}$ is one-to-one corresponding, i.e., $\mathbf{z}_{it}$ is component-wise identifiable.*

*Proof.* By reusing Theorem 2 with more observations, $(\mathbf{z}_{t-1}, \mathbf{x}_{t-1}, \mathbf{z}_t, \mathbf{x}_t)$ is also block-wise identifiable. So we have:

$$
\begin{aligned}
& p(\hat{\mathbf{z}}_t, \hat{\mathbf{x}}_t, \hat{\mathbf{z}}_{t-1}, \hat{\mathbf{x}}_{t-1}) = p(\mathbf{z}_t, \mathbf{x}_t, \hat{\mathbf{z}}_{t-1}, \hat{\mathbf{x}}_{t-1})|\mathbf{J}_h| \\
\iff & p(\hat{\mathbf{z}}_t, \hat{\mathbf{x}}_t | \hat{\mathbf{z}}_{t-1}, \hat{\mathbf{x}}_{t-1}) = p(\mathbf{z}_t, \mathbf{x}_t | \hat{\mathbf{z}}_{t-1}, \hat{\mathbf{x}}_{t-1})|\mathbf{J}_h| \\
\iff & \ln p(\hat{\mathbf{z}}_t, \hat{\mathbf{x}}_t | \hat{\mathbf{z}}_{t-1}, \hat{\mathbf{x}}_{t-1}) = \ln p(\mathbf{z}_t, \mathbf{x}_t | \hat{\mathbf{z}}_{t-1}, \hat{\mathbf{x}}_{t-1}) + \ln |\mathbf{J}_h|,
\end{aligned}
\tag{A57}
$$

where $h : \mathcal{X}, \mathcal{Z} \to \mathcal{X}, \mathcal{Z}$ denotes the invertible transformation. $|\mathbf{J}_h|$ stands for the absolute value of the Jacobian matrix determinant of $h$. For any $\hat{\mathbf{z}}_{t,j}$, suppose that there exist $\hat{\mathbf{x}}_{t,k}$ that $\hat{\mathbf{z}}_{t,j}$ does not contribute to the mixture of $\hat{\mathbf{x}}_{t,k}$.

By using the sparse mixing procedure assumption (A7), we can constrain the sparsity of the estimated mixing function, such that there exist two different estimated latent variables $\hat{\mathbf{u}}_{t,k}$ and $\hat{\mathbf{u}}_{t,l}$ that are not adjacent in the estimated Markov networks $\mathcal{M}_{\mathbf{u}_t}$ and $\frac{\partial^2 \log p(\hat{\mathbf{u}}_t | \hat{\mathbf{z}}_{t-1}, \hat{\mathbf{x}}_{t-1})}{\partial \hat{\mathbf{u}}_{t,k} \partial \hat{\mathbf{u}}_{t,l}} = 0$. Sequentially, we can replace $p(\mathbf{c} | \mathbf{z}_{t-1})$ in Lemma 1 with $p(\hat{\mathbf{u}}_t | \hat{\mathbf{z}}_{t-1}, \hat{\mathbf{x}}_{t-1})$, and then by reusing the proof process of Lemma 1, we can prove that $z$ is component-wise identifiable.

$\square$

## C.8 More Discussion on the Sparse Mixing Procedure

Although recent works like Zheng et al. [2022], Zheng and Zhang [2023] also utilize the sparse mixing process from $\mathbf{z}_t$ to $\mathbf{x}_t$ to achieve identifiability, our assumption is easier to satisfy compared to these methods. The primary reason for this is that our generative process allows for noise in the mixing process from $\mathbf{z}_t$ to $\mathbf{x}_t$, thereby accounting for measurement errors in the observed data. In contrast, methods like Zheng et al. [2022], Zheng and Zhang [2023] require the additional assumption that the mixing process is invertible and free from noise.

# D Experiment Details

## D.1 Synthetic Experiment

### D.1.1 Data Generation Process

We follow Equation (1) to generate the synthetic data. As for the temporally latent processes, we use MLPs with the activation function of LeakyReLU to model the sparse time-delayed. That is:

$$
z_{t,i} = (LeakyReLU(W_{i,:} \cdot \mathbf{z}_{t-1}, 0.2) + V_{<i,i} \cdot \mathbf{z}_{t,<i}) \cdot \epsilon_{t,i} + \epsilon_{t,i}^{\mathbf{z}}
\tag{A58}
$$

where $W_{i,:}$ is the $i$-th row of $W^*$ and $V_{<i,i}$ is the first $i-1$ columns in the $i$-th row of $V$. Moreover, each independent noise $\epsilon_{t,i}$ is sampled from the distribution of normal distribution. We further let the data generation process from latent variables to observed variables be MLPs with LeakyReLU units. And the generation procedure can be formulated as follows:

$$
\mathbf{x}_t = LeakyReLU(LeakyReLU(0.2 \times LeakyReLU(\mathbf{x}_{t-1} \cdot W_{\mathbf{x}}, 0.2) + \mathbf{z}_t + \epsilon_t^{\mathbf{o}}, 0.2) \cdot W_m),
\tag{A59}
$$

where $W_{\mathbf{x}}$ and $W_m$ denote the weights of mixing function. We provide 4 datasets from A to D, whose settings are shown in Table A5.

The total size of the dataset is 100,000, with 1,024 samples designated as the validation set. The remaining samples are the training set.

### D.1.2 Evaluation Metric

To evaluate the identifiability performance of our method under instantaneous dependencies, we employ the Mean Correlation Coefficient (MCC) between the ground-truth $z_t$ and the estimated $\hat{z}_t$.

Table A5: Details of different synthetic datasets.

| | Dimension of Latent Variables | Time Lag | Causal Edge among Observations |
|---|---|---|---|
| A | 5 | 1 | yes |
| B | 5 | 1 | no |
| C | 5 | 2 | yes |
| D | 10 | 1 | yes |

A higher MCC denotes a better identification performance the model can achieve. In addition, we also draw the estimated latent causal process to validate our method. Since the estimated transition function will be a transformation of the ground truth, we do not compare their exact values, but only the activated entries.

### D.1.3 Prior Likelihood Derivation

We first consider the prior of $\ln p(\mathbf{z}_{1:t})$. We start with an illustrative example of stationary latent causal processes with two time-delay latent variables, i.e. $\mathbf{z}_t = [z_{t,1}, z_{t,2}]$ with maximum time lag $L = 1$, i.e., $z_{t,i} = f_i(\mathbf{z}_{t-1}, \epsilon_{t,i})$ with mutually independent noises. Then we write this latent process as a transformation map $\mathbf{f}$ (note that we overload the notation $f$ for transition functions and for the transformation map):

$$
\begin{bmatrix} z_{t-1,1} \\ z_{t-1,2} \\ z_{t,1} \\ z_{t,2} \end{bmatrix} = \mathbf{f} \left( \begin{bmatrix} z_{t-1,1} \\ z_{t-1,2} \\ \epsilon_{t,1} \\ \epsilon_{t,2} \end{bmatrix} \right).
$$

By applying the change of variables formula to the map $\mathbf{f}$, we can evaluate the joint distribution of the latent variables $p(z_{t-1,1}, z_{t-1,2}, z_{t,1}, z_{t,2})$ as

$$
p(z_{t-1,1}, z_{t-1,2}, z_{t,1}, z_{t,2}) = \frac{p(z_{t-1,1}, z_{t-1,2}, \epsilon_{t,1}, \epsilon_{t,2})}{|\det \mathbf{J_f}|}, \tag{A60}
$$

where $\mathbf{J_f}$ is the Jacobian matrix of the map $\mathbf{f}$, where the instantaneous dependencies are assumed to be a low-triangular matrix:

$$
\mathbf{J_f} = \begin{bmatrix} 1 & 0 & 0 & 0 \\ 0 & 1 & 0 & 0 \\ \frac{\partial z_{t,1}}{\partial z_{t-1,1}} & \frac{\partial z_{t,1}}{\partial z_{t-1,2}} & \frac{\partial z_{t,1}}{\partial \epsilon_{t,1}} & 0 \\ \frac{\partial z_{t,2}}{\partial z_{t-1,1}} & \frac{\partial z_{t,2}}{\partial z_{t-1,2}} & \frac{\partial z_{t,2}}{\partial \epsilon_{t,1}} & \frac{\partial z_{t,2}}{\partial \epsilon_{t,2}} \end{bmatrix}.
$$

Given that this Jacobian is triangular, we can efficiently compute its determinant as $\prod_i \frac{\partial z_{t,i}}{\epsilon_{t,i}}$. Furthermore, because the noise terms are mutually independent, and hence $\epsilon_{t,i} \perp \epsilon_{t,j}$ for $j \neq i$ and $\epsilon_t \perp \mathbf{z}_{t-1}$, so we can with the RHS of Equation (A60) as follows

$$
p(z_{t-1,1}, z_{t-1,2}, z_{t,1}, z_{t,2}) = p(z_{t-1,1}, z_{t-1,2}) \times \frac{p(\epsilon_{t,1}, \epsilon_{t,2})}{|\mathbf{J_f}|} = p(z_{t-1,1}, z_{t-1,2}) \times \frac{\prod_i p(\epsilon_{t,i})}{|\mathbf{J_f}|}. \tag{A61}
$$

Finally, we generalize this example and derive the prior likelihood below. Let $\{r_i\}_{i=1,2,3,\cdots}$ be a set of learned inverse transition functions that take the estimated latent causal variables, and output the noise terms, i.e., $\hat{\epsilon}_{t,i} = r_i(\hat{z}_{t,i}, \{\hat{\mathbf{z}}_{t-\tau}\})$. Then we design a transformation $\mathbf{A} \to \mathbf{B}$ with low-triangular Jacobian as follows:

$$
\underbrace{[\hat{\mathbf{z}}_{t-L}, \cdots, \hat{\mathbf{z}}_{t-1}, \hat{\mathbf{z}}_t]^\top}_{\mathbf{A}} \text{ mapped to } \underbrace{[\hat{\mathbf{z}}_{t-L}, \cdots, \hat{\mathbf{z}}_{t-1}, \hat{\epsilon}_{t,i}]^\top}_{\mathbf{B}}, \text{ with } \mathbf{J_{A \to B}} = \begin{bmatrix} \mathbb{I}_{n_s \times L} & 0 \\ * & \mathrm{diag}\left(\frac{\partial r_{i,j}}{\partial \hat{z}_{t,j}}\right) \end{bmatrix}.
$$
$$\tag{A62}$$

Similar to Equation (A61), we can obtain the joint distribution of the estimated dynamics subspace as:

$$
\log p(\mathbf{A}) = \underbrace{\log p(\hat{\mathbf{z}}_{t-L}, \cdots, \hat{\mathbf{z}}_{t-1}) + \sum_{i=1}^{n_s} \log p(\hat{\epsilon}_{t,i})}_{\text{Because of mutually independent noise assumption}} + \log(|\det(\mathbf{J_{A \to B}})|) \tag{A63}
$$

Finally, we have:

$$\log p(\hat{\mathbf{z}}_t | \{\hat{\mathbf{z}}_{t-\tau}\}_{\tau=1}^L) = \sum_{i=1}^{n_s} p(\hat{\epsilon}_{t,i}) + \sum_{i=1}^{n_s} \log |\frac{\partial r_i}{\partial \hat{z}_{t,i}}| \tag{A64}$$

Since the prior of $p(\hat{\mathbf{z}}_{t+1:T} | \hat{\mathbf{z}}_{1:t}) = \prod_{i=t+1}^T p(\hat{\mathbf{z}}_i | \hat{\mathbf{z}}_{i-1})$ with the assumption of first-order Markov assumption, we can estimate $p(\hat{\mathbf{z}}_{t+1:T} | \hat{\mathbf{z}}_{1:t})$ in a similar way.

### D.1.4 Evident Lower Bound

In this subsection, we show the evident lower bound. We first factorize the conditional distribution according to the Bayes theorem.

$$\begin{aligned} \ln p(\mathbf{x}_{1:T}) &= \ln \frac{p(\mathbf{x}_{1:T}, \mathbf{z}_{1:T})}{p(\mathbf{z}_{1:T} | \mathbf{x}_{1:T})} = \mathbb{E}_{q(\mathbf{z}_{1:T} | \mathbf{x}_{1:t})} \ln \frac{p(\mathbf{x}_{1:T}, \mathbf{z}_{1:T}) q(\mathbf{z}_{1:T} | \mathbf{x}_{1:t})}{p(\mathbf{z}_{1:T} | \mathbf{x}_{1:T}) q(\mathbf{z}_{1:T} | \mathbf{x}_{1:t})} \\ &\geq \underbrace{\mathbb{E}_{q(\mathbf{z}_{1:T} | \mathbf{x}_{1:t})} \ln p(\mathbf{x}_{1:T} | \mathbf{z}_{1:T})}_{L_r \text{ and } L_y} - \underbrace{D_{KL}(q(\mathbf{z}_{1:T} | \mathbf{x}_{1:t}) || p(\mathbf{z}_{1:T}))}_{L_{KL}^{\mathbf{z}}} = ELBO. \end{aligned} \tag{A65}$$

### D.1.5 More Synthetic Experiment Results

We repeat each experiment with different random seeds. We further consider CariNG as baselines, experiment results are shown in Table A6.

Table A6: MCC results of synthetic datasets.

|   | TOT | IDOL | CariNG | TDRL |
|---|---|---|---|---|
| A | 0.9258(0.0034) | 0.3788(0.0245) | 0.7354(0.0346) | 0.3572(0.0523) |
| B | 0.9324(0.0078) | 0.8593(0.0092) | 0.0823(0.0092) | 0.8073(0.0786) |
| C | 0.9322(0.0052) | 0.6073(0.0952) | 0.7084(0.0361) | 0.7134(0.0346) |
| D | 0.8433(0.0140) | 0.7800(0.0387) | 0.7371(0.0804) | 0.7747(0.0690) |

## D.2 Real-world Experiment

### D.2.1 Dataset Description

- **ETT** Zhou et al. [2021] is an electricity transformer temperature dataset collected from two separated counties in China, which contains two separate datasets {ETTh2, ETTm1} for one hour level.

- **Exchange** Lai et al. [2018] is the daily exchange rate dataset from of eight foreign countries including Australia, British, Canada, Switzerland, China, Japan, New Zealand, and Singapore ranging from 1990 to.

- **ECL** [4] is an electricity consuming load dataset with the electricity consumption (kWh) collected from 321 clients.

- **Traffic** [5] is a dataset of traffic speeds collected from the California Transportation Agencies (CalTrans) Performance Measurement System (PeMS), which contains data collected from 325 sensors located throughout the Bay Area.

- **Weather** [6] provides 10-minute summaries from an automated rooftop station at the Max Planck Institute for Biogeochemistry in Jena, Germany.

### D.2.2 Implementation Details

The implementations of our method based on different backbones are shown in Table A7 to A11.

---

[4] https://archive.ics.uci.edu/dataset/321/electricityloaddiagrams20112014
[5] https://pems.dot.ca.gov/
[6] https://www.bgc-jena.mpg.de/wetter/

Table A7: LSTD+ToT Architecture details. $T$, length of time series. $|\mathbf{x}_t|$: input dimension. $n$: latent dimension. LeakyReLU: Leaky Rectified Linear Unit. ReLU: Rectified Linear Unit. Tanh: Hyperbolic tangent function.

| Configuration | Description | Output |
|---|---|---|
| $\phi$ | Latent Variable Encoder | |
| Input:$\mathbf{x}_{1:t}$ | Observed time series | Batch Size$\times$t$\times$ $\mathbf{x}$ dimension |
| Dense | Conv1d | Batch Size$\times$640$\times|\mathbf{x}_t|$ |
| Dense | t neurons,LeakyReLu | Batch Size$\times$t$\times|\mathbf{x}_t|$ |
| Dense | T neurons,LeakyReLu | Batch Size$\times$T$\times|\mathbf{x}_t|$ |
| Dense | 512 neurons,LeakyReLu | Batch Size$\times$512$\times|\mathbf{x}_t|$ |
| Dense | t neurons,LeakyReLu | Batch Size$\times$t$\times|\mathbf{x}_t|$ |
| Dense | T neurons,LeakyReLu | Batch Size$\times$T$\times|\mathbf{x}_t|$ |
| $\psi$ | Latent Variable Decoder | |
| Input:$\mathbf{z}_{1:t}$ | Latent Variable | Batch Size$\times$t$\times$2$|\mathbf{x}_t|$ |
| Dense | $|\mathbf{x}_t|$ neurons,LeakyReLu | Batch Size$\times$t$\times|\mathbf{x}_t|$ |
| $\eta$ | Dimensionality reduction | |
| Input:$\mathbf{x}_{1:t}$ | Observed time series | Batch Size$\times$t$\times|\mathbf{x}_t|$ |
| Linear | t neurons,ReLu | Batch Size$\times$t$\times|\mathbf{x}_t|$ |
| $\varphi$ | Regressor | |
| Input:$[\mathbf{z}_{1:T};\eta(\mathbf{x}_{1:t})]$ | Latent Variable | Batch Size$\times$T$\times$2$|\mathbf{x}_t|$ |
| Dense | 512 neurons,LeakyReLU | Batch Size$\times$T$\times$512 |
| Dense | $|\mathbf{x}_t|$ neurons,LeakyReLu | Batch Size$\times$T$\times|\mathbf{x}_t|$ |
| $r_i^{\mathbf{z}}$ | Latent Transition Estimator | |
| Input: $\mathbf{z}_{1:T}$ | Latent Variable | Batch Size$\times$(n+1) |
| Dense | 128 neurons,LeakyReLU | (n+1)$\times$128 |
| Dense | 128 neurons,LeakyReLU | 128$\times$128 |
| Dense | 128 neurons,LeakyReLU | 128$\times$128 |
| Dense | 1 neuron | Batch Size$\times$1 |
| Jacobian Compute | Compute log(det(J)) | Batch Size |
| $r_i^{\mathbf{o}}$ | Observed Transition Estimator | |
| Input: $\mathbf{z}_{1:T}$ | Latent Variable | Batch Size$\times$(n+1) |
| Dense | 128 neurons,LeakyReLU | (n+1)$\times$128 |
| Dense | 128 neurons,LeakyReLU | 128$\times$128 |
| Dense | 128 neurons,LeakyReLU | 128$\times$128 |
| Dense | 1 neuron | Batch Size$\times$1 |
| Jacobian Compute | Compute log(det(J)) | Batch Size |

Table A8: OneNet+ToT Architecture details. $T$, length of time series. $|\mathbf{x}_t|$: input dimension. $n$: latent dimension. LeakyReLU: Leaky Rectified Linear Unit. ReLU: Rectified Linear Unit. Tanh: Hyperbolic tangent function.

| Configuration | Description | Output |
|---|---|---|
| $\phi$ | Latent Variable Encoder | |
| Input:$\mathbf{x}_{1:t}$ | Observed time series | Batch Size$\times$t$\times$ $\mathbf{x}$ dimension |
| Linear | n neurons | Batch Size$\times$n$\times|\mathbf{x}_t|$ |
| Convolution neural networks | 320 neurons | Batch Size$\times$320$\times|\mathbf{x}_t|$ |
| $\psi$ | Latent Variable Decoder | |
| Input:$\mathbf{z}_{1:t}$ | Latent Variable | Batch Size$\times$320$\times$ $|\mathbf{x}_t|$ |
| Linear | t neurons,ReLu | Batch Size$\times$t$\times|\mathbf{x}_t|$ |
| $\eta$ | Dimensionality reduction | |
| Input:$\mathbf{x}_{1:t}$ | Observed time series | Batch Size$\times$t$\times|\mathbf{x}_t|$ |
| Linear | 320 neurons,ReLu | Batch Size$\times$320$\times|\mathbf{x}_t|$ |
| $\varphi$ | Regressor | |
| Input:$[\mathbf{z}_{1:T};\eta(\mathbf{x}_{1:t})]$ | Latent Variable | Batch Size$\times$640$\times|\mathbf{x}_t|$ |
| Linear | $|\mathbf{x}_t|$ neurons,ReLU | Batch Size$\times$T$\times|\mathbf{x}_t|$ |
| Linear | n neurons,ReLu | Batch Size$\times$n$\times|\mathbf{x}_t|$ |
| Convolution neural networks | 320 neurons | Batch Size$\times$320$\times|\mathbf{x}_t|$ |
| Linear | T neurons,ReLu | Batch Size$\times$T$\times|\mathbf{x}_t|$ |
| $r_i^{\mathbf{z}}$ | Latent Transition Estimator | |
| Input: $\mathbf{z}_{1:T}$ | Latent Variable | Batch Size$\times$(n+1) |
| Dense | 128 neurons,LeakyReLU | (n+1)$\times$128 |
| Dense | 128 neurons,LeakyReLU | 128$\times$128 |
| Dense | 128 neurons,LeakyReLU | 128$\times$128 |
| Dense | 1 neuron | Batch Size$\times$1 |
| Jacobian Compute | Compute log(det(J)) | Batch Size |
| $r_i^{\mathbf{o}}$ | Observed Transition Estimator | |
| Input: $\mathbf{z}_{1:T}$ | Latent Variable | Batch Size$\times$(n+1) |
| Dense | 128 neurons,LeakyReLU | (n+1)$\times$128 |
| Dense | 128 neurons,LeakyReLU | 128$\times$128 |
| Dense | 128 neurons,LeakyReLU | 128$\times$128 |
| Dense | 1 neuron | Batch Size$\times$1 |
| Jacobian Compute | Compute log(det(J)) | Batch Size |

Table A9: OneNet-T+TOT Architecture details. $T$, length of time series. $|\mathbf{x}_t|$: input dimension. $n$: latent dimension. LeakyReLU: Leaky Rectified Linear Unit. ReLU: Rectified Linear Unit. Tanh: Hyperbolic tangent function.

| Configuration | Description | Output |
|---|---|---|
| $\phi$ | Latent Variable Encoder | |
| Input:$\mathbf{x}_{1:t}$ | Observed time series | Batch Size$\times$t$\times$ $\mathbf{x}$ dimension |
| Linear | n neurons,ReLu | Batch Size$\times$n$\times|\mathbf{x}_t|$ |
| Dilation convolution | T neurons,10 layers | Batch Size$\times$T$\times|\mathbf{x}_t|$ |
| $\psi$ | Latent Variable Decoder | |
| Input:$\mathbf{z}_{1:t}$ | Latent Variable | Batch Size$\times$320$\times$ $|\mathbf{x}_t|$ |
| Linear | t neurons,ReLu | Batch Size$\times$t$\times|\mathbf{x}_t|$ |
| $\eta$ | Dimensionality reduction | |
| Input:$\mathbf{x}_{1:t}$ | Observed time series | Batch Size$\times$t$\times|\mathbf{x}_t|$ |
| Linear | n neurons,ReLu | Batch Size$\times$n$\times|\mathbf{x}_t|$ |
| Dilation convolution | T neurons,10 layers | Batch Size$\times$T$\times|\mathbf{x}_t|$ |
| $\varphi$ | Regressor | |
| Input:$[\mathbf{z}_{1:T};\eta(\mathbf{x}_{1:t})]$ | Latent Variable | Batch Size$\times$T$\times$2$|\mathbf{x}_t|$ |
| Linear | $|\mathbf{x}_t|$ neurons,ReLU | Batch Size$\times$T$\times|\mathbf{x}_t|$ |
| Linear | n neurons,ReLU | Batch Size$\times$n$\times|\mathbf{x}_t|$ |
| Convolution neural networks | 320 neurons | Batch Size$\times$320$\times|\mathbf{x}_t|$ |
| Linear | T neurons,ReLu | Batch Size$\times$T$\times|\mathbf{x}_t|$ |
| $r_i^{\mathbf{z}}$ | Latent Transition Estimator | |
| Input: $\mathbf{z}_{1:T}$ | Latent Variable | Batch Size$\times$(n+1) |
| Dense | 128 neurons,LeakyReLU | (n+1)$\times$128 |
| Dense | 128 neurons,LeakyReLU | 128$\times$128 |
| Dense | 128 neurons,LeakyReLU | 128$\times$128 |
| Dense | 1 neuron | Batch Size$\times$1 |
| Jacobian Compute | Compute log(det(J)) | Batch Size |
| $r_i^{\mathbf{o}}$ | Observed Transition Estimator | |
| Input: $\mathbf{z}_{1:T}$ | Latent Variable | Batch Size$\times$(n+1) |
| Dense | 128 neurons,LeakyReLU | (n+1)$\times$128 |
| Dense | 128 neurons,LeakyReLU | 128$\times$128 |
| Dense | 128 neurons,LeakyReLU | 128$\times$128 |
| Dense | 1 neuron | Batch Size$\times$1 |
| Jacobian Compute | Compute log(det(J)) | Batch Size |

Table A10: online-T+TOT Architecture details. $T$, length of time series. $|\mathbf{x}_t|$: input dimension. $n$: latent dimension. LeakyReLU: Leaky Rectified Linear Unit. ReLU: Rectified Linear Unit. Tanh: Hyperbolic tangent function.

| Configuration | Description | Output |
|---|---|---|
| $\phi$ | Latent Variable Encoder | |
| Input:$\mathbf{x}_{1:t}$ | Observed time series | Batch Size$\times$t$\times$ $\mathbf{x}$ dimension |
| Linear | n neurons,ReLu | Batch Size$\times$n$\times|\mathbf{x}_t|$ |
| Convolution neural networks | 320 neurons | Batch Size$\times$320$\times|\mathbf{x}_t|$ |
| Linear | t neurons,ReLu | Batch Size$\times$t$\times|\mathbf{x}_t|$ |
| $\psi$ | Latent Variable Decoder | |
| Input:$\mathbf{z}_{1:t}$ | Latent Variable | Batch Size$\times$320$\times$ $|\mathbf{x}_t|$ |
| Linear | t neurons,ReLu | Batch Size$\times$t$\times|\mathbf{x}_t|$ |
| $\eta$ | Dimensionality reduction | |
| Input:$\mathbf{x}_{1:t}$ | Observed time series | Batch Size$\times$t$\times|\mathbf{x}_t|$ |
| Linear | t neurons,ReLu | Batch Size$\times$t$\times|\mathbf{x}_t|$ |
| $\varphi$ | Regressor | |
| Input:$[\mathbf{z}_{1:T};\eta(\mathbf{x}_{1:t})]$ | Latent Variable | Batch Size$\times$2t$\times|\mathbf{x}_t|$ |
| Moving average | kernel size,stride=1 | Batch Size$\times$2t$\times|\mathbf{x}_t|$ |
| Dilation convolution | 320 neurons,5 layers,ReLu | Batch Size$\times$320$\times|\mathbf{x}_t|$ |
| Padding | patch length=6,ReLu | Batch Size$\times|\mathbf{x}_t|\times$patch length$\times$patch num |
| Transformer | n neurons | Batch Size$\times|\mathbf{x}_t|\times$n$\times$patch num |
| Linear | T neurons,ReLu | Batch Size$\times$T$\times|\mathbf{x}_t|$ |
| $r_i^{\mathbf{z}}$ | Latent Transition Estimator | |
| Input: $\mathbf{z}_{1:T}$ | Latent Variable | Batch Size$\times$(n+1) |
| Dense | 128 neurons,LeakyReLU | (n+1)$\times$128 |
| Dense | 128 neurons,LeakyReLU | 128$\times$128 |
| Dense | 128 neurons,LeakyReLU | 128$\times$128 |
| Dense | 1 neuron | Batch Size$\times$1 |
| Jacobian Compute | Compute log(det(J)) | Batch Size |
| $r_i^{\mathbf{o}}$ | Observed Transition Estimator | |
| Input: $\mathbf{z}_{1:T}$ | Latent Variable | Batch Size$\times$(n+1) |
| Dense | 128 neurons,LeakyReLU | (n+1)$\times$128 |
| Dense | 128 neurons,LeakyReLU | 128$\times$128 |
| Dense | 128 neurons,LeakyReLU | 128$\times$128 |
| Dense | 1 neuron | Batch Size$\times$1 |
| Jacobian Compute | Compute log(det(J)) | Batch Size |

Table A11: Proceed-T+TOT Architecture details. $T$, length of time series. $|\mathbf{x}_t|$: input dimension. $n$: latent dimension. LeakyReLU: Leaky Rectified Linear Unit. ReLU: Rectified Linear Unit. Tanh: Hyperbolic tangent function.

| Configuration | Description | Output |
|---|---|---|
| $\phi$ | Latent Variable Encoder | |
| Input:$\mathbf{x}_{1:t}$ | Observed time series | Batch Size$\times$t$\times$ x dimension |
| Linear | n neurons,ReLu | Batch Size$\times$n$\times|\mathbf{x}_t|$ |
| Dilation convolution | T neurons,10 layers | Batch Size$\times$T$\times|\mathbf{x}_t|$ |
| $\psi$ | Latent Variable Decoder | |
| Input:$\mathbf{z}_{1:t}$ | Latent Variable | Batch Size$\times$320$\times$ $|\mathbf{x}_t|$ |
| Linear | t neurons,ReLu | Batch Size$\times$t$\times|\mathbf{x}_t|$ |
| $\eta$ | Dimensionality reduction | |
| Input:$\mathbf{x}_{1:t}$ | Observed time series | Batch Size$\times$t$\times|\mathbf{x}_t|$ |
| Linear | n neurons,ReLu | Batch Size$\times$n$\times|\mathbf{x}_t|$ |
| Dilation convolution | T neurons,10 layers | Batch Size$\times$T$\times|\mathbf{x}_t|$ |
| $\varphi$ | Regressor | |
| Input:$[\mathbf{z}_{1:T};\eta(\mathbf{x}_{1:t})]$ | Latent Variable | Batch Size$\times$T$\times2|\mathbf{x}_t|$ |
| Linear | $|\mathbf{x}_t|$ neurons,ReLu | Batch Size$\times$T$\times|\mathbf{x}_t|$ |
| $r_i^{\mathbf{z}}$ | Latent Transition Estimator | |
| Input: $\mathbf{z}_{1:T}$ | Latent Variable | Batch Size$\times$(n+1) |
| Dense | 128 neurons,LeakyReLU | (n+1)$\times$128 |
| Dense | 128 neurons,LeakyReLU | 128$\times$128 |
| Dense | 128 neurons,LeakyReLU | 128$\times$128 |
| Dense | 1 neuron | Batch Size$\times$1 |
| Jacobian Compute | Compute log(det(J)) | Batch Size |
| $r_i^{\mathbf{o}}$ | Observed Transition Estimator | |
| Input: $\mathbf{z}_{1:T}$ | Latent Variable | Batch Size$\times$(n+1) |
| Dense | 128 neurons,LeakyReLU | (n+1)$\times$128 |
| Dense | 128 neurons,LeakyReLU | 128$\times$128 |
| Dense | 128 neurons,LeakyReLU | 128$\times$128 |
| Dense | 1 neuron | Batch Size$\times$1 |
| Jacobian Compute | Compute log(det(J)) | Batch Size |

### D.2.3 More Experiment Results

We further consider MIR Aljundi et al. [2019a] and TFCL Aljundi et al. [2019b] as the backbone networks, experimental results are shown in Table A12.

Table A12: MSE and MAE results of different datasets on TFCL and MIR backbone.

| Models | Len | TFCL | | TFCL+TOT | | MIR | | MIR+TOT | | Models | Len | TFCL | | TFCL+TOT | | MIR | | MIR+TOT | |
|---|---|---|---|---|---|---|---|---|---|---|---|---|---|---|---|---|---|---|---|
| | | MSE | MAE | MSE | MAE | MSE | MAE | MSE | MAE | | | MSE | MAE | MSE | MAE | MSE | MAE | MSE | MAE |
| **ETTh2** | 1 | 0.557 | 0.472 | **0.463** | **0.382** | 0.486 | 0.41 | **0.447** | **0.378** | **ECL** | 1 | **2.732** | 0.524 | 3.815 | **0.44** | **2.575** | **0.504** | 3.396 | 0.589 |
| | 24 | 0.846 | **0.548** | **0.825** | 0.554 | 0.812 | 0.541 | **0.652** | **0.465** | | 24 | 12.094 | 1.256 | **10.083** | **1.105** | 9.265 | 1.066 | **6.142** | **1.041** |
| | 48 | 1.208 | 0.592 | **0.87** | **0.555** | 1.103 | 0.565 | **0.842** | **0.526** | | 48 | 12.11 | 1.303 | **10.685** | **1.075** | 9.411 | **1.079** | **6.479** | 1.090 |
| **ETTm1** | 1 | 0.087 | 0.198 | **0.081** | **0.187** | 0.085 | 0.197 | **0.083** | **0.188** | **Traffic** | 1 | 0.306 | 0.297 | **0.304** | **0.263** | 0.298 | 0.284 | **0.294** | **0.267** |
| | 24 | 0.211 | 0.341 | **0.186** | **0.32** | 0.192 | 0.325 | **0.132** | **0.267** | | 24 | 0.441 | 0.493 | **0.389** | **0.314** | 0.451 | 0.443 | **0.39** | **0.339** |
| | 48 | 0.236 | 0.363 | **0.196** | **0.331** | 0.210 | 0.342 | **0.129** | **0.265** | | 48 | 0.438 | 0.531 | **0.393** | **0.316** | 0.502 | 0.397 | **0.419** | **0.345** |
| **WTH** | 1 | 0.177 | 0.24 | **0.154** | **0.197** | 0.179 | 0.244 | **0.154** | **0.199** | **Exchange** | 1 | 0.106 | 0.153 | **0.045** | **0.142** | 0.095 | 0.118 | **0.056** | 0.162 |
| | 24 | 0.301 | 0.363 | **0.295** | **0.359** | 0.291 | 0.355 | **0.184** | **0.265** | | 24 | 0.098 | 0.227 | **0.062** | **0.166** | 0.104 | 0.204 | **0.067** | **0.178** |
| | 48 | 0.323 | 0.382 | **0.294** | **0.36** | 0.297 | 0.361 | **0.195** | **0.278** | | 48 | 0.101 | 0.183 | **0.098** | **0.207** | 0.101 | 0.209 | **0.047** | **0.137** |

To demonstrate that the improvements of our approach are not due to an increase in parameters, we increase the number of parameters of the baseline methods by adding additional layers to the neural

Table A13: Evaluation on Same Size of Model

| Model | Len | LSTD(Original) | | LSTD(Same size) | | LSTD+TOT | |
|---|---|---|---|---|---|---|---|
| Matric | | MSE | MAE | MSE | MAE | MSE | MAE |
| ETTh2 | 1 | 0.377 | 0.347 | 0.378 | 0.349 | **0.374** | **0.346** |
| | 24 | 0.543 | 0.411 | 0.778 | 0.465 | **0.532** | **0.390** |
| | 48 | 0.616 | 0.423 | 0.620 | 0.443 | **0.564** | **0.420** |
| WTH | 1 | **0.153** | **0.200** | 0.155 | 0.202 | **0.153** | **0.200** |
| | 24 | 0.136 | 0.223 | 0.126 | 0.215 | **0.116** | **0.207** |
| | 48 | 0.157 | 0.242 | 0.168 | 0.251 | **0.152** | **0.239** |
| ECL | 1 | 2.112 | 0.226 | 2.109 | 0.234 | **2.038** | **0.221** |
| | 24 | 1.422 | 0.292 | 1.420 | 0.285 | **1.390** | **0.278** |
| | 48 | **1.411** | 0.294 | 1.442 | 0.303 | 1.413 | **0.289** |
| Traffic | 1 | 0.231 | 0.225 | 0.231 | 0.225 | **0.229** | **0.224** |
| | 24 | 0.398 | 0.316 | 0.402 | 0.319 | **0.397** | **0.313** |
| | 48 | 0.426 | 0.332 | 0.427 | 0.333 | **0.421** | **0.328** |
| Exchange | 1 | **0.013** | 0.070 | 0.013 | 0.071 | **0.013** | **0.069** |
| | 24 | 0.039 | 0.132 | 0.040 | 0.135 | **0.037** | **0.129** |
| | 48 | 0.043 | 0.142 | 0.043 | 0.142 | **0.042** | **0.142** |

network, making the number of parameters of our method and the baseline methods comparable. Experiment results are shown in Table A13. According to the experiment results, we can find that our method still achieves the general improvement.

### D.2.4 Experiment Results of Mean and Standard Deviation

The mean and standard deviation of MAE and MSE are shown in Table A14, A15, A16, and A17, respectively.

Table A14: Mean values of MSE on different datasets.

| Models | Len | LSTD | LSTD+ToT | Proceed-T | Proceed-T+ToT | OneNet | OneNet+TOT | OneNet-T | OneNet-T+TOT | MIR | MIR+TOT | Online-T | Online-T+TOT | TFCL | TFCL+TOT |
|---|---|---|---|---|---|---|---|---|---|---|---|---|---|---|---|
| ETTh2 | 1 | 0.375 | **0.374** | 1.537 | **1.186** | 0.377 | **0.365** | 0.394 | **0.391** | 0.524 | **0.451** | 0.617 | **0.444** | 0.531 | **0.466** |
| | 24 | 0.543 | **0.540** | 2.908 | **2.444** | 0.548 | **0.515** | 0.943 | **0.697** | 1.098 | **0.587** | 0.832 | **0.757** | 0.851 | **0.830** |
| | 48 | 0.616 | **0.616** | 4.056 | **4.013** | 0.622 | **0.574** | 0.926 | **0.783** | 1.098 | **0.740** | 1.188 | **0.977** | 1.211 | **0.891** |
| ETTm1 | 1 | 0.082 | **0.081** | 0.106 | **0.105** | 0.086 | **0.082** | 0.091 | **0.079** | 0.082 | **0.085** | 0.208 | **0.077** | 0.085 | **0.082** |
| | 24 | **0.102** | 0.108 | 0.531 | **0.516** | 0.105 | **0.097** | 0.213 | **0.174** | 0.189 | **0.119** | 0.263 | **0.224** | 0.216 | **0.195** |
| | 48 | **0.115** | 0.118 | 0.704 | **0.703** | 0.110 | **0.102** | 0.216 | **0.188** | 0.223 | **0.138** | 0.271 | **0.255** | 0.240 | **0.203** |
| WTH | 1 | 0.155 | **0.153** | 0.346 | **0.336** | 0.157 | **0.151** | 0.157 | **0.161** | 0.182 | **0.152** | 0.213 | **0.145** | 0.176 | **0.160** |
| | 24 | 0.139 | **0.136** | 0.707 | **0.697** | 0.173 | **0.158** | 0.276 | **0.264** | 0.286 | **0.166** | 0.312 | **0.272** | 0.311 | **0.295** |
| | 48 | 0.167 | **0.164** | 0.959 | **0.956** | 0.196 | **0.175** | 0.289 | **0.273** | 0.289 | **0.169** | 0.298 | **0.279** | 0.324 | **0.297** |
| ECL | 1 | 2.228 | **2.116** | 3.27 | **3.156** | 2.675 | **2.330** | 2.413 | **2.278** | 2.568 | 3.513 | 3.312 | **2.258** | 2.806 | 3.781 |
| | 24 | 1.557 | **1.514** | 5.907 | **5.895** | 2.090 | **2.035** | 4.551 | **4.580** | 9.157 | **6.095** | 11.594 | **4.463** | 11.891 | **10.932** |
| | 48 | 1.720 | **1.654** | 7.192 | 7.500 | 2.438 | **2.198** | 4.488 | **4.472** | 9.391 | **8.209** | 11.912 | **4.548** | 12.109 | **10.235** |
| Traffic | 1 | 0.234 | **0.231** | 0.333 | **0.326** | 0.241 | **0.229** | 0.236 | **0.222** | 0.298 | **0.296** | 0.334 | **0.211** | 0.306 | **0.304** |
| | 24 | 0.417 | **0.401** | 0.413 | **0.412** | 0.438 | **0.419** | 0.425 | **0.413** | 0.451 | **0.435** | 0.481 | **0.411** | 0.441 | **0.366** |
| | 48 | 0.431 | **0.422** | 0.454 | **0.452** | 0.473 | **0.439** | 0.451 | **0.439** | 0.502 | **0.458** | 0.503 | **0.425** | 0.438 | **0.391** |
| Exchange | 1 | 0.014 | **0.013** | 0.012 | **0.009** | 0.017 | **0.016** | 0.031 | **0.018** | 0.095 | **0.057** | 0.113 | **0.010** | 0.106 | **0.054** |
| | 24 | 0.039 | **0.036** | 0.129 | **0.105** | 0.047 | **0.041** | 0.060 | **0.041** | 0.104 | **0.077** | 0.116 | **0.026** | 0.098 | **0.081** |
| | 48 | 0.049 | **0.046** | 0.267 | **0.200** | 0.062 | **0.056** | 0.065 | **0.056** | 0.101 | **0.085** | 0.168 | **0.029** | 0.101 | **0.099** |

Table A15: Mean values of MAE on different datasets.

| Models | Len | LSTD | LSTD+ToT | Proceed-T | Proceed-T+ToT | OneNet | OneNet+TOT | OneNet-T | OneNet-T+TOT | MIR | MIR+TOT | Online-T | Online-T+TOT | TFCL | TFCL+TOT |
|---|---|---|---|---|---|---|---|---|---|---|---|---|---|---|---|
| ETTh2 | 1 | 0.347 | **0.347** | 0.447 | **0.401** | 0.354 | **0.347** | 0.373 | **0.364** | 0.418 | **0.373** | 0.443 | **0.352** | 0.466 | **0.381** |
| | 24 | 0.411 | **0.394** | 0.659 | **0.619** | 0.415 | **0.406** | 0.532 | **0.510** | 0.543 | **0.439** | 0.545 | **0.497** | 0.539 | **0.569** |
| | 48 | 0.423 | **0.437** | 0.767 | **0.732** | 0.448 | **0.435** | 0.535 | **0.520** | 0.572 | **0.489** | 0.598 | **0.549** | 0.591 | **0.585** |
| ETTm1 | 1 | 0.189 | **0.187** | 0.19 | **0.187** | 0.192 | **0.186** | 0.207 | **0.186** | 0.201 | **0.190** | 0.218 | **0.181** | 0.192 | **0.191** |
| | 24 | **0.217** | 0.240 | 0.447 | **0.442** | 0.234 | **0.226** | 0.343 | **0.306** | 0.327 | **0.252** | 0.376 | **0.348** | 0.346 | **0.329** |
| | 48 | 0.259 | **0.251** | 0.521 | **0.507** | 0.242 | **0.232** | 0.348 | **0.322** | 0.347 | **0.273** | 0.415 | **0.376** | 0.357 | **0.339** |
| WTH | 1 | 0.200 | **0.200** | 0.143 | **0.140** | 0.202 | **0.193** | 0.206 | 0.212 | 0.182 | **0.200** | 0.210 | **0.189** | 0.169 | 0.204 |
| | 24 | 0.224 | **0.221** | 0.382 | **0.375** | 0.255 | **0.239** | 0.337 | **0.334** | 0.317 | **0.250** | 0.317 | 0.332 | **0.314** | 0.360 |
| | 48 | 0.250 | **0.247** | 0.493 | **0.493** | 0.277 | **0.255** | 0.354 | **0.345** | 0.289 | **0.255** | 0.334 | 0.339 | **0.325** | 0.368 |
| ECL | 1 | 0.232 | **0.230** | 0.286 | **0.282** | 0.268 | **0.254** | 0.280 | 0.337 | **0.519** | 0.580 | 0.641 | **0.214** | **0.273** | 0.436 |
| | 24 | 0.288 | **0.282** | 0.387 | **0.384** | 0.341 | **0.340** | 0.405 | **0.393** | **1.035** | 1.036 | 1.291 | **0.330** | 1.194 | **1.098** |
| | 48 | 0.348 | **0.302** | 0.431 | **0.425** | 0.367 | **0.360** | 0.423 | **0.405** | 1.184 | **0.992** | 1.219 | **0.344** | 1.304 | **1.079** |
| Traffic | 1 | 0.229 | **0.227** | 0.268 | **0.263** | 0.240 | **0.228** | 0.236 | **0.212** | 0.284 | **0.269** | 0.284 | **0.202** | 0.297 | **0.264** |
| | 24 | 0.332 | **0.315** | 0.291 | **0.285** | 0.346 | **0.338** | 0.346 | **0.318** | 0.443 | **0.359** | 0.385 | **0.311** | 0.493 | **0.310** |
| | 48 | 0.344 | **0.329** | 0.308 | 0.311 | 0.371 | **0.338** | 0.355 | **0.340** | 0.397 | **0.381** | 0.380 | **0.325** | 0.531 | **0.313** |
| Exchange | 1 | 0.070 | **0.069** | 0.063 | **0.051** | 0.085 | **0.081** | 0.117 | **0.087** | 0.118 | **0.144** | 0.169 | **0.057** | 0.153 | **0.147** |
| | 24 | 0.132 | **0.128** | 0.211 | **0.191** | 0.148 | **0.135** | 0.166 | **0.137** | 0.204 | **0.188** | 0.213 | **0.106** | 0.227 | **0.191** |
| | 48 | 0.150 | **0.147** | 0.3 | **0.263** | 0.170 | **0.157** | 0.173 | **0.161** | 0.209 | **0.196** | 0.258 | **0.113** | 0.183 | 0.208 |

Table A16: Standard deviation of MSE on different datasets.

| Models | Len | LSTD | LSTD+ToT | Proceed-T | Proceed-T+ToT | OneNet | OneNet+TOT | OneNet-T | OneNet-T+TOT | MIR | MIR+TOT | Online-T | Online-T+TOT | TFCL | TFCL+TOT |
|---|---|---|---|---|---|---|---|---|---|---|---|---|---|---|---|
| ETTh2 | 1 | 0.0246 | 0.0031 | 0.1038 | 0.1351 | 0.0085 | 0.0041 | 0.0076 | 0.0233 | 0.1229 | 0.0168 | 0.0149 | 0.0535 | 0.1827 | 0.0049 |
| | 24 | 0.0260 | 0.0078 | 0.0413 | 0.0620 | 0.0137 | 0.0128 | 0.1563 | 0.0087 | 0.1618 | 0.0309 | 0.0098 | 0.0341 | 0.0699 | 0.0070 |
| | 48 | 0.0295 | 0.0494 | 0.1030 | 0.0414 | 0.0263 | 0.0145 | 0.0829 | 0.0137 | 0.0827 | 0.0561 | 0.0172 | 0.1233 | 0.1320 | 0.0206 |
| ETTm1 | 1 | 0.00097 | 0.0004 | 0.0008 | 0.0006 | 0.0025 | 0.0007 | 0.0085 | 0.0023 | 0.1382 | 0.0019 | 0.0093 | 0.0007 | 0.1259 | 0.0018 |
| | 24 | 0.0003 | 0.0014) | 0.0197 | 0.0129 | 0.0018 | 0.0014 | 0.0025 | 0.0341 | 0.1107 | 0.0142 | 0.0080 | 0.0051 | 0.1643 | 0.0079 |
| | 48 | 0.0157 | 0.0012 | 0.0108 | 0.0049 | 0.0012 | 0.0044 | 0.0068 | 0.0093 | 0.0370 | 0.0173 | 0.0142 | 0.0151 | 0.0764 | 0.0063 |
| WTH | 1 | 0.00084 | 0.0003 | 0.0043 | 0.0040 | 0.0006 | 0.0004 | 0.0004 | 0.0157 | 0.1747 | 0.0006 | 0.0162 | 0.0001 | 0.1241 | 0.0057 |
| | 24 | 0.0023 | 0.0024 | 0.0025 | 0.0043 | 0.0026 | 0.0020 | 0.0023 | 0.0007 | 0.1212 | 0.0031 | 0.0167 | 0.0036 | 0.1782 | 0.0028 |
| | 48 | 0.0055 | 0.0044 | 0.0081 | 0.0050 | 0.0041 | 0.0191 | 0.0100 | 0.0112 | 0.1338 | 0.0035 | 0.0108 | 0.0035 | 0.1790 | 0.0035 |
| ECL | 1 | 0.0197 | 0.0189 | 0.0360 | 0.0673 | 0.0449 | 0.0415 | 0.0344 | 0.0731 | 0.0119 | 0.1042 | 0.0152 | 0.0582 | 0.0816 | 0.0484 |
| | 24 | 0.0256 | 0.0662 | 0.0232 | 0.0612 | 0.0205 | 0.0120 | 0.0326 | 0.1260 | 0.0051 | 0.0907 | 0.0178 | 0.0669 | 0.0104 | 0.7811 |
| | 48 | 0.2065 | 0.0278 | 0.2163 | 0.0867 | 0.1002 | 0.0908 | 0.0152 | 0.0246 | 0.0126 | 0.2878 | 0.0092 | 0.0799 | 0.0081 | 0.5162 |
| Traffic | 1 | 0.0027 | 0.0008 | 0.0186 | 0.0079 | 0.0010 | 0.0017 | 0.0015 | 0.0097 | 0.0149 | 0.0014 | 0.0462 | 0.0125 | 0.0052 | 0.0002 |
| | 24 | 0.0070 | 0.0065 | 0.0135 | 0.0034 | 0.0068 | 0.0654 | 0.0046 | 0.0186 | 0.0148 | 0.0064 | 0.0134 | 0.0271 | 0.0103 | 0.0045 |
| | 48 | 0.0024 | 0.0007 | 0.0063 | 0.0020 | 0.0289 | 0.0415 | 0.0015 | 0.0274 | 0.0159 | 0.0345 | 0.0180 | 0.0021 | 0.0175 | 0.0011 |
| Exchange | 1 | 0.0005 | 0.0002 | 0.0005 | 0.00003 | 0.0011 | 0.0008 | 0.0021 | 0.0017 | 0.0130 | 0.0101 | 0.0086 | 0.0008 | 0.0158 | 0.0084 |
| | 24 | 0.0004 | 0.0030 | 0.0027 | 0.0033 | 0.0064 | 0.0097 | 0.0010 | 0.0027 | 0.0070 | 0.0077 | 0.0137 | 0.0079 | 0.0139 | 0.0076 |
| | 48 | 0.0054 | 0.0023 | 0.0037 | 0.0035 | 0.0170 | 0.0137 | 0.0022 | 0.0062 | 0.0038 | 0.0130 | 0.0067 | 0.0031 | 0.0126 | 0.0017 |

Table A17: Standard deviation of MAE on different datasets.

| Models | Len | LSTD | LSTD+ToT | Proceed-T | Proceed-T+ToT | OneNet | OneNet+TOT | OneNet-T | OneNet-T+TOT | MIR | MIR+TOT | Online-T | Online-T+TOT | TFCL | TFCL+TOT |
|---|---|---|---|---|---|---|---|---|---|---|---|---|---|---|---|
| ETTh2 | 1 | 0.0073 | 0.0004 | 0.0105 | 0.0031 | 0.0053 | 0.0008 | 0.0066 | 0.0120 | 0.0130 | 0.0040 | 0.0175 | 0.0049 | 0.0183 | 0.0013 |
| | 24 | 0.0032 | 0.0035 | 0.0094 | 0.0023 | 0.0061 | 0.0029 | 0.0174 | 0.0024 | 0.0408 | 0.0109 | 0.0066 | 0.0068 | 0.0470 | 0.0005 |
| | 48 | 0.0060 | 0.0226 | 0.0104 | 0.0038 | 0.0099 | 0.0004 | 0.0125 | 0.0019 | 0.0098 | 0.0092 | 0.0115 | 0.0222 | 0.0037 | 0.0065 |
| ETTm1 | 1 | 0.0021 | 0.0008 | 0.0013 | 0.0006 | 0.0045 | 0.0008 | 0.0123 | 0.0044 | 0.0143 | 0.0033 | 0.0175 | 0.0017 | 0.0517 | 0.0046 |
| | 24 | 0.0008 | 0.0026 | 0.0096 | 0.0048 | 0.0022 | 0.0016 | 0.0020 | 0.0317 | 0.0526 | 0.0148 | 0.0048 | 0.0043 | 0.0571 | 0.0074 |
| | 48 | 0.0118 | 0.0021 | 0.0012 | 0.0020 | 0.0012 | 0.0049 | 0.0062 | 0.0087 | 0.0086 | 0.0173 | 0.0126 | 0.0115 | 0.0565 | 0.0067 |
| WTH | 1 | 0.0011 | 0.0006 | 0.0010 | 0.0020 | 0.0005 | 0.0008 | 0.0004 | 0.0222 | 0.0091 | 0.0013 | 0.0078 | 0.0003 | 0.0163 | 0.0045 |
| | 24 | 0.0010 | 0.0017 | 0.0004 | 0.0011 | 0.0027 | 0.0002 | 0.0023 | 0.0009 | 0.0078 | 0.0031 | 0.0157 | 0.0028 | 0.0089 | 0.0048 |
| | 48 | 0.0044 | 0.0039 | 0.0014 | 0.0019 | 0.0039 | 0.0183 | 0.0050 | 0.0060 | 0.0132 | 0.0031 | 0.0063 | 0.0034 | 0.0184 | 0.0075 |
| ECL | 1 | 0.0055 | 0.0010 | 0.0006 | 0.0012 | 0.0013 | 0.0016 | 0.0045 | 0.0391 | 0.0085 | 0.0084 | 0.0013 | 0.0014 | 0.0575 | 0.0044 |
| | 24 | 0.0044 | 0.0065 | 0.0004 | 0.0064 | 0.0025 | 0.0021 | 0.0001 | 0.0130 | 0.0581 | 0.0105 | 0.0152 | 0.0085 | 0.0176 | 0.0768 |
| | 48 | 0.0686 | 0.0079 | 0.0018 | 0.0032 | 0.0047 | 0.0038 | 0.0013 | 0.0027 | 0.0125 | 0.0132 | 0.0179 | 0.0022 | 0.0056 | 0.0226 |
| Traffic | 1 | 0.0028 | 0.0013 | 0.0044 | 0.0053 | 0.0006 | 0.0014 | 0.0014 | 0.0082 | 0.0072 | 0.0020 | 0.0105 | 0.0041 | 0.0125 | 0.0010 |
| | 24 | 0.0042 | 0.0046 | 0.0142 | 0.0105 | 0.0027 | 0.0411 | 0.0027 | 0.0088 | 0.0106 | 0.0077 | 0.0182 | 0.0137 | 0.0150 | 0.0066 |
| | 48 | 0.0021 | 0.0012 | 0.0044 | 0.0019 | 0.0144 | 0.0241 | 0.0013 | 0.0162 | 0.0028 | 0.0188 | 0.0017 | 0.0016 | 0.0165 | 0.0012 |
| Exchange | 1 | 0.0018 | 0.0006 | 0.0022 | 0.0006 | 0.0031 | 0.0024 | 0.0034 | 0.0048 | 0.0090 | 0.0072 | 0.0172 | 0.0034 | 0.0010 | 0.0061 |
| | 24 | 0.0004 | 0.0047 | 0.0018 | 0.0024 | 0.0095 | 0.0154 | 0.0014 | 0.0051 | 0.0139 | 0.0119 | 0.0012 | 0.0152 | 0.0051 | 0.0098 |
| | 48 | 0.0033 | 0.0036 | 0.0034 | 0.0018 | 0.0241 | 0.0210 | 0.0031 | 0.0089 | 0.0014 | 0.0153 | 0.0002 | 0.0054 | 0.0174 | 0.0025 |

# E   Broader Impacts

The proposed method for online time series forecasting presents a novel approach to address the challenges posed by distribution shifts in temporal data. By leveraging the identification of latent variables and their causal transitions, our framework demonstrates a provable reduction in Bayes risk, with significant improvements in forecasting accuracy, making the method highly applicable to real-time forecasting tasks in fields such as finance, healthcare, and energy management.

Our method not only outperforms existing models like IDOL and TDRL in both synthetic and real-world experiments, but it also enhances the scalability and adaptability of forecasting systems in dynamic environments. The theoretical advancements, coupled with the plug-and-play architecture, facilitate seamless integration into existing forecasting pipelines, further promoting the use of causal modeling in practical scenarios.

Furthermore, the broader implications of this work extend beyond time series forecasting. The ability to identify and utilize latent variables in real-time systems opens the door to new applications in domains such as anomaly detection, predictive maintenance, and environmental monitoring. With its potential for improving decision-making processes in critical industries, our method sets a strong foundation for future advancements in online forecasting and causal representation learning, thus contributing to the evolution of machine learning models that can effectively handle complex, dynamic data in real-world environments.

