# OpenReview forum: "Online Time Series Forecasting with Theoretical Guarantees"
_NeurIPS.cc/2025/Conference — NeurIPS 2025 poster_

### Official Review · Reviewer_CFrf · 2025-06-16

**Clarity:** 1
**Significance:** 2
**Originality:** 2
**Rating:** 3
**Confidence:** 3

**Summary:**

This paper proposes a theoretical and practical framework called TOT for online time series forecasting, addressing the challenge of temporal distribution shifts caused by latent variables. It establishes rigorous theoretical guarantees, showing that incorporating latent variables into forecasting significantly reduces predictive risk. The authors further prove identifiability conditions, demonstrating that both latent states and their causal dynamics can be inferred using minimal consecutive observations. Additionally, the paper presents a plug-and-play model architecture involving a temporal decoder and noise estimators. Extensive experiments on synthetic and real-world datasets validate the theoretical insights and show consistent performance improvements over existing methods.

**Questions:**

See summary or Weaknesses.

**Ethical Concerns:**

["NO or VERY MINOR ethics concerns only"]

**Final Justification:**

I have carefully read the author's reply and I have decided to keep my score.

**Limitations:**

See summary or Weaknesses.

**Paper Formatting Concerns:**

The authors are recommended to verify their citation formatting carefully, as it appears inconsistent with standard practice. Additionally, citations in the main text do not link correctly to the corresponding references listed.

**Quality:**

2

**Strengths And Weaknesses:**

1.	The authors are recommended to verify their citation formatting carefully, as it appears inconsistent with standard practice. Additionally, citations in the main text do not link correctly to the corresponding references listed.

2.	The authors need to clarify the rationale for emphasizing "online time-series forecasting" explicitly. If there is no substantial distinction from general time-series forecasting, it would be beneficial to unify the terminology, enhancing clarity and facilitating progress within the field.

3.	Please explicitly state the theoretical framework’s practical implications, detailing how it theoretically informs real-world applications and contributes to advancing the research area.

4.	The experimental validation presented in the manuscript is insufficient, as it employs only two metrics (MSE and MAE). It is advised to repeat experiments multiple times, reporting both mean and standard deviation of the results to ensure robustness.

5.	From the current numerical results, the proposed method does not exhibit significant improvements in forecasting performance. Additional experimental evidence should be provided to clearly demonstrate the advantages of the proposed approach.

6.	Visualization experiments are crucial for time-series forecasting tasks; hence, the authors should consider including them to enhance interpretability and clarity.

---

> ### Author Rebuttal · Authors · 2025-07-31
>
> Dear Review CFrf, we sincerely appreciate your informative feedback and helpful suggestions that helped clarify our contributions, improve the readability of our theories, and the completeness of our experiments. Please see our point-to-point response below.
>
> >Q1: The authors are recommended to verify their citation formatting carefully, as it appears inconsistent with standard practice. Additionally, citations in the main text do not link correctly to the corresponding references listed.
>
> A1: Thank you for pointing this out. We appreciate your careful reading. We have thoroughly checked all citations and corrected inconsistencies. Additionally, we have fixed the incorrect linking between in-text citations and the reference list in the revised manuscript.
>
> >Q2: The authors need to clarify the rationale for emphasizing "online time-series forecasting" explicitly. If there is no substantial distinction from general time-series forecasting, it would be beneficial to unify the terminology, enhancing clarity and facilitating progress within the field.
>
> A2: Thank you for this constructive comment, which helps us better clarify the scope of our work.
>
> The explicit emphasis on **“online time-series forecasting”** is important because the setting differs substantially from general time-series forecasting. In online forecasting:
> - We **cannot access the entire historical data**; instead, we can only use sequentially arriving data streams and rely on the most recent observations within a moving window for forecasting.
> - Handling **nonstationarity** in such a constrained setting is crucial. One of our key theoretical contributions (Theorem 2) proves that **at least 4 consecutive observations** are sufficient to identify the latent variables driving nonstationarity, which aligns directly with the requirements of online forecasting.
>
> We agree that our methodology is general and also applicable to nonstationary time-series forecasting in offline settings. Following your suggestion, we have applied our method to standard nonstationary forecasting tasks, and the experiment results are shown as follows:
>
> |||TOT||koopa||FITS||DLinear+SOILD||
> |---|---|:---:|:---:|:---:|:---:|:---:|:---:|:---:|:---:|
> |Metric||MSE|MAE|MSE|MAE|MSE|MAE|MSE|MAE|
> |ECL|48|0.129|0.228|0.13|0.234|0.147|0.25|0.183|0.262|
> ||96|0.131|0.223|0.136|0.236|0.134|0.256|0.188|0.254|
> ||144|0.147|0.239|0.149|0.247|0.227|0.341|0.185|0.297|
> ||192|0.157|0.25|0.156|0.254|0.198|0.265|0.217|0.284|
> |ETTh2|48|0.225|0.298|0.226|0.3|0.256|0.329|0.268|0.328|
> ||96|0.284|0.34|0.297|0.349|0.315|0.365|0.34|0.361|
> ||144|0.312|0.365|0.333|0.381|0.351|0.391|0.355|0.429|
> ||192|0.336|0.379|0.356|0.393|0.358|0.398|0.387|0.467|
> |Exchange|48|0.042|0.141|0.042|0.143|0.054|0.165|0.047|0.179|
> ||96|0.086|0.205|0.083|0.207|0.096|0.146|0.103|0.25|
> ||144|0.125|0.254|0.13|0.261|0.145|0.304|0.148|0.311|
> ||192|0.164|0.296|0.184|0.309|0.181|0.265|0.2|0.356|
> |ILI|24|1.456|0.778|1.621|0.8|1.633|0.851|1.577|0.845|
> ||36|1.79|0.839|1.803|0.855|1.952|0.969|2.048|0.952|
> ||48|1.746|0.885|1.768|0.903|2.256|1.069|2.029|1.117|
> ||60|1.831|0.89|1.743|0.891|2.084|1.352|1.994|1.003|
> |Traffic|48|0.36|0.231|0.415|0.274|0.482|0.251|0.481|0.374|
> ||96|0.323|0.218|0.401|0.275|0.361|0.234|0.401|0.33|
> ||144|0.315|0.217|0.397|0.276|0.358|0.259|0.428|0.299|
> ||192|0.315|0.22|0.403|0.284|0.359|0.317|0.42|0.297|
>
> In light of your suggestions, we have revised the manuscript to make this distinction clearer while also emphasizing the broader applicability of our framework.
>
> >Q3: Please explicitly state the theoretical framework’s practical implications, detailing how it theoretically informs real-world applications and contributes to advancing the research area.
>
> A3: Thank you for this question. We would like to respectfully clarify that we have already provided detailed discussions of the theoretical framework’s **practical implications** and real-world validity **in the original paper (Lines 175–197) and Appendix C**. Please kindly let us know if you cannot find it. Specifically:
>
> - **Assumption 1 (Bounded and continuous densities):**
> This assumption only requires that the transition probability densities of observed and latent variables are bounded and continuous, which is easy to satisfy in practice. With sufficient data, the estimated probability density functions are naturally bounded. For instance, temperature records in climate data change continuously within a reasonable range.
> - **Assumption 2 (Injectivity of linear operators):**
> This condition only requires sufficient variation in the density of one variable for different distributions of its parent variables. For example:
> $$
> L_{x_{t}, x_{t+1} \mid x_{t-1}, x_{t-2}}
> $$
> implies that historical observations have a strong influence on future observations—e.g., historical temperature values strongly influence future temperature values. Multiple practical examples of injectivity verification are provided in Appendix C.3.
> - **Assumption 3 (Spectral decomposition uniqueness):**
> This assumption requires that changes in the latent variables \(z_t\) produce a sufficiently large and distinguishable effect on the mapping \(x_{t-1} \to x_t\). For example, if \(z_t\) encodes seasonal information and \(x_t\) denotes temperature, the day-to-day temperature change in winter versus summer differs markedly. Since this assumption only requires the existence of any two such distinguishable states, it is satisfied in many real-world scenarios with continuous \(x_t\) (details in Appendix C.6).
> - **Assumption 4 (Sparse mixing procedure):**
> This assumption is mild and common in real-world scenarios. For instance, in financial data, if \(z_{t,i}\) denotes a policy and \(x_t\) denotes stock prices, a single policy typically does not influence all stocks.
> This assumption can also be empirically checked by evaluating the Jacobian matrix of \(x_t\) with respect to \(z_t\) during reconstruction; if the estimated Jacobian is sparse, it validates the sparse mixing procedure.
>
> In light of your comment, we have further emphasized these points in the revised version to make the practical implications of the theoretical framework clearer.
>
> >Q4: The experimental validation presented in the manuscript is insufficient, as it employs only two metrics (MSE and MAE). It is advised to repeat experiments multiple times, reporting both mean and standard deviation of the results to ensure robustness.
>
> A4: Thank you for your suggestion. We would like to clarify that we have already addressed this concern in the current manuscript. Specifically, as stated in **Lines 343–344**, since some methods report the best results in their original papers, we show the best results in the main text. We also provide the experimental results with the mean and variance over three random seeds in Appendix D.2.4. Please kindly let us know if you cannot find it.
>
> Thus, the robustness of our method has been verified, and the corresponding results are reported in the appendix for completeness.
>
> >Q5: From the current numerical results, the proposed method does not exhibit significant improvements in forecasting performance. Additional experimental evidence should be provided to clearly demonstrate the advantages of the proposed approach.
>
> A5: Thank you for your comment. We respectfully disagree with the statement that our method does not exhibit significant improvements in forecasting performance.
>
> As summarized below, we have computed the **average relative improvements** of our method over different backbones and datasets:
>
> |Backbone| MSE Improvement|MAE Improvement|
> |-|-|-|
> |LSTD|1.79%|1.91%|
> |Proceed-T|8.54%|4.85%|
> |OneNet|11.40%|5.88%|
> |OneNet-T|14.14%|12.32%|
> |Online-T|34.31%|31.19%|
>
> These results clearly show that our method consistently improves forecasting performance across **all tested backbones**. Notably, the improvement is larger for relatively older or simpler backbones, demonstrating that our framework is **general and complementary** to different model architectures.
>
> Following your suggestion, we further evaluated additional backbone networks, including ER[1] and DERpp[2]. Experiment results are shown as follows:
>
> |DERpp|DERpp++|ER|ER++|
> |:-:|:-:|:-:|:-:|
> |0.508|0.733|0.508|0.456|
> |0.828|0.741|0.808|0.762|
> |1.157|1.006|1.136|0.926|
> |0.083|0.078|0.086|0.079|
> |0.196|0.191|0.202|0.197|
> |0.208|0.212|0.22|0.183|
> |0.174|0.161|0.18|0.153|
> |0.287|0.281|0.293|0.279|
> |0.294|0.289|0.297|0.284|
>
> [1] Chaudhry,et al. On tiny episodic memories in continual learning.
> [2] Buzzega et al. Dark experience for general continual
> learning: a strong, simple baseline.
>
> >Q6: Visualization Results.
>
> A6: Thank you for this helpful suggestion. We fully agree that visualization is important for improving interpretability in time-series forecasting.
>
> Due to policy restrictions, we are unable to include the figures in the manuscript. However, to address this concern, we have presented the forecasting trends in a tabular format, which conveys the same information as line plots. Additionally, we have included corresponding analyses to highlight the interpretability of the results.
>
> |Timestep|1|2|3|4|5|6|7|8|9|10|11|12|13|14|15|16|17|18|19|20|21|22|23|24|
> |-|-:|-:|-:|-:|-:|-:|-:|-:|-:|-:|-:|-:|-:|-:|---:|---:|---:|---:|---:|-:|-:|-:|-:|-:|
> |GroundTrue|0.078|-0.139|-0.440|-0.693|-0.681|-0.488|0.053|0.174|-0.500|0.029|-0.247|-0.151|0.330|0.427|0.306|0.198|0.679|1.125|0.788|0.571|-0.982|0.090|0.667|0.451|
> |Baseline|-0.917|-0.598|-0.501|-0.970|-1.050|-1.350|-1.257|-1.242|-1.432|-1.292|-1.213|-1.244|-0.894|-1.069|-1.120|-1.377|-1.138|-0.926|-1.011|-0.528|-0.689|-0.768|-0.424|-0.878|
> |Ours|-0.172|-1.015|-0.710|-1.076|-1.273|-1.137|-0.928|-0.857|-0.928|-0.581|-0.401|-0.174|-0.396|-0.244|-0.145|0.128|-0.086|-0.100|0.325|0.209|0.221|-0.137|-0.136|-0.661|
>
> The visualization results from the above table show that our method achieves a better performance.

---

> > ### Comment · Reviewer_CFrf · 2025-08-01
> >
> > Thank you very much for the author's reply.

---

> > > ### Author Response · Authors · 2025-08-09
> > >
> > > Dear Reviewer CFrf,
> > >
> > > Thank you very much for taking the time to review our work. We are sincerely grateful for your constructive comments.
> > >
> > > We are pleased that we have addressed your concerns, and we would be happy to clarify any remaining questions you may have. If you feel that all your concerns have been resolved, we would be grateful if you might consider updating your rating accordingly.
> > >
> > > Thank you again for your thoughtful and invaluable feedback!
> > >
> > > With best wishes,
> > > Authors of submission #8290

---

> ### Author Response · Authors · 2025-08-06
>
> Dear Reviewer CFrf,
>
> Thank you for dedicating time to review and provide feedback on our submission. We hope our responses effectively address your concerns. If there are additional matters you'd like us to consider, we eagerly await the opportunity to respond.
>
> Best regards,
>
> Authors of submission #8290

---

### Official Review · Reviewer_2i9r · 2025-06-18

**Clarity:** 2
**Significance:** 3
**Originality:** 2
**Rating:** 4
**Confidence:** 2

**Summary:**

This paper proposes TOT (Theoretical framework for Online Time-series forecasting), addressing distribution shifts in online time series forecasting caused by latent variables. The work aims to provide theoretical guarantees showing that incorporating latent variables reduces Bayes risk, prove identifiability of latent variables from four consecutive observations, and develop a model-agnostic architecture combining temporal decoders with noise estimators.

**Questions:**

- The paper mentions that "latent variables introduce unknown distribution shifts", can you elaborate? Is the paper also interested in shifts in the data space?

- Can you elaborate more on time-delayed parents and instantaneus parents from Section 2>

- How can practitioners verify assumptions A2-A4 in real datasets? Can you provide diagnostic tools or relaxed conditions? What happens to identifiability guarantees when assumptions are violated, and how robust is the method to such violations?

- What are the computational requirements compared to baseline methods? The architecture involves multiple components (encoder, decoder, two noise estimators, forecaster) - how does this scale with sequence length and dimensionality? Can you provide runtime comparisons?

-  Why are exactly 4 consecutive observations needed for identifiability? Is this the theoretical minimum, or could the framework work with 3 or 5 observations? How does performance degrade with fewer observations?

**Ethical Concerns:**

["NO or VERY MINOR ethics concerns only"]

**Final Justification:**

After the rebuttal,

- (+) I am convinced of the assumptions and their practicality and limitations in real-world settings.
- (+) The contributions of the paper are explained clearly.
- (+) Runtime analysis seems to show the efficiency and low overhead that was added.
- (-) I am skeptical about using the term "shift" in $p(x_t)$.

Hence, I am bumping my score from 3 to 4.

**Limitations:**

Yes

**Quality:**

2

**Strengths And Weaknesses:**

This paper presents a well-executed theoretical framework for online time series forecasting that successfully bridges formal guarantees with practical implementation. The work demonstrates strong technical quality through its coherent three-theorem structure, where Theorem 1 establishes risk reduction via latent variables, Theorem 2 proves block-wise identifiability from just 4 observations, and Theorem 3 shows component-wise identifiability under sparsity constraints. The theoretical development is mathematically fine with clear proofs, and the paper offers an experimental validation through comprehensive synthetic and real-world experiments that consistently support the theoretical claims.

I believe the paper suffers from several significant limitations that constrain its practical impact and theoretical novelty. The identifiability results rely on restrictive assumptions (A1-A4) including injectivity conditions, spectral decomposition uniqueness, and sparse mixing procedures that may not hold in real applications, yet the paper provides no guidance on how practitioners can verify these assumptions or assess their validity. The theoretical contributions are quite incremental, adapting existing eigenvalue-eigenfunction decomposition techniques from causal representation learning rather than developing fundamentally new methods. The work's scope is also limited to discrete, regularly sampled time series, excluding continuous or irregular sampling scenarios.

---

> ### Author Rebuttal · Authors · 2025-07-31
>
> Dear Reviewer 2i9r, we highly appreciate the valuable comments and helpful suggestions on our paper and the time dedicated to reviewing it. Below, please see our point-to-point responses.
>
> >Q1&Q7: As for the explanation of assumptions.
>
> A1&Q7: Thank you for your question. We would like to respectfully clarify that these conditions are standards in the field of latent variables identifiability [1,2,3]. Moreover, we have already provided detailed discussions of the practical implications and real-world validity of these assumptions in the original paper **(Lines 175–197) and Appendix C**. Specifically:
>
> - Bound and Continuous Density: This assumption only requires that the transition probability densities of are bounded and continuous. It is easy to satisfy in practice because, with sufficient data, the estimated probability density functions are naturally bounded. For instance, temperature records in climate data change continuously within a reasonable range. This assumption is naturally satisfied since we use VAE-based model.
>
> - Injectivity: The injectivity condition only requires that there is sufficient variation in the density of one variable for different distributions of its parent variables. For example, $L_{x_{t},x_{t+1}\mid x_{t-1},x_{t-2}}$ implies that historical observations have enough influence on future observations—e.g., historical temperature values strongly influence future temperature values. We further provide multiple examples of the injectivity of linear operators in Appendix C.3. However, this assumption is not testable. In a word, when the mapping from latent variables to observations is not fully injective, our model can only recover the injective components of the latent variables.
>
> - Uniqueness of Spectral Decomposition: This assumption asks that changes in the latent variables $z_t$ produce a sufficiently large and distinguishable effect on the mapping $x_{t-1} \to x_t$. For example, if $z_t$ encodes seasonal information and $x_t$ denotes temperature, the day-to-day temperature jump in winter versus summer differs markedly. Since A3 only requires the existence of any two such distinguishable states, it is satisfied in many real-world scenarios with continuous $x_t$. More details are provided in Appendix C.6. Importantly, this decomposition uniqueness condition is empirically testable. After identifying the latent variables, we can use the observed variables to directly test whether the decomposition is unique by using tensor unique decomposition tools.
>
> - Sparse Mixing Procedure: The Sparse mixing procedure assumption is also common in real-world scenarios. Specifically, it denotes that the Markov network over $z_t$ and $x_t$ is sparse. For example, in the field of financial data, consider $z_{t,i}$ and $x_t$ as one policy and stocks, and this policy may not have an influence on all the stocks. In principle, this assumption is not testable. When it does not hold, we may no longer achieve component-wise identifiability. However, our method can still ensure block-wise identifiability of $z_t$, which is sufficient to model how $z_t$ influences the mapping from $x_{t-1}$ to $x_t$. This allows the framework to remain effective for online forecasting, even when full sparsity is violated.
>
> In light of your comment, we have provided a further discussion in the updated version.
>
> [1] Fu, et al. "Identification of Nonparametric Dynamic Causal Structure and Latent Process in Climate System.
> [2] Lachapelle et al. "Discovering Latent Causal Variables via Mechanism Sparsity: A New Principle for Nonlinear ICA."
> [3] Hu et al. "Instrumental variable treatment of nonclassical measurement error models."
>
> >Q2: As for the contribution.
>
> A2: Thank you for your comment, which helps us highlight our contributions. We would like to clarify that our contribution lies in addressing a fundamentally different and practically motivated **real-world problem**. Specifically, we aim to address the online time series forecasting, which has been rarely handled in the field of time series forecasting. To address this problem, we have to formulate the specific model and then tailor the techniques, like eigenvalue–eigenfunction decomposition, and incorporate further constraints to achieve our identifiability results.
>
> Moreover, we would like to highlight the contributions of Theorems 1 and 3, which are, to the best of our knowledge, entirely new:
>   1. **Theorem 1** formally establishes how latent variables affect predictive performance under nonstationarity.
>   2. **Theorem 3** introduces the sparse mixing procedure into temporal causal representation learning and proves that latent variables are identifiable under this mild and practically meaningful assumption.
>
> >Q4: As for the work's scope.
>
> A4: Thank you. We would like to clarify that we have already explicitly acknowledged this limitation in the manuscript (see **Lines 370-372**). Our current theoretical analysis and experiments focus on regularly sampled time series, as it allows for a clear formulation of the identifiability conditions.
>
> Importantly, within this scope, our framework is general; it accommodates a wide range of time series data with a discrete time step. And we believe it is unrealistic to expect a single theoretical framework to cover all possible data types, especially in the early stages of developing identifiability theory for latent-variable models. Extending our framework to continuous or irregularly sampled time series is indeed an interesting direction, and we have emphasized this as future work in the revised manuscript.
>
> >Q5: The paper mentions that "latent variables introduce unknown distribution shifts", can you elaborate? Is the paper also interested in shifts in the data space?
>
> A5: Thanks a lot. By "latent variables introduce unknown distribution shifts" we refer to the fact that the evolution of latent variables influences the causal process of observed variables, which in turn changes their marginal distribution over time. Specifically, in the data generation process
> $$
> \\begin{array}{ccc}
> z_{t} & \\rightarrow & z_{t+1} \\\\
> \\downarrow & & \\downarrow \\\\
> x_{t} & \\rightarrow & x_{t+1}
> \\end{array}
> $$
> the transition $x_t \rightarrow x_{t+1}$ is governed by $z_t$. As $z_t$ evolves, the marginal distribution \(p(x_t)\) also changes over time, leading to distribution shifts. Because \(z_t\) is unobserved, these distribution shifts are "unknown".
>
> Regarding the second part of your question, our work focuses on **distribution shifts in the data space (observed variables)**. We have explicitly highlighted  how latent variables have influence on the evolution of the observed distribution in the updated version.
>
> >Q6: As for time-delayed and instantaneus parents
>
> A6: Thank you for your question. Let us consider the following causal graph:
> $$
> \\begin{array}{ccc}
> z_{t,1} & \\rightarrow & z_{t+1,1} \\\\
> \\downarrow &  & \\downarrow \\\\
> z_{t,2} & \\rightarrow & z_{t+1,2}
> \\end{array}
> $$
> In this example:
> - Time-delayed parents:
> The historical latent variables, such as $z_{t,1}$ and $z_{t,2}$, are time-delayed parents of $z_{t+1,1}$ and $z_{t+1,2}$, respectively, because they influence these variables **through temporal transitions** $z_{t,1} \to z_{t+1,1}$
>
> - Instantaneous parents:
> At the current time step $t$, $z_{t,1}$ is an instantaneous parent of $z_{t,2}$, as it directly influences $z_{t,2}$ within the same timestep. Similarly, $z_{t+1,1}$ is an instantaneous parent of $z_{t+1,2}$.
>
> >Q8: As for computational requirements compared to baseline methods.
>
> A8: Thanks! We have provided a detailed clarification regarding computational requirements, scalability with sequence length and dimensionality, and runtime comparisons.
>
> 1. **Experimental environment:**
> All experiments were conducted on a single NVIDIA GTX 3090 (24GB) GPU and an Intel i7-7700k CPU with 64GB memory.
> 2. **Scalability of the architecture:**
> As discussed in Line 252, our framework is **model-agnostic** and can be combined with different backbone networks. The computational scaling thus mainly depends on the chosen backbone:
>    - **Encoder and forecaster:** Their scalability with sequence length and input dimensionality follows the standard complexity of the selected backbone (e.g., LSTD or OneNet).
>    - **Decoder:** The decoder is implemented as a simple MLP and is independent of sequence length; it only scales with the output dimensionality.
>    - **Noise estimators:** Each latent variable dimension corresponds to two noise estimators. Thus, the complexity of these components grows **linearly** with the latent dimensionality.
> 3. **Runtime comparisons:**
> We have conducted runtime evaluations across different datasets and backbones to provide a fair comparison with baseline methods. The table below shows the average runtime (seconds) per training epoch for LSTD and Proceed backbones.
>
> |Model|LSTD|LSTD+TOT|Proceed-T|Proceed-T+TOT|
> |-|-|-|-|-|
> |ETTh2|12.969|6.122|28.513|55.02|
> |Exchange|7.701|3.351|12.12|13.352|
>
> >Q9: 4 consecutive observations for identifiability.
>
> A9: Thank you for this insightful question. As discussed in **Lines 152–159**, Theorem 2 implies that at least 4 consecutive observations are theoretically required to describe the variation of the latent variables $z_t$. This is the theoretical minimum for identifiability in our framework. Specifically, with only 3 consecutive observations, the distributional changes are insufficient to uniquely recover the latent transitions, making $z_t$ unidentifiable under the assumptions in Theorem 2. And using more than 4 observations (e.g., 5) is allowed and does not violate the theoretical framework. Following your suggestion, we conducted additional experiments by varying the number of observations used for identification. Experiment results are shown as follows:
>
> |len(x)| MCC|
> |-|-|
> |1|0.7232(0.1784)|
> |2|0.8435(0.1326)|
> |3|0.9224(0.0841)|
> |4|0.9543(0.0541)|
> |5|0.9578(0.0439)|

---

> > ### Comment · Reviewer_2i9r · 2025-08-02
> >
> > I thank the authors for their replies. I have no further questions.

---

> > > ### Author Response · Authors · 2025-08-06
> > >
> > > Dear Reviewer 2i9r,
> > >
> > > We are glad to address your concerns. In light of your treasurable suggestions, we have provided more experiments about different lengths of observations, more details of model implementation, and highlighted our contributions in the updated version.
> > >
> > > With best wishes,
> > >
> > > Authors of submission #8290

---

> ### Author Response · Authors · 2025-08-09
>
> Dear Reviewer 2i9r,
>
> Thank you very much for taking the time to review our work. We are sincerely grateful for your constructive comments.
>
> We feel delighted that your concerns have been fully addressed. We would be grateful if you might consider updating your rating accordingly. Thank you again for your thoughtful and invaluable feedback!
>
> With best wishes,
>
> Authors of submission #8290

---

### Official Review · Reviewer_hiQT · 2025-07-01

**Clarity:** 4
**Significance:** 3
**Originality:** 4
**Rating:** 5
**Confidence:** 3

**Summary:**

This paper proposes a theoretical framework for online time series forecasting (TOT) to address unknown distribution shifts caused by latent variables. The authors theoretically prove that incorporating latent variables tightens the Bayes risk, with the benefit increasing as latent variable identifiability improves. They show that latent variables and their causal dynamics can be uniquely identified using only four consecutive observations. Based on these results, a model-agnostic architecture is designed, featuring a temporal decoder and two noise estimators. Experiments on synthetic and real-world datasets validate the theoretical claims, demonstrating improved forecasting performance across multiple benchmarks.

**Questions:**

1. Will the sampling frequency affect this module, thus causing the assumption conditions to be violated and making four observation points insufficient for identifying the latent variables?
2. This framework seems very universal. Can it be used for offline model training?

**Ethical Concerns:**

["NO or VERY MINOR ethics concerns only"]

**Limitations:**

The authors wrote about the limitation in the conclusion.

**Quality:**

3

**Strengths And Weaknesses:**

### Strengths
1. **Rigorous Theoretical Foundation**: The paper provides a solid theoretical framework, including risk-bound guarantees and identifiability proofs for latent variables. The derivation of using four adjacent observations to identify latent variables is innovative and addresses a key challenge in nonstationary time series modeling. The theoretical results are well-supported by mathematical proofs (Appendix C), enhancing the credibility of the framework.

2. **Data Efficiency and Generalizability**: The method’s ability to identify latent variables with minimal observations (four consecutive points) makes it highly data-efficient, which is valuable for real-world scenarios with limited data. The model-agnostic blueprint allows integration with various backbones (e.g., OneNet, LSTD), demonstrating broad applicability and practical utility across different forecasting tasks.

3. **Empirical Validation**: The experiments are comprehensive, covering both synthetic and real-world datasets (e.g., ETT, Exchange, Traffic). The results show consistent improvements over baselines, with clear performance metrics (MSE, MAE) and statistical significance (repeated runs with error bars). The ablation study (Figure 4) further validates the necessity of each component, providing insights into the model’s design.

### Weaknesses
1. **Assumption Constraints**: The theoretical results rely on assumptions such as bounded continuous densities (A1) and injective linear operators (A2), which may not hold in highly nonstationary or noisy real-world scenarios. For example, abrupt distribution shifts (e.g., sudden policy changes) could violate the continuity assumption, potentially limiting the framework’s robustness in extreme cases.

---

> ### Author Rebuttal · Authors · 2025-07-31
>
> Dear Reviewer hiQT, we sincerely appreciate your encouraging and insightful feedback, which has significantly helped us clarify the boundary of our method and make our experiment solid. Please find our point-by-point responses below.
>
>
> >Q1: The theoretical results rely on assumptions such as bounded continuous densities (A1) and injective linear operators (A2), which may not hold in highly nonstationary or noisy real-world scenarios. For example, abrupt distribution shifts (e.g., sudden policy changes) could violate the continuity assumption, potentially limiting the framework’s robustness in extreme cases.
>
> A1: Thank you for raising this important concern. We would like to clarify that the continuity assumption in (A1) refers specifically to the probability density functions of the latent and observed variables, not to the continuity of the data generation process. Therefore, even in the presence of abrupt distribution shifts, such as sudden policy changes, our theoretical results remain valid as long as the latent and observed variables admit bounded continuous densities. Moreover, such conditions still hold in many realistic nonstationary or noisy settings, where the the distributions do not become degenerate.
>
>
> >Q2: Will the sampling frequency affect this module, thus causing the assumption conditions to be violated and making four observation points insufficient for identifying the latent variables?
>
> A2: Thank you for the thoughtful question. The sampling frequency can indeed affect the identifiability of latent variables, particularly when it induces a form of subsampling in which some observation variables become unobservable. In such cases, the corresponding latent variables may no longer be fully identifiable unless additional observations are incorporated.
>
> For example, consider the following generative structure:
> $$
> \\begin{array}{ccccccccccc}
> z_1 & \\rightarrow & z_2 & \\rightarrow & z_3 & \\rightarrow & z_4 \\\\
> \\downarrow & & \\downarrow & & \\downarrow & & \\downarrow \\\\
> x_1 & \\rightarrow & x_2 & \\rightarrow & x_3 & \\rightarrow & x_4
> \\end{array}
> $$
> If $x_2$ becomes unobservable due to subsampling or sensor failure, then the identifiability of $z_1$ and $z_3$ may require additional observable variables beyond the original four observed variables. However, for online forecasting tasks, even if some latent variables are only partially identified, our method can still leverage the identifiable components to make effective predictions of future observations.
>
>
> >Q3: This framework seems very universal. Can it be used for offline model training?
>
>
> A3: Thank you for the great suggestion, which extends the boundary of our contributions. Indeed, our methodology is universal and can naturally be applied to offline model training, where the distributions of historical time series change over time. In light of your questions, we conducted additional experiments on the offline training scenarios with non-stationary time series. Specifically, we follow the setting of [1], and consider Koopa [1], FITS [2], SOILD [3] as baselines. Moreover, we evaluate our method on ECL, ETTh2, Exchange, ILI, and Traffic datasets. Experiment results are shown as follows:
>
> |  |  | TOT |  | Koopa |  | FITS |  | DLinear+SOLID |  |
> |---|---|:---:|:---:|:---:|:---:|:---:|:---:|:---:|:---:|
> | Metric |  | MSE | MAE | MSE | MAE | MSE | MAE | MSE | MAE |
> | ECL | 48 | 0.130 | 0.228 | 0.13 | 0.234 | 0.147 | 0.25 | 0.183 | 0.262 |
> |  | 96 | 0.131 | 0.223 | 0.136 | 0.236 | 0.134 | 0.256 | 0.188 | 0.254 |
> |  | 144 | 0.147 | 0.239 | 0.149 | 0.247 | 0.227 | 0.341 | 0.185 | 0.297 |
> |  | 192 | 0.157 | 0.25 | 0.156 | 0.254 | 0.198 | 0.265 | 0.217 | 0.284 |
> | ETTh2 | 48 | 0.225 | 0.298 | 0.226 | 0.3 | 0.256 | 0.329 | 0.268 | 0.328 |
> |  | 96 | 0.284 | 0.34 | 0.297 | 0.349 | 0.315 | 0.365 | 0.34 | 0.361 |
> |  | 144 | 0.312 | 0.365 | 0.333 | 0.381 | 0.351 | 0.391 | 0.355 | 0.429 |
> |  | 192 | 0.336 | 0.379 | 0.356 | 0.393 | 0.358 | 0.398 | 0.387 | 0.467 |
> | Exchange | 48 | 0.042 | 0.141 | 0.042 | 0.143 | 0.054 | 0.165 | 0.047 | 0.179 |
> |  | 96 | 0.086 | 0.205 | 0.083 | 0.207 | 0.096 | 0.146 | 0.103 | 0.25 |
> |  | 144 | 0.125 | 0.254 | 0.13 | 0.261 | 0.145 | 0.304 | 0.148 | 0.311 |
> |  | 192 | 0.164 | 0.296 | 0.184 | 0.309 | 0.181 | 0.265 | 0.2 | 0.356 |
> | ILI | 24 | 1.456 | 0.778 | 1.621 | 0.8 | 1.633 | 0.851 | 1.577 | 0.845 |
> |  | 36 | 1.79 | 0.839 | 1.803 | 0.855 | 1.952 | 0.969 | 2.048 | 0.952 |
> |  | 48 | 1.746 | 0.885 | 1.768 | 0.903 | 2.256 | 1.069 | 2.029 | 1.117 |
> |  | 60 | 1.831 | 0.89 | 1.743 | 0.891 | 2.084 | 1.352 | 1.994 | 1.003 |
> | Traffic | 48 | 0.36 | 0.231 | 0.415 | 0.274 | 0.482 | 0.251 | 0.481 | 0.374 |
> |  | 96 | 0.323 | 0.218 | 0.401 | 0.275 | 0.361 | 0.234 | 0.401 | 0.33 |
> |  | 144 | 0.315 | 0.217 | 0.397 | 0.276 | 0.358 | 0.259 | 0.428 | 0.299 |
> |  | 192 | 0.315 | 0.22 | 0.403 | 0.284 | 0.359 | 0.317 | 0.42 | 0.297 |
>
> We can find that our method can still achieve the ideal performance in offline scenarios.
>
> [1] Liu, Yong, et al. "Koopa: Learning non-stationary time series dynamics with Koopman predictors." NeurIPS 2023
> [2] Xu, Zhijian, Ailing Zeng, and Qiang Xu. "FITS: Modeling time series with $10 k $ parameters."ICLR2024
> [3] Chen, Mouxiang, et al. "Calibration of time-series forecasting: Detecting and adapting context-driven distribution shift." KDD2024.

---

> > ### Comment · Reviewer_hiQT · 2025-08-04
> >
> > Thanks for the authors' responses to my review comments and questions, and I have no further questions. I will keep my score.

---

> > > ### Author Response · Authors · 2025-08-06
> > >
> > > Dear Reviewer
> > >
> > > We are glad to address your confusion of theories and thank you for the valuable suggestions. We will provide more experiments about nonstationary forecasting and a discussion about sampling frequency.
> > >
> > > With best wishes,
> > >
> > > Authors of submission #8290

---

### Official Review · Reviewer_uFSc · 2025-07-04

**Clarity:** 3
**Significance:** 3
**Originality:** 3
**Rating:** 5
**Confidence:** 3

**Summary:**

The focus of this paper is forecasting time series online, in the presence of distribution shifts that are governed by some kind of unobserved latent dynamics. Here, by "distribution shifts", the authors refer to both changes in the latent state as well as changes in how the observed variables depend on the latent state. In contrast to previous works, the authors consider the possibility of there being a sparse dependence structure of the observed variables on the latent variables at a given point in time. This, in turn, leads to identifiability of the latent variables and is shown to have beneficial downstream effects on forecasting performance. Put differently, it is more beneficial to have a sparse dependence structure that is identifiable than not using a dependence structure at all, in the presence of distribution shift. When there is no distribution shift, it is shown that the use of such a model does not result in performance decay. The paper also looks at some theoretical results having to do with identifiability and concludes with empirical examples that illustrate the performance of the proposed methodology.

**Questions:**

It would be nice to have a cleaner description of the "online" component: how are updates done once a single new data point comes in?

Following up on the above. Can we try online learning in non-stationary environments? This methodology seems well-suitable for this kind of approach. And, do multi-step-ahead forecasting in such environments would be the ultimate robustness test of this methodology.

Do we explicitly care about the dynamics of the z_t's at any point? Or only about which z_t's are relevant at t?

**Ethical Concerns:**

["NO or VERY MINOR ethics concerns only"]

**Final Justification:**

The authors' comprehensive response and additional experiments have helped me understand aspects of the paper better, and as a result, I have adjusted my score.

**Limitations:**

Yes.

**Paper Formatting Concerns:**

None.

**Quality:**

3

**Strengths And Weaknesses:**

Strengths: the model's structural assumptions seem well-founded and the framework appears to be able to handle scenarios with fairly complex distribution shifts. Careful consideration is given to the identifiability problem of latent variables. The authors also consider the problem of predictive risk and its relationship with the latent variable structure. The use of both observations and latent variables in constructing a prediction leads to more accurate forecasting possibilities.

Weaknesses: the authors should spend a bit more discussing a simple experiment within the confines of the main text. No details are given on the data generation process, e.g., time series length, the type of sparsity, and so on. This makes it hard to develop an intuition for how the method works. The "sparsity of the mixing procedure" should also be made more clear. The way I understood it, the sparsity is in having each z_i only influence a few of the x_i's. Is this indeed the case? This would mean that each z_i must be "active" but at the same time there is a limitation on how many x_i's it can be connected to? So even if the distribution changes in a way that some x_i becomes highly correlated with a block of x_i's driven by a latent variable, the sparsity constraint can prevent this x_i from being associated with that z_i?

---

> ### Author Rebuttal · Authors · 2025-07-31
>
> Dear Reviewer uFSc, we sincerely appreciate your informative feedback and helpful suggestions that make our paper more complete. We have added the new experiments following your suggestions, provided more discussions about the sparse mixing procedure and online forecasting, as well as modified the paper and appendix accordingly. Please see our point-to-point response below.
>
> >Q1. The authors should spend a bit more discussing a simple experiment within the confines of the main text. No details are given on the data generation process, e.g., time series length, the type of sparsity, and so on. This makes it hard to develop an intuition for how the method works.
>
> A1: Thanks for your insightful suggestions, which make our experiment more solid. In light of your suggestion, we have added more simulation experiments. Specifically, we have considered four datasets with different types of sparsity: (1). the sparse latent process but the dense mixing procedure; (2). the dense latent process but the sparse mixing procedure; (3). the sparse latent process and the sparse mixing procedure; and (4). The dense latent process and the dense mixing procedure. All of these datasets contain 10000 5-dimensional time series samples with a length of 7. Experiment results are shown as follows:
>
> |dataset|MCC|
> |-|-|
> |(1)|0.9521(0.032)|
> |(2)|0.9432(0.019)|
> |(3)|0.9542(0.085)|
> |(4)|0.8143(0.173)|
>
> We can find that the latent variables are hard to identify when both the mixing and latent dynamics are sparse.
>
> >Q2: The "sparsity of the mixing procedure" should also be made more clear. The way I understood it, the sparsity is in having each z_i only influence a few of the x_i's. Is this indeed the case? This would mean that each z_i must be "active" but at the same time there is a limitation on how many x_i's it can be connected to?
>
> A2: Thank you for your question, which indeed helps us improve the readability of our paper. You are correct that the sparse mixing procedure means each $z_{t,i}$ only influences a limited number of $x_{t,j}$. However, we would like to clarify that our sparsity requirement is mild. Specifically, we only assume that the Markov network over $z_t, x_t$ is sparse, rather than imposing strong structural sparsity across the entire latent process. Given a toy example as follows
> $$
> \\begin{array}{ccc}
> z_{t-1,1} & \\leftarrow & z_{t-1,2} \\\\
> \\downarrow & \\searrow & \\downarrow \\\\
> z_{t,1} & \\leftarrow & z_{t,2} \\\\
> \\downarrow & \\searrow & \\downarrow \\\\
> x_{t,1} & \\leftarrow & x_{t,2}
> \\end{array}
> $$
> ,the sparsity assumption is already satisfied although $z_{t,1}$ is fully connected to $x_{t,1}$ and $x_{t,2}$. This is because the corresponding Markov network is shown as follows
> $$
> \\begin{array}{ccc}
> z_{t-1,1} &—& z_{t-1,2} \\\\
> | & \\diagdown & | \\\\
> z_{t,1} &—& z_{t,2} \\\\
> | & \\diagdown & | \\\\
> x_{t,1} &—& x_{t,2}
> \\end{array}
> $$
> is also sparse. And the intimate neighbor set (defined in Line 202-204) of latent variables $z_{t,1}$ and $z_{t,2}$  are empty sets, which implies identifiability by using sufficient changes from historical information.
>
> This mild assumption is sufficient to guarantee the component-wise identifiability of latent variables, as shown in Theorem 3. In light of your insightful question, we have provided a detailed explanation with the aforementioned graphical example in the updated version.
>
> >Q3: So even if the distribution changes in a way that some x_i becomes highly correlated with a block of x_i's driven by a latent variable, the sparsity constraint can prevent this x_i from being associated with that z_i?
>
> A3: Thanks for your question! We would like to highlight that the sparsity constraint does not mean to prevent some observations from being associated with others.
>
> Let us using the same example in A2, $x_{t,1}$ is indirectly influenced by $z_{t,2}$ through $x_{t,2}$. In this case, $x_{t,1}$ is associated with $z_{t,2}$ even if the data generation process is sparse.
>
> >Q4: It would be nice to have a cleaner description of the "online" component: how are updates done once a single new data point comes in?
>
> A4: Thank you for this helpful suggestion. We follow the definition of online time series forecasting from [1,2], where the forecasting model generates future predictions using a moving window and updates the model immediately after the corresponding ground truth time series data arrive.
>
> For a better understanding, we assume a first-order Markov process for both observations and latent variables as follows:
> $$
> \\begin{aligned}
> & z_1 \\rightarrow z_2 \\rightarrow z_3 \\rightarrow z_4 \\rightarrow z_5 \\rightarrow z_6 \\\\
> & \\downarrow\\quad\\;\\;\\;\\downarrow\\\;\\;\\;\\quad\downarrow\\quad\\\;\\;\\;\\downarrow\\quad\\;\\;\\;\\downarrow\\quad\\;\\;\\;\\;\\downarrow \\\\
> & x_1 \\rightarrow x_2 \\rightarrow x_3 \\rightarrow x_4 \\rightarrow x_5 \\rightarrow x_6
> \\end{aligned}
> $$
> where we use the new arrived $(x_5)$ and the historical observations in moving window, e.g. $(x_1,x_2,x_3,x_4)$ to predict the values of $x_6$.
> Based on this example, the update procedure proceeds as follows:
> 1. **Latent Variables Identification**: When new data arrive, we use only the most recent adjacent observations to identify the latent variables. For example, given $(x_1, x_2, x_3, x_4, x_5)$, we can estimate $\hat{z}_3$ and $\hat{z}_4$ from $(x_1, x_2, x_3, x_4)$ and $(x_2, x_3, x_4, x_5)$, respectively (leverage our consecutive four observed variables).
> 2. **Latent Variables Estimation**: Online transition update: With the newly identified latent states, we learn the latent transition $p(z_t|z_{t-1})$ incrementally (e.g., updating from $(z_3,z_4)$ to infer $z_5, z_6$) as the transition of latent variables is stationary.
> 3. **Prediction with Latent Variables**: Finally, we combine the estimated latent variables $z_6$ with the latest available observations, i.e., $(x_1, x_2, x_3, x_4, x_5)$ to forecast the future observation $x_6$.
>
> Thanks for your insightful question. We have provided a detailed example and an algorithm flow chart in the updated version.
>
> [1] Lau, Ying-yee Ava, Zhiwen Shao, and Dit-Yan Yeung. "Fast and Slow Streams for Online Time Series Forecasting Without Information Leakage." ICLR 2025.
> [2] Zhao, Lifan, and Yanyan Shen. "Proactive model adaptation against concept drift for online time series forecasting." KDD 2025.
>
> >Q5: Following up on the above. Can we try online learning in non-stationary environments? This methodology seems well-suitable for this kind of approach. And, do multi-step-ahead forecasting in such environments would be the ultimate robustness test of this methodology.
>
> A5: Thank you for this insightful suggestion. Indeed, our methodology is general and naturally applicable to non-stationary environments. However, we would like to clarify the difference between our online forecasting setting and conventional non-stationary forecasting. In the nonstationary time series forecasting, we can leverage more historical time series data. In online forecasting, we cannot leverage the entire historical data; instead, we only have access to sequentially arriving data streams and must rely on the most recent observations for updating. Our theoretical contribution lies in showing how to identify the latent variables that drive non-stationarity with minimal adjacent observations, which is critical for real-time online updates.
>
> We appreciate your suggestion to further evaluate our method on non-stationary time series forecasting. Specifically, we follow the experimental setting of [1] and use Koopa[1], FITS[2], and SOILD[3] as baselines. We evaluate our method on the ECL, ETTh2, Exchange, ILI, and Traffic datasets. The results are presented below.
>
> |||TOT||Koopa||FITS||SOILD||
> |-|-|:-:|:-:|:-:|:-:|:-:|:-:|:-:|:-:|
> |Metric||MSE|MAE|MSE|MAE|MSE|MAE|MSE|MAE|
> |ECL|48|0.129|0.228|0.13|0.234|0.147|0.25|0.183|0.262|
> ||96|0.131|0.223|0.136|0.236|0.134|0.256|0.188|0.254|
> ||144|0.147|0.239|0.149|0.247|0.227|0.341|0.185|0.297|
> ||192|0.157|0.25|0.156|0.254|0.198|0.265|0.217|0.284|
> |ETTh2|48|0.225|0.298|0.226|0.3|0.256|0.329|0.268|0.328|
> ||96|0.284|0.34|0.297|0.349|0.315|0.365|0.34|0.361|
> ||144|0.312|0.365|0.333|0.381|0.351|0.391|0.355|0.429|
> ||192|0.336|0.379|0.356|0.393|0.358|0.398|0.387|0.467|
> |Exchange|48|0.042|0.141|0.042|0.143|0.054|0.165|0.047|0.179|
> ||96|0.086|0.205|0.083|0.207|0.096|0.146|0.103|0.25|
> ||144|0.125|0.254|0.13|0.261|0.145|0.304|0.148|0.311|
> ||192|0.164|0.296|0.184|0.309|0.181|0.265|0.2|0.356|
> |ILI|24|1.456|0.778|1.621|0.8|1.633|0.851|1.577|0.845|
> ||36|1.79|0.839|1.803|0.855|1.952|0.969|2.048|0.952|
> ||48|1.746|0.885|1.768|0.903|2.256|1.069|2.029|1.117|
> ||60|1.831|0.89|1.743|0.891|2.084|1.352|1.994|1.003|
> |Traffic|48|0.36|0.231|0.415|0.274|0.482|0.251|0.481|0.374|
> ||96|0.323|0.218|0.401|0.275|0.361|0.234|0.401|0.33|
> ||144|0.315|0.217|0.397|0.276|0.358|0.259|0.428|0.299|
> ||192|0.315|0.22|0.403|0.284|0.359|0.317|0.42|0.297|
>
> We can find that our method can still achieve ideal performance in non-stationary environments.
>
> [1] Liu. et al. "Koopa: Learning non-stationary time series dynamics with koopman predictors." NeurIPS 2023
> [2] Xu. et al. "FITS: Modeling time series with $10 k $ parameters."ICLR2024
> [3] Chen et al. "Calibration of time-series forecasting: Detecting and adapting context-driven distribution shift."KDD2024.
>
> >Q6: Do we explicitly care about the dynamics of the $z_t$'s at any point? Or only about which $z_t$'s are relevant at t?
>
> A6: Thank you for this thoughtful question. We do care about the dynamics of $z_t$ and we explicitly model the dynamics of $z_t$ in **Section 4.1**. Specifically, when identifying the latent variables, we explicitly estimate the independent noise terms across different time steps, i.e., $\epsilon_{t,i} = r(z_{t,i}, z_{t-1})$. This procedure essentially models the temporal dynamics $z_{t-1}\rightarrow z_t$. In light of your questions, we have highlighted it in the final version.

---

> > ### Comment · Reviewer_uFSc · 2025-08-05
> >
> > Dear authors, thank you for the additional explanations and experiments. I am satisfied with these, and I will adjust my rating accordingly.

---

> > > ### Author Response · Authors · 2025-08-06
> > >
> > > Dear Reviewer uFSc,
> > >
> > > We are delighted that you found the response well addressed your concerns. Thank you once again for your valuable comments and suggestions!
> > >
> > > With best wishes,
> > >
> > > Authors of submission #8290

---

### Author Response · Authors · 2025-08-08
**Thank You Letter and General Response**

Comment: Dear Reviewers and ACs,

We sincerely thank you for your time, effort, and thoughtful insights throughout both the initial review and discussion phases. Your feedback has been invaluable in refining and clarifying our work.

We are grateful for your recognition of our contributions. In particular, Reviewer hiQT acknowledged the strength and data-efficiency of our theoretical framework, as well as the comprehensiveness of our experiments. Reviewer uFSc noted that our theoretical results are well-founded. We also appreciate that Reviewer 2i9r did not raise further concerns after our rebuttal.

While there might be some points of disagreement, we value these as part of the scientific process. We believe we are sharing the common goal of improving and clarifying the work, ensuring key ideas are clearly conveyed, and aligning on its contributions. We would therefore like to use this letter to also provide a general response to several points and briefly outline the additions we will make in the updated version.


**As for the contribution**: Our main contribution lies in presenting a unified theoretical framework for online time series forecasting, which addresses a fundamentally different and practically motivated problem. To the best of our knowledge, this is the first theoretical framework in this domain. While some of our mathematical tools are inspired by prior work, Theorem 2, as noted in Lines 152–159, significantly generalizes prior formulations by allowing causal influence across different time steps and removing restrictive assumptions such as monotonicity and normalization. These extensions make our approach more suitable for real-world applications. Moreover, Theorems 1 and 3 are entirely novel contributions.


**As for explanation of assumptions**: We have explicitly provided detailed discussions of the practical implications and real-world validity of these assumptions in the original paper (**Lines 175–197**) and **Appendix C**. And we hope the reviewers can find them. Moreover, we have provided more explanation in the rebuttals.


**As for the limited work's scope**: While we acknowledge that our method does not address all possible real-world scenarios (e.g., continuous or irregular sampling), we respectfully note that it is unrealistic to expect a single theoretical framework to cover all settings. Our framework targets a broad and practically relevant case of time series data with discrete timestamps and provides a general solution for such cases, as highlighted by Reviewers uFSc and hiQT.


**As for the implementation details**: We would like to clarify that our method is model-agnostic, and its scalability with respect to sequence length and dimensionality follows that of existing models such as LSTD and Proceed.


**As for the experiment results**: We appreciate that Reviewers hiQT and 2i9r recognized the consistent improvements of our method over strong baselines. We are also encouraged by the positive remarks regarding the method’s generality and applicability to nonstationary forecasting. Motivated by these suggestions, we have devoted substantial effort to extend our experiments to nonstationary settings, and we hope these results will further demonstrate the breadth and strength of our approach.


We sincerely hope that our rebuttal and this general response have helped clarify the contributions and address any remaining concerns. As the discussion phase draws to a close, we kindly invite any final questions or suggestions, and we will be glad to respond promptly.

Once again, we deeply thank all reviewers and the Area Chairs for your time, effort, and constructive feedback. Your insights have been essential in improving the clarity and strength of our work.

Warm regards,

Authors of submission #8290

---

### Note · Authors · 2025-08-11

Dear Reviewers and AC,

We are grateful for the time and effort you devoted to the review and discussion phases. For ease of the final discussion, we summarize our responses as follows:

----
**Contributions:** We propose a unified framework for online time series forecasting. Specifically, we (i) establish formal risk-bound guarantees under latent variables, (ii) provide theory that provably identifies latent variables, and (iii) develop a model-agnostic method to align theory with practice.

----
Reviewer **uFSc** finds our assumptions **well-founded** and recognizes that the framework can handle scenarios with complex distribution shifts. The reviewer suggested adding a simple illustrative experiment, conducting real-world experiments on nonstationary scenarios, and providing a clearer description.

In the rebuttal, we addressed these points by adding more experiments, clarifying the explanations, and updating the manuscript. The reviewer was **satisfied** with our answers and adjusted the rating accordingly.

---
Reviewer **hiQT** considers our **rigorous** theoretical foundation, broad **applicability**, and **comprehensive** experiments. Following the suggestions, we demonstrated that several assumptions hold in diverse real-world scenarios and added experiments on nonstationary forecasting. The reviewer **had no further questions** and maintained a positive score.

---
Reviewer **2i9r** considers our theoretical framework **well-executed**, with clear proofs. In the rebuttal, we clarified the practical implications of our assumptions, addressed a **misunderstanding** regarding the theoretical contributions, and provided additional experiments. The reviewer also **had no further questions**, and we believe our responses addressed the concerns.

---
We believe there may be a **misunderstanding** from Reviewer **CFrf**. Specifically, the practical implications and real-world validity of our theoretical framework were **already discussed** in the original submission. Additionally, the mean and standard deviation were **explicitly** reported.

Furthermore, we conducted extensive experiments, including results on standard nonstationary forecasting tasks and with additional backbone networks, consistently showing general improvement. We hope the reviewer can take note of these points.

Thanks again for your time and effort in reviewing this paper. We hope our responses have properly addressed your concerns.

With best regards,

Authors of submission #8290

---

### Decision · Program_Chairs · 2025-09-17

**Decision:**

Accept (poster)

**Comment:**

The paper introduces a framework for online time series forecasting under latent-driven distribution shifts, with theoretical guarantees on reduced risk and identifiability, supported by empirical validation.  Its main strengths are strong theoretical grounding, novel identifiability results, model-agnostic design, and consistent empirical improvements.  Weaknesses include restrictive assumptions, clarity issues on the online component and sparsity, and initially limited experimental reporting.  The decision to accept is based on the combination of rigorous theory and practical relevance, which outweigh incremental aspects and assumptions.  During rebuttal, the authors provided clarifications, additional experiments, and stronger justifications, satisfying most reviewers; only one (less-engaged) reviewer remained skeptical, but consensus leaned toward recognizing the work’s solid and impactful contributions.